# Proximity-labeling proteomics reveals remodeled interactomes and altered localization of pathogenic SHP2 variants

Anne E van Vlimmeren [1,2], Lauren C Tang[2], Ziyuan Jiang[1], Abhishek Iyer[2], Rashmi Voleti[1], Konstantin Krismer[3], Jellert T Gaublomme[2], Marko Jovanovic[2] & Neel H Shah [1,4]✉

## Abstract

Missense mutations in *PTPN11*, which encodes the protein tyrosine phosphatase SHP2, are common in several developmental disorders and cancers. While many mutations disrupt auto-inhibition and hyperactivate SHP2, several do not enhance catalytic activity. Both activating and non-activating mutations could potentially drive pathogenic signaling by altering SHP2 interactions or localization. We employed proximity-labeling proteomics to map the interaction networks of wild-type SHP2, ten clinically relevant mutants, and SHP2 bound to an inhibitor that stabilizes its auto-inhibited state. Our analyses reveal mutation- and inhibitor-dependent alterations in the SHP2 interactome, with several mutations also changing localization. Some mutants show increased mitochondrial localization and impact mitochondrial function. This study provides a resource for exploring SHP2 signaling and offers new insights into the molecular basis of SHP2-driven diseases. Furthermore, this work highlights the capacity for proximity-labeling proteomics to detect missense-mutation-dependent changes in protein interactions and localization.

**Keywords** PTPN11; Tyrosine Phosphatase; Protein-Protein Interaction; Mitochondria; TurboID
**Subject Categories** Methods & Resources; Molecular Biology of Disease; Post-translational Modifications & Proteolysis

## Introduction

Missense mutations are a prevalent type of genetic variation, and they often drive human diseases (Sun et al, 2024; Landrum et al, 2014). Missense mutations can change protein function by perturbing active or regulatory sites, disrupting conformational stability and flexibility (Stefl et al, 2013; Nussinov et al, 2020), changing protein–protein interactions (Simanshu et al, 2017;

Creixell et al, 2015; van Vlimmeren et al, 2024), and altering subcellular localization (Lacoste et al, 2024; McConville et al, 2025). Thus, the comprehensive characterization of a disease-associated missense mutation requires not only an analysis of the biochemical properties and signaling activities of that protein, but also an unbiased assessment of its broader cellular properties, including its localization and interaction network.

SHP2 is a protein tyrosine phosphatase with hundreds of documented disease-associated missense mutations (Grossmann et al, 2010; Landrum et al, 2014). This signaling enzyme has critical roles in diverse cell signaling pathways (Salmond and Alexander, 2006; Lauriol et al, 2015; Asmamaw et al, 2022), most notably as an activator of the Ras/MAPK pathway (Agazie and Hayman, 2003b). SHP2 contains a catalytic protein tyrosine phosphatase (PTP) domain and two phosphotyrosine-recognizing SH2 domains that regulate its localization and activation (Fig. 1A) (Hof et al, 1998). The N-SH2 domain inhibits the PTP domain when not bound to phosphoproteins (Fig. 1B), whereas the C-SH2 domain primarily mediates localization (Marasco et al, 2020). SHP2 dysregulation by missense mutations leads to diverse diseases with a wide range of phenotypes (Tartaglia et al, 2001; Keilhack et al, 2005; Grossmann et al, 2010; Edouard et al, 2010; Nabinger et al, 2013; Furcht et al, 2013; Lauriol et al, 2015). Mutations in SHP2 underlie a large fraction of cases of Noonan syndrome (NS) and Noonan syndrome with multiple lentigines (NSML), two clinically distinct congenital disorders (Roberts et al, 2013; Gelb and Tartaglia, 1993). SHP2 mutations are also associated with leukemias, including juvenile myelomonocytic leukemia (JMML), acute myeloid leukemia (AML), and acute lymphoblastic leukemia (ALL), and are occasionally found in solid tumors (Bentires-Alj et al, 2004; Scheiter et al, 2024).

Most well-characterized pathogenic SHP2 mutations hyperactivate the enzyme by disrupting the auto-inhibitory interface between the N-SH2 and PTP domains (Fig. 1C). These mutations can also alter protein–protein interactions, not only by allowing access the catalytic site, but also by putting the SH2 domains in a binding-competent state and exposing other cryptic binding interfaces (Walter et al, 1999; Paardekooper Overman et al, 2014; Yi et al, 2020; Lin et al, 2021) (Fig. 1D). Many clinically-observed mutations occur outside this interface and do not enhance basal catalytic

[1]Department of Chemistry, Columbia University, New York, NY 10027, USA. [2]Department of Biological Sciences, Columbia University, New York, NY 10027, USA. [3]Koch Institute for Integrative Cancer Research, Massachusetts Institute of Technology, Cambridge, MA 02139, USA. [4]Herbert Irving Comprehensive Cancer Center, Columbia University, New York, NY 10032, USA. ✉E-mail: neel.shah@columbia.edu

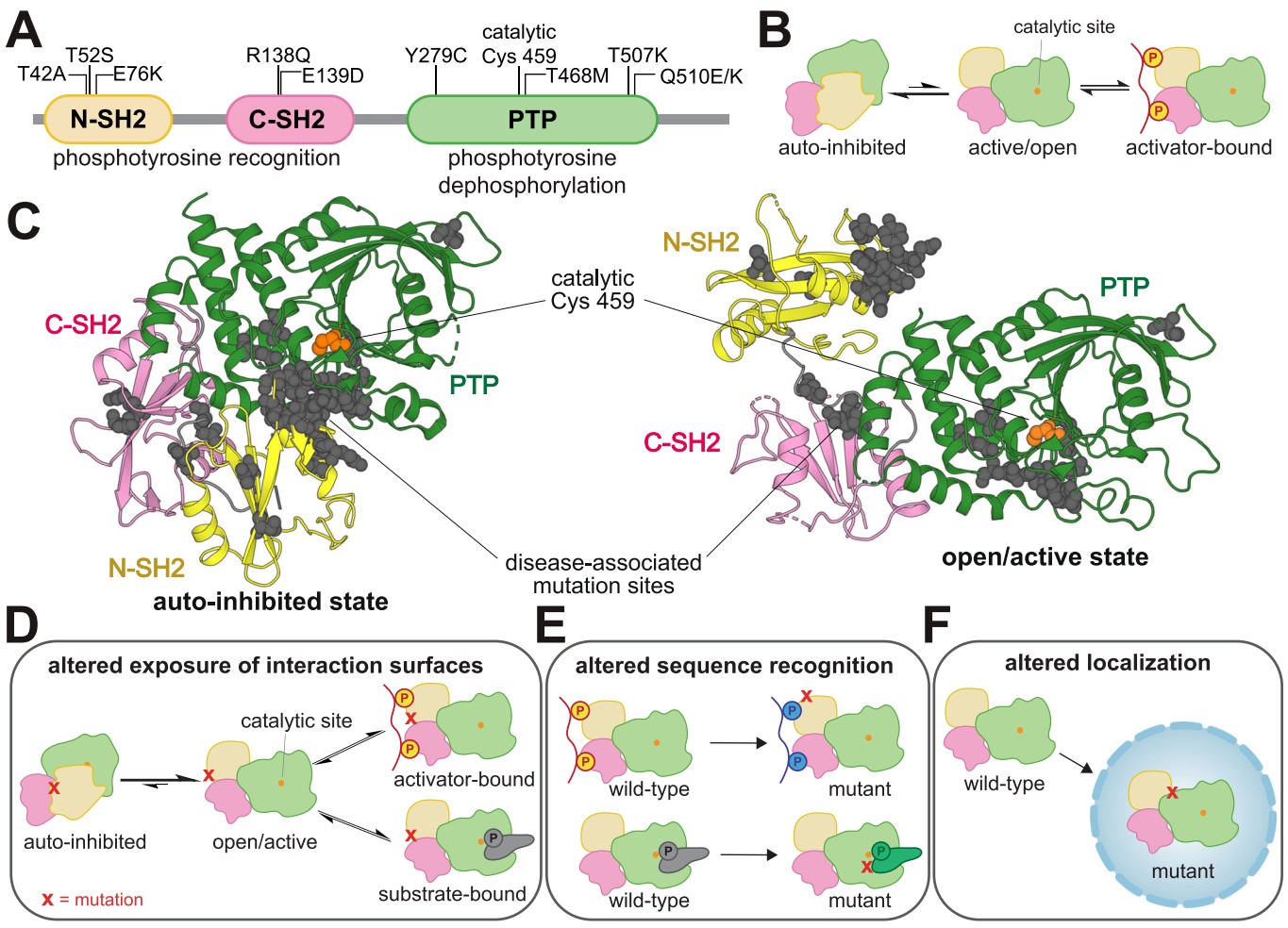

**Figure 1. SHP2 regulation and dysregulation by mutations.**

(A) Domain architecture diagram of SHP2. Relevant mutations and the catalytic cysteine residue are labeled. (B) SHP2 is activated by the SH2 domains binding to phosphoproteins. (C) Visualization of common disease-associated mutations in SHP2 (gray spheres) in the auto-inhibited state (left, PDB code 4DGP) and open state (right, PDB code 6CRF). Cancer mutations cluster at the auto-inhibitory interface between the N-SH2 and PTP domains. Schematic diagrams showing that (D) mutations can alter the exposure of catalytic or non-catalytic interaction surfaces in SHP2, (E) mutations can alter sequence-recognition in either the SH2 domain binding pockets or PTP domain active site, and (F) mutations can alter subcellular localization. Source data are available online for this figure.

activity (Jiang et al, 2025), suggesting alternative disease mechanisms (Fig. 1E,F) (van Vlimmeren et al, 2024; Yu et al, 2014). For example, the T42A mutation alters the N-SH2 ligand-binding pocket, changing binding affinity and specificity for phosphoproteins (van Vlimmeren et al, 2024; Martinelli et al, 2008). Several studies have suggested that mutations in SHP2 can rewire its role in signaling networks by remodeling protein–protein interactions (van Vlimmeren et al, 2024; Zhu et al, 2020; Zhang et al, 2020), underscoring the need for a broad and unbiased functional analysis of SHP2 missense mutations.

Here, we profile mutations-specific changes to the SHP2 interaction network using proximity-labeling proteomics. We report the interactomes of wild-type SHP2 (SHP2^WT) and 10 disease-associated mutants, chosen for their diverse molecular and clinical effects. Our experiments identify mutation-driven changes in SHP2 protein–protein interactions. Our results also pinpoint a role for SHP2 in the mitochondria, as well as enhanced mitochondrial localization and interactions for select mutants.

We find that destabilizing SHP2 mutations increase localization to the mitochondria, potentially affecting mitochondrial homeostasis. We also examine how the clinically relevant allosteric inhibitor TNO155 remodels the SHP2 interactome. Our datasets expand the mechanistic paradigm for mutational effects in SHP2 and will be a resource for understanding SHP2 signaling and pathogenicity. Furthermore, our work demonstrates a robust strategy for identifying mutation-driven effects on protein function using TurboID-proximity labeling.

## Results

### Proximity-labeling proteomics identifies novel wild-type SHP2 interactors

The overarching goal of this study is to examine how mutations alter SHP2 protein–protein interactions; however, we lack a

comprehensive SHP2^WT interactome dataset. Thus, we conducted affinity-purification mass spectrometry of myc-tagged SHP2^WT expressed in SHP2-knock-out HEK 293 cells to identify interacting proteins. Consistent with previous work (St-Denis et al, 2016), this experiment identified few interacting proteins, none of which were known SHP2 interactors (Appendix Fig. S1A; Dataset EV1). Furthermore, SHP2^T42A, a mutant with enhanced N-SH2 binding affinity, did not pull down more phosphoproteins (Appendix Fig. S1B; Dataset EV1). Signaling processes often require transient protein–protein interactions in order to be dynamic and reversible, and this may preclude facile identification of interactors by immunopurification (Westermarck et al, 2013). Indeed, PTP and SH2 domains are known to have weak and transient interactions with phosphoproteins, making both substrate and interactor identification by traditional immunopurification methods difficult (Stanford et al, 2014; Oh et al, 2012). Given these innate features of signaling proteins, coupled with our low-yielding affinity-purification mass spectrometry data, we reasoned that an alternative method was needed to map the SHP2 interactome.

To identify transient SHP2 interactions, co-localized proteins, and mutational differences, we implemented proximity-labeling proteomics using the engineered high-activity biotin ligase TurboID (Fig. 2A) (Branon et al, 2018). Previous studies have used the low-activity biotin ligase BirA to analyze or validate the interactomes of various tyrosine phosphatases (St-Denis et al, 2016; Fearnley et al, 2019; Gong et al, 2021), however, the data for SHP2 are sparse. We fused TurboID to the C-terminus of SHP2, to preserve N-SH2 mediated auto-inhibition. We confirmed that SHP2^WT-TurboID retains catalytic activity, albeit with a fourfold slower rate in vitro than SHP2^WT without TurboID (Fig. EV1A). Importantly, we also confirmed that SHP2^WT-TurboID can be activated by SH2 domain engagement of phosphopeptides (Fig. EV1A), and that the hyperactivating cancer mutation E76K can still exert its effect (Fig. EV1B), demonstrating that the TurboID fusion does not disrupt interdomain regulation. Furthermore, we ensured that SHP2-TurboID retains activity in cells and that this activity can be further enhanced by the E76K mutation (Fig. EV1C–E).

We next expressed SHP2^WT-TurboID and a TurboID-only control in SHP2 knock-out HEK 293 cells to profile the SHP2^WT interactome (Zhu et al, 2022). SHP2-TurboID was expressed at comparable levels to endogenous SHP2 in wild-type HEK 293 cells; however, we observed partial cleavage of SHP2 from TurboID (Fig. EV1F,G). Following established protocols, biotinylated proteins were enriched and identified by mass spectrometry (Cho et al, 2020). We identified 78 proteins with significant enrichment for SHP2^WT over the TurboID-only control, applying a $p$ value cut-off of <0.05 and filtering for proteins with a log$_2$ fold-change of >1 (Fig. 2B, top; Dataset EV2). Among the significantly enriched proteins were known SHP2 interactors, such as components of the Paf1 complex (Paf1, Ctr9, Leo1) and the scaffold protein Gab1 (Fig. 2B, top) (Takahashi et al, 2011; Shi et al, 2000; Schaeper et al, 2000; Cunnick et al, 2000) We also observed enrichment of the mitochondrial peptidyl-prolyl isomerase PPIF, a protein previously shown to co-fractionate with SHP2, giving further confidence to the identified interactome (Havugimana et al, 2012).

In addition to known interactors, our data show that SHP2 interacts or co-localizes with components of several functionally diverse protein complexes and pathways (Fig. EV2A; Dataset EV2). For example, we identified GDI1 and GDI2, which regulate Rab

proteins; and UBL5 and SNRPE, which are components of the spliceosome. Previously, SNRNP27, another spliceosome component, was identified as a putative interactor for SHP2 (Huttlin et al, 2021). We also identified GBF1 and TMED2, which coordinate Golgi trafficking of COPI vesicles; as well as UPP1 and PNP, enzymes involved in pyrimidine and purine metabolism, respectively. Finally, we observed a distinctive signal for many mitochondrial proteins, including components of the electron transport chain, which we discuss extensively below. To confirm that these results are not due to increased protein abundance in SHP2-expressing cells, we also analyzed the total proteome of the matched cell lysates from which our TurboID experiments were performed. Proteins that were significantly enriched in the SHP2^WT-TurboID experiments did not show substantial changes in protein abundance (Fig. EV1H; Dataset EV3).

Next, we stimulated the epidermal growth factor receptor (EGFR), which phosphorylates many downstream proteins that bind to and activate SHP2 (Cunnick et al, 2000). We compared the interactome of SHP2^WT-TurboID to the TurboID-only control in cells stimulated with 100 ng/mL EGF for 10 min and identified 31 proteins that were significantly enriched by SHP2^WT (Figs. 2C, top and EV2B; Dataset EV2). A few of these proteins, including Gab1 and components of the Paf1 complex, were also observed in unstimulated samples. However, most proteins were uniquely enriched in the EGF-stimulated samples. For example, we detected PI3KR2, a regulatory subunit of phosphatidylinositol-3 (PI3) kinase, which is a component of the EGF pathway and an established SHP2 interactor (Brehme et al, 2009). As done for the unstimulated sample, we confirmed that enriched proteins were not the result of increased protein abundance (Fig. EV1I; Dataset EV3). Direct comparison of SHP2^WT-TurboID with and without EGF stimulation revealed 25 proteins with EGF-dependent enhanced proximity-labeling, including several components of the EGF pathway, including the endocytic adapter NUMB, which facilitates EGFR trafficking upon stimulation (Fig. 2D, top) (Abdi et al, 2019). Interestingly, 30 proteins showed reduced biotinylation upon EGF stimulation, including PACSIN2 and SYNJ1, which are involved in EGFR endocytosis in the absence of EGF stimulation (de Kreuk et al, 2012; Kourtidis et al, 2017). We also identified BIRC2, a modulator of mitogenic kinase signaling, and its interactor USP19 (Mei et al, 2011; Tan et al, 2013), as well as checkpoint kinase ATM. Collectively, these experiments reveal the SHP2^WT interactome, its restructuring by EGF stimulation, and several potentially novel functions for SHP2.

## Protein tyrosine phosphorylation partly shapes the SHP2^WT interactome

Since all three globular domains of SHP2 engage tyrosine-phosphorylated proteins, we reasoned that a fraction of the SHP2 interactome would depend on tyrosine phosphorylation, particularly in cells with increased abundance of phosphoproteins due to EGF stimulation. To test this hypothesis, we used the PhosphoSitePlus database to assess what fraction of the proteins enriched in our TurboID dataset have potential tyrosine phosphorylation sites (Hornbeck et al, 2019). Proteins enriched by SHP2^WT-TurboID were more likely to have tyrosine phosphorylation sites than those enriched by the control (Fig. 2B,C, bottom). Furthermore, we found that the EGF-stimulated samples had a slight bias for proteins with

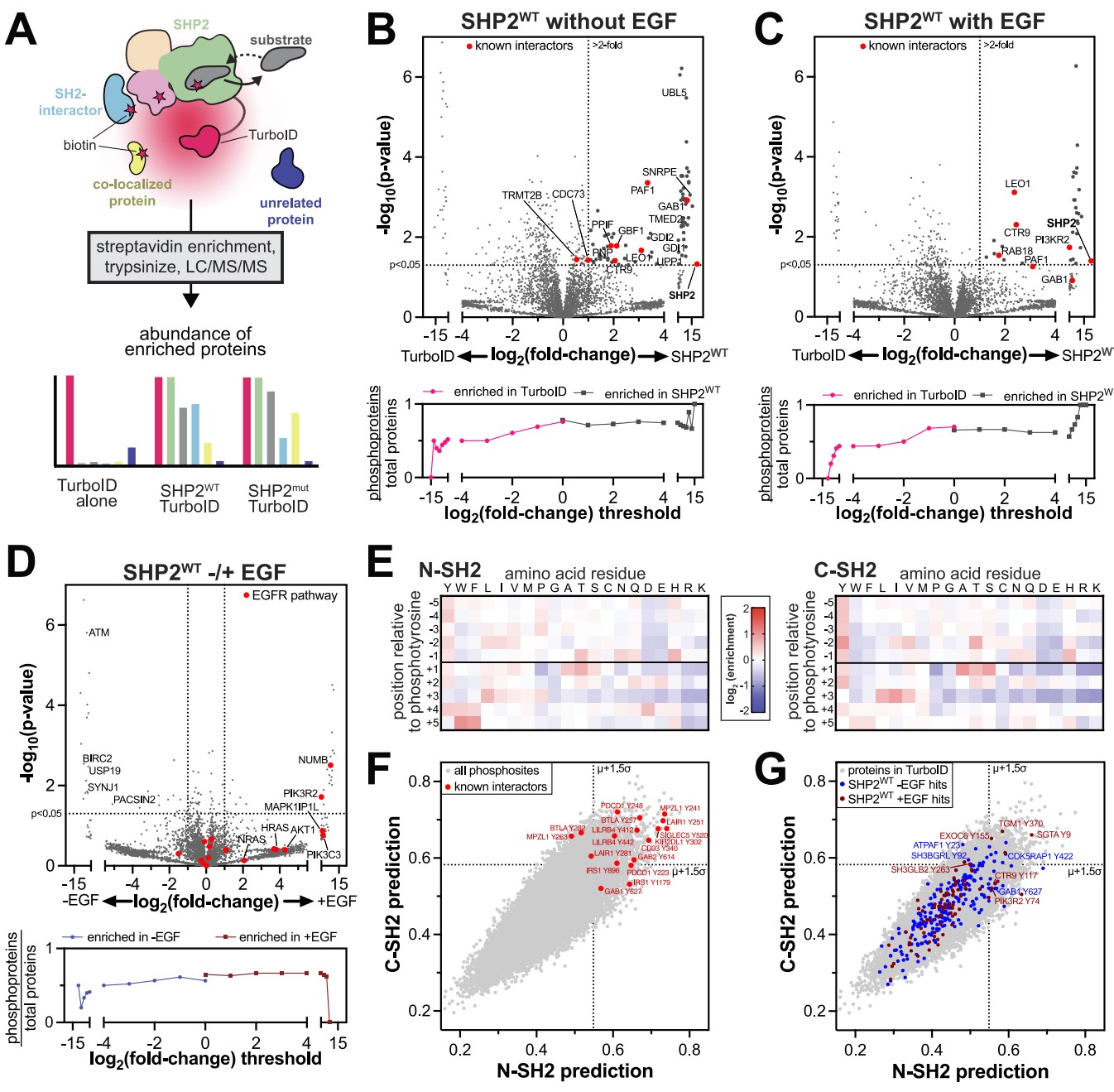

**Figure 2. Profiling the interactomes of wild-type SHP2.**

(A) Schematic diagram of SHP2-TurboID experiments. TurboID generates reactive biotin derivatives, which label substrates, interactors, and co-localized proteins. MS/MS can be used to detect and quantify the interactome. (B) top: Comparison of SHP2^WT-TurboID with a TurboID-only control, in the absence of EGF stimulation. Several known SHP2 interactors were identified, including proteins that are completely absent from the TurboID control. Gaps in the x-axis exclude non-significant points. Points at extreme x-values reflect proteins that were abundant in all SHP2-TurboID replicates and completely absent (imputed) in the TurboID-only control, or vice versa. Bottom: Analysis of the total number of potential phosphoproteins, as identified by PhosphoSitePlus, per total number of observed proteins above each fold-change threshold. (C) Same as (B) but with EGF stimulation. (D) top: Comparison of SHP2^WT-TurboID in unstimulated and EGF-stimulated conditions. Proteins involved in transcription, the spliceosome, and the EGF pathway are indicated. Bottom: Analysis of the total number of potential phosphoproteins, as identified by PhosphoSitePlus, per total number of observed proteins above each fold-change threshold, for SHP2^WT-TurboID with and without EGF stimulation. (E) Heatmaps depicting position-specific weight matrices for peptide recognition by the SHP2 N-SH2 and C-SH2 domains. (F) Predicted scores for binding to the N-SH2 and C-SH2 domains for all phosphotyrosine sites in PhosphoSitePlus (gray dots). Phosphosites on several known interactors are labeled (red dots). (G) Predicted scores for binding to the N-SH2 and C-SH2 domains for all phosphosites on proteins in our proteomics dataset (gray dots). Phosphosites on proteins in the SHP2^WT interactomes are colored in blue (no EGF) and dark red (+EGF). Source data are available online for this figure.

tyrosine phosphorylation sites when compared with unstimulated samples (Fig. 2D, bottom). This analysis demonstrates that SHP2 interactions in our datasets are partly driven by tyrosine phosphorylation.

Next, we used the sequence specificities of the N- and C-SH2 domains to identify potential SH2 domain binding sites in our TurboID datasets. We conducted a high-throughput binding assay with the SHP2 N- and C-SH2 domains, using a degenerate ~10^6 peptide library containing random sequences with a central phosphotyrosine (Fig. 2E; Dataset EV4) (Li et al, 2023). This approach yielded position-specific weight matrices, which we used to score the ~40,000 tyrosine phosphorylation documented in the PhosphoSitePlus database (Dataset EV4) (Hornbeck et al, 2019; Li et al, 2023). We confidently identified several known SHP2 binding sites as high-affinity sequences (Fig. 2F). In the SHP2^WT-TurboID interactome, we identified 26 and 10 predicted tight-binding phosphosites in unstimulated and EGF-stimulated conditions, respectively (Fig. 2G). These include phosphosites on known SHP2 binders, such as Gab1, Ctr9, and PI3KR2, as well as potential novel interactors, such as EXOC6, which has been linked to both EGFR and PI3 kinase signaling (An et al, 2022), and SH3BGRL proteins, which modulate phosphotyrosine signaling through direct interaction with EGFR-family kinases (Li et al, 2020). This juxtaposition of biochemical and proteomic datasets provides additional mechanistic insights into specific SHP2 interactions.

## Pathogenic mutations have divergent effects on the SHP2 interactome

Having established a robust method to map the SHP2^WT interactome, we then examined mutation-dependent changes in protein–protein interactions. We selected ten disease-associated SHP2 mutants, chosen for their distribution across the protein, varying effects on structure and activity, and diverse clinical outcomes (Fig. 3A). Some of these mutants have been thoroughly characterized, whereas others remain more elusive. The Noonan Syndrome-associated T42A mutation enhances N-SH2 domain phosphoprotein binding affinity and specificity with a marginal effect on basal phosphatase activity (van Vlimmeren et al, 2024; Martinelli et al, 2008; Toto et al, 2020). The T52S mutation, identified in JMML patients, does not alter basal activity, ligand-binding affinity, or ligand-binding specificity (van Vlimmeren et al, 2024), and its pathogenic mechanism is unknown. E76K is a well-studied JMML mutation that disrupts auto-inhibition, yielding a constitutively active state of SHP2 (Pádua et al, 2018; LaRochelle et al, 2018). The melanoma mutation R138Q severely attenuates binding of the C-SH2 domain to phosphoproteins and might adopt a slightly more open protein conformation (van Vlimmeren et al, 2024). Curiously, the NS and JMML mutation E139D causes a substantial increase in activity despite not being at the auto-inhibitory interface, and reports of its effect on ligand recognition by the C-SH2 domain have been conflicting (van Vlimmeren et al, 2024; Martinelli et al, 2008). Y279C and T468M, both located in the PTP domain and found in NSML, reduce SHP2 catalytic efficiency while destabilizing the auto-inhibited state (Yu et al, 2013). T507K has been identified in solid tumors and alters substrate specificity (Zhang et al, 2020). Finally, Q510K and Q510E both reduce catalytic activity (Jiang et al, 2025), however Q510E has been identified in NSML and ALL, whereas Q510K has only been

reported in patients with ALL. The interactomes and localization of most of these mutants have not been systematically explored.

We conducted TurboID proximity labeling with each of these mutants, analogous to our SHP2^WT experiments. Interactomes for each mutant were defined by comparison with the TurboID-only control, using the same $p$ value (<0.05) and enrichment cut-off (>2-fold) used for SHP2^WT (Dataset EV2). In the absence of EGF stimulation, the SHP2 mutants shared between 13 and 39 hits with SHP2^WT, which showed enrichment for 78 proteins over the TurboID-control (Fig. EV3A). In the presence of EGF, the interactome of each mutant was generally smaller (Fig. EV3B). One plausible explanation for this is that, in unstimulated cells, SHP2 is more diffusely localized throughout the cell, whereas in EGF-stimulated cells, membrane-proximal tyrosine kinase activity leads to recruitment of SHP2 to specific subcellular locations, as has been shown previously (Paulsen et al, 2012). Furthermore, SHP2 has also been shown to be oxidized upon EGFR stimulation, which may further impact its interaction with signaling partners (Paulsen et al, 2012). To accommodate potential data sparsity, we also calculated a relaxed pairwise overlap, where both $p$ value and fold-change cut-offs were applied to the first variant, but only one of these cut-offs was applied to the second variant (Fig. EV3C,D). This revealed substantial overlap between SHP2 variants (e.g., between 52 and 61 shared proteins for each mutant with SHP2^WT out of the 78 hits without EGF stimulation), providing further confidence in the interactors we identified.

Using the strict combined $p$ value and enrichment cut-offs, we defined a core SHP2 interactome as proteins significantly enriched over the TurboID-only control for at least 4 out of the 11 SHP2 variants in unstimulated cells and 3 out of 11 variants with EGF stimulation, approximately 240 proteins in each case (Fig. 3B). Gene ontology analysis on this core interactome revealed conserved mitochondrial localization and interactions across SHP2 mutants (Fig. 3C,D; Appendix Fig. S5). Both with and without EGF stimulation, these core SHP2 interactomes included mitochondrial ribosome proteins, but components of the respiratory chain complexes (e.g., NDUF and COX proteins) were distinctively enriched in the absence of EGF. By contrast, many cell cycle regulatory proteins were enriched in the EGF-stimulated core SHP2 interactome (e.g., CDK4 and CDC27), as was the EGFR-family kinase Her2 (ERBB2). To further support our analysis, we compared the core interactomes and individual variant inter-actomes with SHP2 interactions compiled in the BioGrid database (Oughtred et al, 2021). Of the 328 unique interactors reported for SHP2 in BioGrid, 130 were observed in our TurboID datasets, and 49 of these proteins were significantly enriched for at least one SHP2 variant over the TurboID-only control (Fig. 3E; Dataset EV5). Of note, six of these previously reported interactors are mitochondrial proteins, corroborating a role for SHP2 in mitochondria (Fig. 3E, marked with an asterisk).

Although many interactors identified in our study are shared across SHP2 variants, individual mutants did display somewhat distinct interactomes from SHP2^WT and from each other (Fig. EV3A–D; Dataset EV2). While some of these differences may arise from data sparsity or noise in our datasets, we hypothesize that many differences reflect mutation-induced changes in protein conformation or molecular recognition, as discussed in subsequent sections. Collectively, the analyses in this section show that pathogenic SHP2 mutants generally preserve some core functions

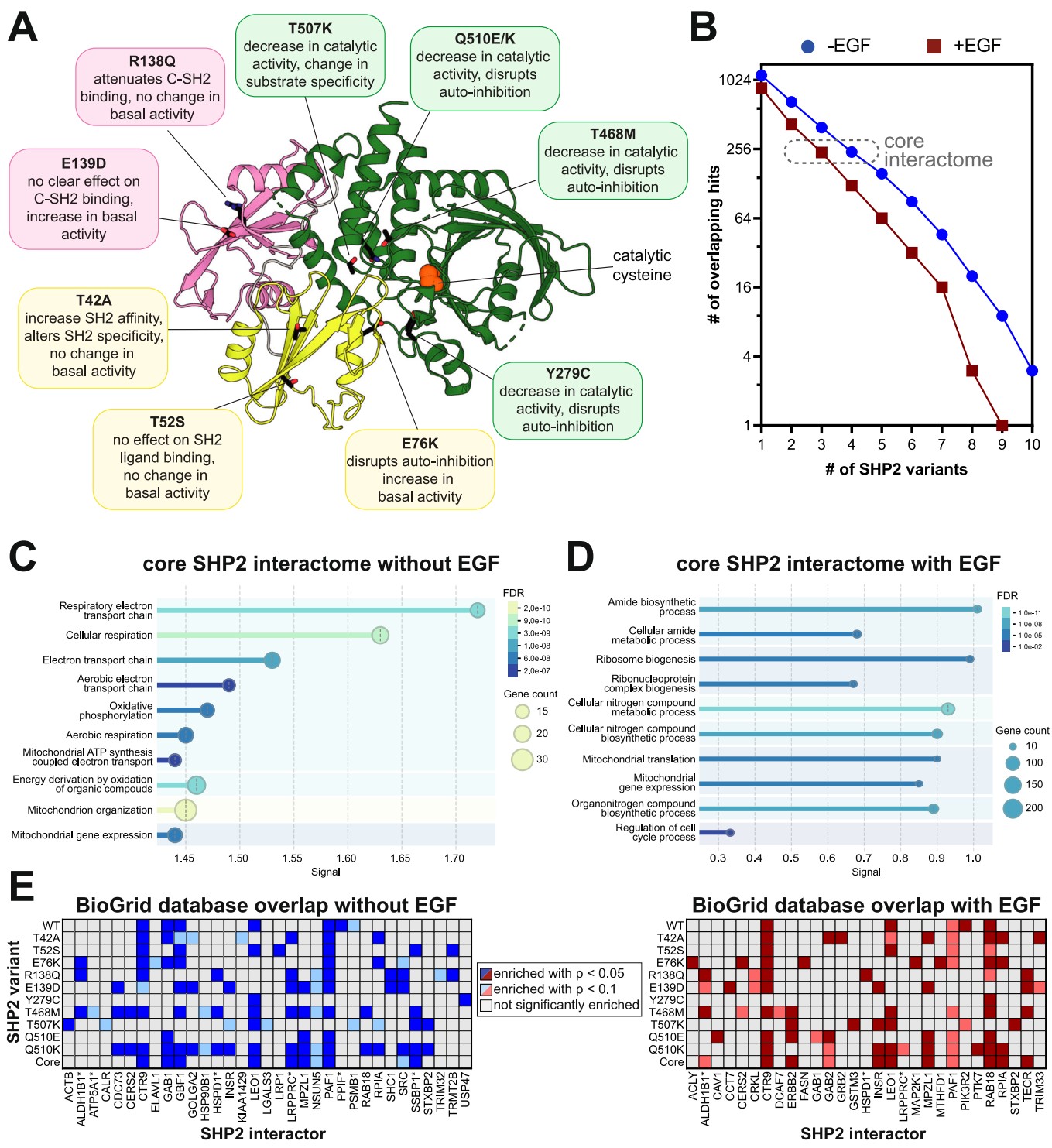

**Figure 3. The interactomes of diverse pathogenic SHP2 variants.**

(A) Overview of mutations in this study. Their position on the auto-inhibited structure (PDB code 4DGP) are indicated, as are known characteristics of each mutant. (B) Number of overlapping proteins that are significantly enriched over the negative control for multiple SHP2 variants, both with and without EGF stimulation. (C) GO enrichment analysis for biological processes of the core SHP2 interactome without EGF stimulation. (D) GO enrichment analysis for biological processes of the core SHP2 interactome with EGF stimulation. (E) Overlap between significantly enriched hits in each TurboID dataset with proteins reported to interact with SHP2 in the BioGrid database. Interactor names marked with an asterisk correspond to mitochondrial proteins. $P$ values were determined using a heteroscedastic, two-tailed $t$-test comparing SHP2 samples to TurboID-only control samples. Source data are available online for this figure.

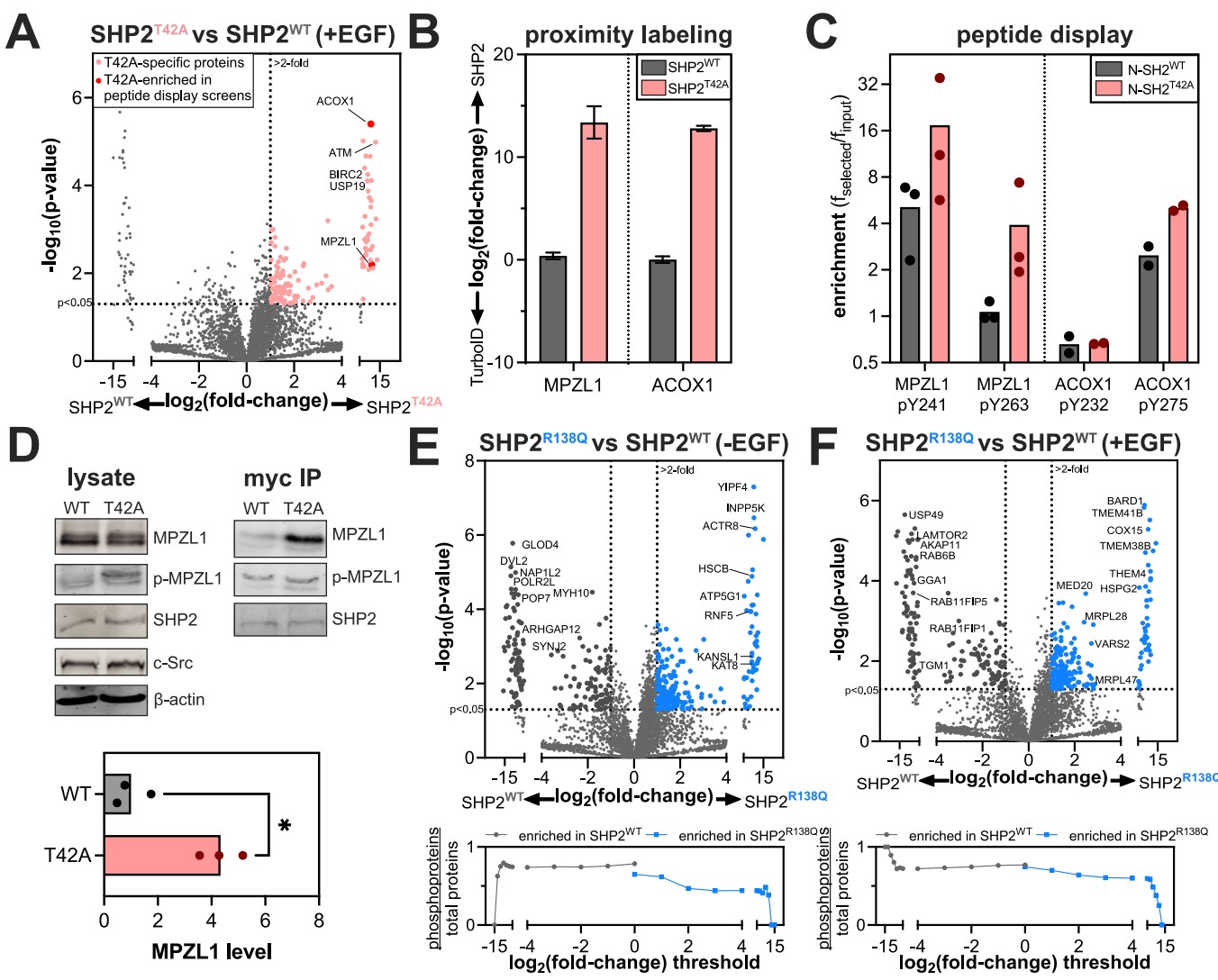

**Figure 4. Rewired interactomes in SHP2 mutants with altered phosphoprotein binding properties.**

(A) Volcano plot showing enriched proteins in SHP2$^{T42A}$, compared to SHP2$^{WT}$. (B) Log$_2$ fold-change in enrichment for MPZL1 and ACOX1 by SHP2$^{WT}$ and SHP2$^{T42A}$ relative to the TurboID control. Bar height indicates the ratio of the mean value for SHP2 samples relative to the mean value for control samples, and error bars represent standard deviation after error propagation. P values were determined using a heteroscedastic, two-tailed t-test between SHP2$^{WT}$ and SHP2$^{T42A}$. For MPZL1, $p = 0.0064$. For ACOX1, $p = 0.000004$. (C) Enrichment of MPZL1- and ACOX1-derived phosphopeptides from peptide display screens using the SHP2 N-SH2$^{WT}$ and N-SH2$^{T42A}$ domains. (D) Phosphorylation levels and co-immunopurification of MPZL1 by SHP2$^{WT}$ and SHP2$^{T42A}$. Quantified MPZL1 levels in the myc-SHP2 co-IP are shown in the graph. All values are normalized to a SHP2 wild-type mean of 1. The p value ($p = 0.0264$) was determined using a paired, two-tailed t-test. (E) (top) Volcano plots highlighting enriched proteins in SHP2$^{R138Q}$, compared to SHP2$^{WT}$ in the absence of EGF stimulation ($n = 3$ biological replicates). P values were determined using a heteroscedastic, two-tailed t-test. (bottom) Analysis of the total number of potential phosphoproteins, as identified by PhosphoSitePlus, per total number of observed proteins past each fold-change threshold for SHP2$^{WT}$ versus SHP2$^{R138Q}$. The SHP2$^{R138Q}$ interactome is less enriched in potentially tyrosine-phosphorylated proteins than the SHP2$^{WT}$ interactome. No negative selection against tyrosine phosphorylation sites is observed for SHP2$^{WT}$. (F) Same as (E), but with EGF stimulation ($n = 3$ biological replicates). Source data are available online for this figure.

of wild-type SHP2, but specific mutations partly remodel the SHP2 interactome, and this can be further altered by EGF stimulation.

## Mutation-dependent changes in phosphoprotein recognition underlie altered SHP2 interactomes

Since many SHP2 interactions are driven by phosphoprotein binding, we first focused on the T42A and R138Q mutations in the N- and C-SH2 domains, which are known to impact

phosphoprotein recognition (Fig. 3A). We and others previously showed that the T42A mutation enhances the affinity and alters specificity of the N-SH2 domain for phosphoproteins (van Vlimmeren et al, 2024; Martinelli et al, 2008). Consistent with this, we observed an overall increase in the number of proteins enriched by SHP2$^{T42A}$ over SHP2$^{WT}$ in our proximity-labeling datasets (Figs. 4A and EV3A–D). Among these SHP2$^{T42A}$-enriched proteins were MPZL1 and ACOX1 (Fig. 4B). In our previous work using phosphopeptide libraries to profile SHP2 SH2 mutants, we

found that phosphopeptides derived from these proteins showed enhanced binding to N-SH2$^{T42A}$ with respect to N-SH2$^{WT}$, providing a mechanistic explanation for the enrichment of these proteins in our SHP2$^{T42A}$ interactome (Fig. 4C) (van Vlimmeren et al, 2024).

MPZL1 is a known SHP2 interactor, and enhancement of this interaction through SHP2 NSML mutations is linked to hypertrophic cardiomyopathy (Paardekooper Overman et al, 2014; Yi et al, 2020). Unlike SHP2$^{T42A}$, which is a Noonan Syndrome mutant, NSML mutants like SHP2$^{T468M}$ and SHP2$^{Q510E}$ have a higher propensity to adopt the open conformation, allowing them to more readily bind phosphoproteins, and these mutants generally also have impaired catalytic activity (Yu et al, 2014). In NSML, MPZL1 is hyperphosphorylated on Y241 and Y263, because open conformation SHP2 mutants are more competent for MPZL1 binding and protect these sites from dephosphorylation (Vemulapalli et al, 2021). Indeed, like SHP2$^{T42A}$, the NSML SHP2$^{T468M}$ and SHP2$^{Q510E}$ mutants showed increased interaction with MPZL1 in our TurboID data relative to SHP2$^{WT}$ (Fig. EV3E). We validated the enrichment of MPZL1 by SHP2$^{T42A}$ by co-immunopurification, which showed that the T42A mutation both enhances MPZL1 binding and shows increased MPZL1 phosphorylation (Fig. 4D). This demonstrates that SHP2$^{T42A}$, despite not adopting a more open conformation than SHP2$^{WT}$, converges on increased interaction with MPZL1 through altered sequence-recognition in the N-SH2 domain (van Vlimmeren et al, 2024).

We previously showed that the R138Q mutation severely weakens phosphotyrosine binding to the C-SH2 domain by removing an Arg residue in the conserved FLVR motif that is required for SH2-phosphotyrosine interactions (van Vlimmeren et al, 2024). Consistent with this, we see reduced labeling of interactors with known or predicted high-affinity C-SH2 binding sites (e.g., Gab1, Dataset EV4) by SHP2$^{R138Q}$ relative to SHP2$^{WT}$ and other variants (Fig. EV3F; Dataset EV2). Surprisingly, however, we observed many proteins enriched by SHP2$^{R138Q}$ over SHP2$^{WT}$, suggestive of the mutation enhancing interactions or co-localization with some proteins (Figs. 4E,F, top and EV3A–D). Since this mutation disrupts phosphoprotein interactions, we suspected that proximity-labeling with SHP2$^{R138Q}$ would be less dependent on tyrosine-phosphorylated proteins than with SHP2$^{WT}$. Indeed, proteins enriched by SHP2$^{R138Q}$ are less likely to be phosphoproteins than those enriched by SHP2$^{WT}$ (Fig. 4E,F, bottom) (Hornbeck et al, 2019). Since the C-SH2 domain of SHP2 is critical for localization, one plausible explanation for the increased proximity-labeling with SHP2$^{R138Q}$ is that the loss of C-SH2 function leads to SHP2 mislocalization and promiscuous protein–protein interactions. Overall, our analyses of SHP2$^{T42A}$ and SHP2$^{R138Q}$ reveal how changes in phosphoprotein recognition can measurably alter the interactome of a signaling protein.

## Select pathogenic mutations enhance SHP2 mitochondrial localization and interactions

As noted above, mitochondrial proteins constitute a large part of the core SHP2 interactome (Fig. 3C,D; Appendix Fig. S2). SHP2$^{WT}$ showed enrichment of 21 mitochondrial proteins over the TurboID-only control in unstimulated cells (Fig. 5A), consistent with previous work demonstrating that SHP2 can localize to mitochondria (Salvi et al, 2004; Arachiche et al, 2008; Zang et al, 2012; Xu et al, 2013; Guo et al, 2017; Morgenstern et al, 2021; Olou

et al, 2023; Kan et al, 2024). Levels of biotin and ATP, the substrates of TurboID, are higher in mitochondria than in the cytoplasm or other organelles, which could bias the subcellular TurboID signal (Petrelli et al, 1978; Baldet et al, 1992; Said et al, 1992). To examine this potential bias, we assessed how the addition of external biotin impacted signal intensity in our TurboID datasets for the whole proteome and for the sub-proteome that is annotated to have mitochondrial localization (Appendix Fig. S3A). In the absence of exogenous biotin, the mitochondrial proteome had higher median TurboID labeling than the whole cellular proteome, both for the TurboID-only control and for SHP2$^{WT}$-TurboID, irrespective of EGF stimulation. This suggests that endogenous biotin levels in mitochondria may artificially elevate the TurboID signal in this organelle. However, upon addition of exogenous biotin, the mitochondrial bias disappeared for the TurboID-only controls and was minimal for SHP2$^{WT}$-TurboID, suggesting that any elevated signal was SHP2-specific (Appendix Fig. S3A). Indeed, we observed a distinctive enhancement in mitochondrial protein labeling for SHP2$^{WT}$-TurboID over the TurboID-only control only in the presence of exogenous biotin, in the absence of EGF stimulation (Appendix Fig. S3B).

We further examined SHP2 mitochondrial localization using confocal fluorescence microscopy. First, we compared wild-type HEK 293 cells to the SHP2 knock-out cells used for proteomics, both without transfection of any SHP2 construct (Fig. EV4A). In wild-type HEK 293 cells, we observed robust staining of endogenous SHP2 in the cytoplasm and modest nuclear staining, using a SHP2-specific antibody. This signal was completely absent in the SHP2 knock-out cells, confirming the specificity of the antibody. Mitochondria were selectively stained using a Tom20-specific antibody (Fig. EV4A). A large fraction of the Tom20-positive pixels overlapped with SHP2-positive pixels, suggesting SHP2 co-localization with mitochondria (Fig. EV4B), however, we note that this analysis cannot confirm whether SHP2 is outside or inside mitochondria. Upon stimulation of untransfected wild-type HEK 293 cells with EGF, we observed a reduction in the fraction of mitochondrial pixels that overlapped with endogenous SHP2, but this change may be due to a reduction in SHP2 intensity in these cells (Fig. EV4C,D). Finally, we imaged SHP2 knock-out cells transfected with SHP2$^{WT}$ or SHP2$^{WT}$-TurboID. Without TurboID fusion, SHP2 was predominantly cytoplasmic with some nuclear localization, but SHP2$^{WT}$-TurboID was largely excluded from the nucleus because our TurboID construct has a nuclear exit sequence (Fig. EV4E). In the SHP2-transfected cells, most Tom20-positive pixels overlapped with SHP2 pixels, again suggesting SHP2-mitochondria co-localization. Given the higher abundance of SHP2 in these cells, differences between unstimulated and EGF-stimulated cells could not be unambiguously discerned by microscopy (Fig. EV4B,C).

In our TurboID proteomics datasets, we noticed a distinctive increase in mitochondrial signal for R138Q, T468M, Q510K, and to a lesser extent, E76K (Fig. 5B; Appendix Fig. S4). To validate this, we expressed these mutants without TurboID fusion in SHP2 knock-out HEK 293 cells, isolated mitochondria, blotted for SHP2, and confirmed that some SHP2 mutants have more mitochondrial localization than SHP2$^{WT}$ (Fig. 5C). SHP2-associated Noonan Syndrome and cardiomyopathies have symptoms that resemble those in primary mitochondrial disorders (Lee et al, 2010), and disease-associated activating SHP2 mutants have been shown to

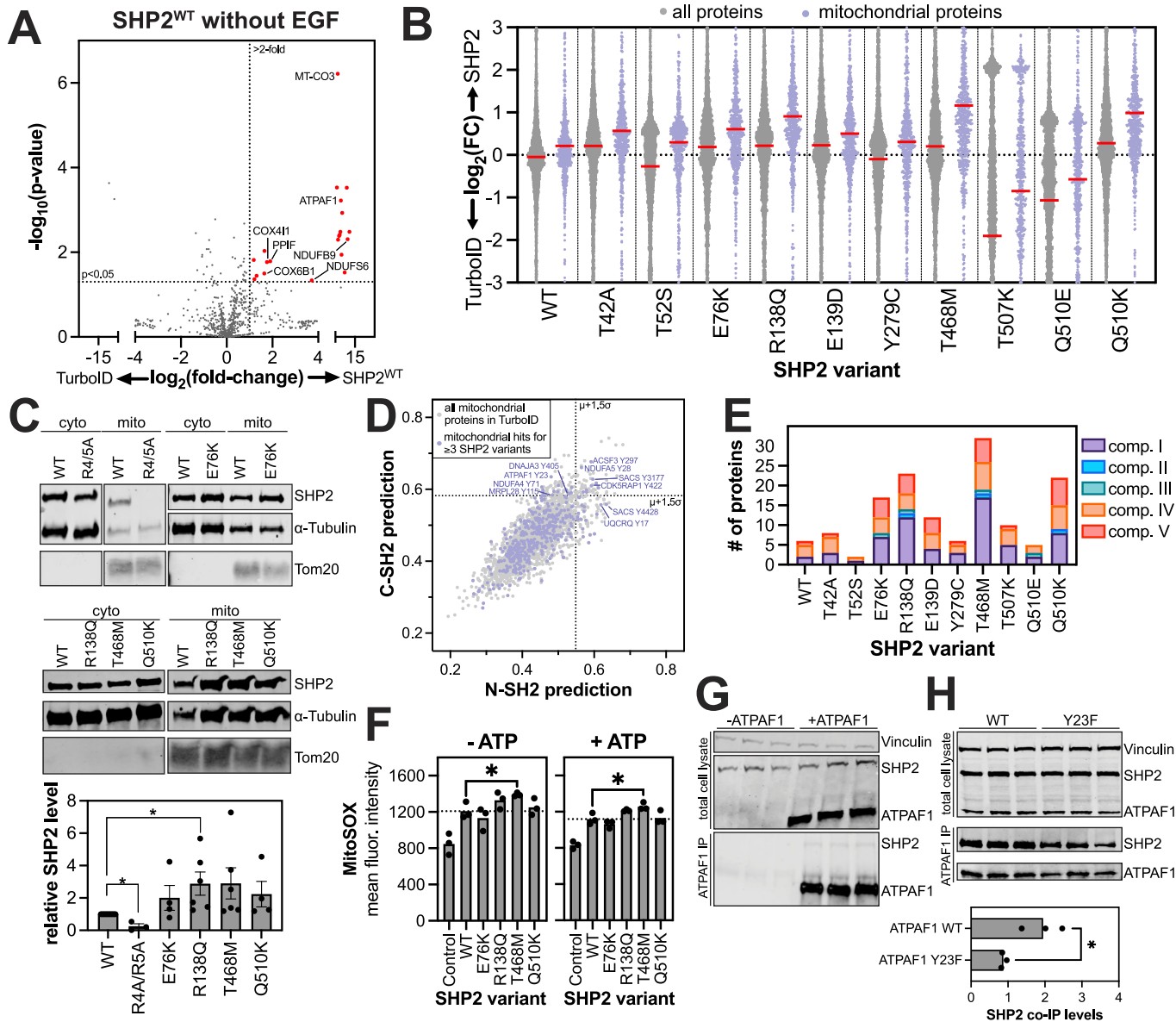

**Figure 5. SHP2 interactions in mitochondria.**

(**A**) Volcano plot of the mitochondrial proteins in our TurboID dataset, showing many mitochondrial proteins that are significantly enriched by SHP2$^{WT}$-TurboID compared to the TurboID-only control, in the absence of EGF stimulation (red dots) ($n = 3$ biological replicates). $P$ values were determined using a heteroscedastic, two-tailed $t$-test. (**B**) Enhanced proximity labeling of mitochondrial proteins by SHP2 variants relative to the TurboID-only control (lilac), compared to proximity labeling of the whole proteome (gray). Red line denotes the median ($n = 3$ biological replicates). (**C**) Cellular fractionation and western blots confirming enhanced mitochondrial localization of SHP2$^{E76K}$, SHP2$^{R138Q}$, SHP2$^{T468M}$, and SHP2$^{Q510K}$, relative to SHP2$^{WT}$. SHP2$^{R4/5A}$ (a R4A + R5A double mutant), which has a defective mitochondrial localization signal, was used as a control and shows reduced mitochondrial abundance. Error bars denote standard deviation ($n = 3$–6 biological replicates). $P$ values were determined using paired, two-tailed $t$-tests. For SHP2$^{WT}$ vs SHP2$^{R4A/R5A}$, $p = 0.0385$; for SHP2$^{WT}$ vs SHP2$^{R138Q}$, $p = 0.0471$. (**D**) N-SH2 and C-SH2 predictions for human phosphosites. Select high-scoring mitochondrial phosphosites are labeled. (**E**) The number of electron transport chain proteins was significantly enriched over TurboID for each SHP2 variant in the absence of EGF stimulation. (**F**) MitoSOX staining showing increased reactive oxygen species for SHP2$^{T468M}$. $P$ values were determined using a two-tailed, unpaired $t$-test. For SHP2$^{T468M}$ vs SHP2$^{WT}$, $p = 0.0273$ without ATP and $p = 0.0318$ with ATP. (**G**) Co-immunopurification of SHP2$^{WT}$ by ATPAF1, co-expressed in HEK 293 cells. Three independent replicates from three independent transfections are shown on the same blots. (**H**) Impact of ATPAF1 Y23F mutation on SHP2 co-immunopurification. Three independent replicates from three independent transfections are shown on the same blots. Quantification of SHP2 band intensity, normalized to total cell lysate levels, shows a significant reduction in SHP2 co-immunopurification upon Y23F mutation. The $p$ value ($p = 0.0143$) was determined using a heteroscedastic, unpaired $t$-test. Source data are available online for this figure.

impact mitochondrial function and homeostasis (Kan et al, 2024; Karampitsakos et al, 2023). Thus, the increased mitochondrial localization for some mutants could contribute to pathogenicity, warranting a deeper analysis of the specific interactions and functions of SHP2 in this organelle.

## SHP2 interacts with proteins that regulate core mitochondrial processes

Given the significant mitochondrial signal in our datasets, and the mutant-specific enhancement in this signal, we next sought to identify specific mitochondrial protein complexes and processes that SHP2 might modulate. To achieve this, we juxtaposed our TurboID data with mitochondrial entries for SHP2 in the BioGrid database (Fig. 3E; Dataset EV5), we examined reports of SHP2 mitochondrial function, and we utilized our SHP2 SH2 binding predictions (Fig. 5D; Dataset EV4). From these analyses, we found that SHP2 likely plays a significant role in regulating mitochondrial respiration. SHP2 has been reported to interact with proteins in complex I and III of the electron transport chain, and SHP2 mutations can modulate mitochondrial respiration (Kan et al, 2024; Zheng et al, 2013). Consistent with this, our data show that many SHP2 variants proximity-label electron transport chain proteins (Fig. 5E), and some of these potential interactors also have high-scoring SH2-binding phosphosites (e.g., NDUFA5, UQCRQ, and NDUFA4, components of complex I, III, and IV of the electron transport chain, respectively) (Fig. 5D). Previously, SHP2 has also been shown to directly bind to NDUFB8 (Kan et al, 2024), which did not pass the significance threshold in our TurboID experiments but has a high-scoring phosphosite for both domains. Interestingly, SHP2$^{WT}$ primarily labeled components of complex I and complex IV, while some mutants showed different labeling patterns. For example, whereas SHP2$^{WT}$ only labeled accessory subunits of complex I, SHP2$^{T468M}$ also labeled assembly- and core-subunits, suggestive of a longer residency near complex I. Notably, we saw no labeling of complex III by SHP2$^{WT}$, and this was recapitulated in most SHP2 mutants.

Several mutants showed an increase in labeling of complex V (ATP synthetase) proteins (Fig. 5E). ATP5A1 is a reported interactor of SHP2 from co-fractionation proteomics (Havugimana et al, 2022), although this protein is only enriched with SHP2$^{T468M}$ (Fig. 3E). However, several other components of complex V (ATP5F1, ATP5L, ATP5G1, ATP5H, ATP5I, ATP5J2, and ATP5O) are significantly enriched with multiple SHP2 variants, with ATP5F1 enriched by six variants (Dataset EV2). This suggests that SHP2 mutants could potentially alter core mitochondrial functions, such as oxidative phosphorylation and ATP production. Indeed, we found that a few SHP2 mutants, particularly SHP2$^{T468M}$, enhanced formation of mitochondrial reactive oxygen species, indicating mutant-specific effects on respiration (Fig. 5F). We also identified ATPAF1, an assembly factor of the F(1) component of ATP synthase, as an enriched protein for several SHP2 variants, and we found a high-scoring SH2-binding phosphosite (Y23) on this protein (Fig. 5D). We confirmed that SHP2 could interact with ATPAF1 by co-expression and co-immunopurification (Fig. 5G). This interaction was reduced but not abolished by the Y23F mutation, both supporting the importance of this site and suggesting other interaction interfaces (Fig. 5H).

As noted earlier, another intriguing mitochondrial hit, previously observed in co-fractionation experiments (Havugimana et al, 2012), was the mitochondrial peptidyl-prolyl isomerase PPIF, also known as cyclophilin D. Like ATPAF1, PPIF regulates assembly of the ATP synthase machinery (Giorgio et al, 2009; Beutner et al, 2017). It also plays a crucial role in the functioning of the mitochondrial permeability transition pore, and it interacts with a key regulator of the mitochondrial permeability transition, ANT1 (Clarke et al, 2002). Notably, ANT1 is a known substrate of SHP2, further supporting the co-localization of SHP2 with key regulators of mitochondrial function (Guo et al, 2017). We were intrigued by the SHP2/PPIF interaction as SHP2 is not known to be regulated by proline isomerization. AlphaFold 3 models of this interaction suggest an interface involving the active site of PPIF and a region of SHP2 in the C-terminal tail that does not contain a proline residue (Appendix Fig. S5) (Abramson et al, 2024). This mode of interaction strongly resembles that seen for high-affinity inhibitors of PPIF (Peterson et al, 2022), suggesting that PPIF could be acting as a scaffold or SHP2, or SHP2 may be inhibiting PPIF, both of which warrant future investigation.

Finally, we observed significant proximity labeling of other proteins that regulate core mitochondrial processes. For example, SSBP1 (single-stranded DNA binding protein 1), which plays a role in mitochondrial biogenesis, was a statistically significant TurboID hit for several of our SHP2 variants, and it is a reported interactor of SHP2 in BioGrid (Fig. 3E). Notably, SSBP1 is reported to be phosphorylated at Y73 by mitochondrially-located Src kinase, and this phosphorylation event regulates SSBP1 function (Djeungoue-Petga et al, 2019). This phosphosite is not a predicted high-affinity binder for the SHP2 SH2 domains (Dataset EV4), but it is plausible that SHP2 dephosphorylates this site to regulate SSBP1. We also observed significant labeling of many components of the mitochondrial ribosome (Dataset EV2; Appendix Fig. S2). One component, MRPL42, while not observed in our datasets, is a reported SHP2 interactor in BioGrid (Dataset EV5). MRPL28, by contrast, is a TurboID hit for several SHP2 variants and has a predicted high-affinity C-SH2 binding site (Fig. 5D; Dataset EV4). Finally, several mitochondrial import and chaperone proteins were also significant hits across several SHP2 mutants, which we discuss in the next section. Collectively, our interactomics data expand the scope of SHP2 interactors in mitochondria and suggest plausible roles in this subcellular compartment.

## Destabilization of the SHP2 protein structure might contribute to mitochondrial import

The mechanism by which some SHP2 mutants increasingly localize to mitochondria is unclear. The Tom40 complex has been shown to mediate SHP2 translocation (Guo et al, 2017). SHP2$^{WT}$ and most mutants proximity-labeled Tom40 more than the TurboID-only control, but we did not observe mutant-specific enhancement of Tom40 engagement (Fig. 6A). Tom20, a component of the Tom40 complex, binds mitochondrial localization sequences on cytoplasmic proteins, often at their N-termini (Abe et al, 2000). SHP2 has a mitochondrial localization motif near its N-terminus (R4 to H8) (Guo et al, 2017), and mutation of Arg4 and Arg5 to alanine residues diminishes mitochondrial localization (Fig. 5C) (Guo et al, 2017). Furthermore, in a recent Tom20 proximity-labeling study,

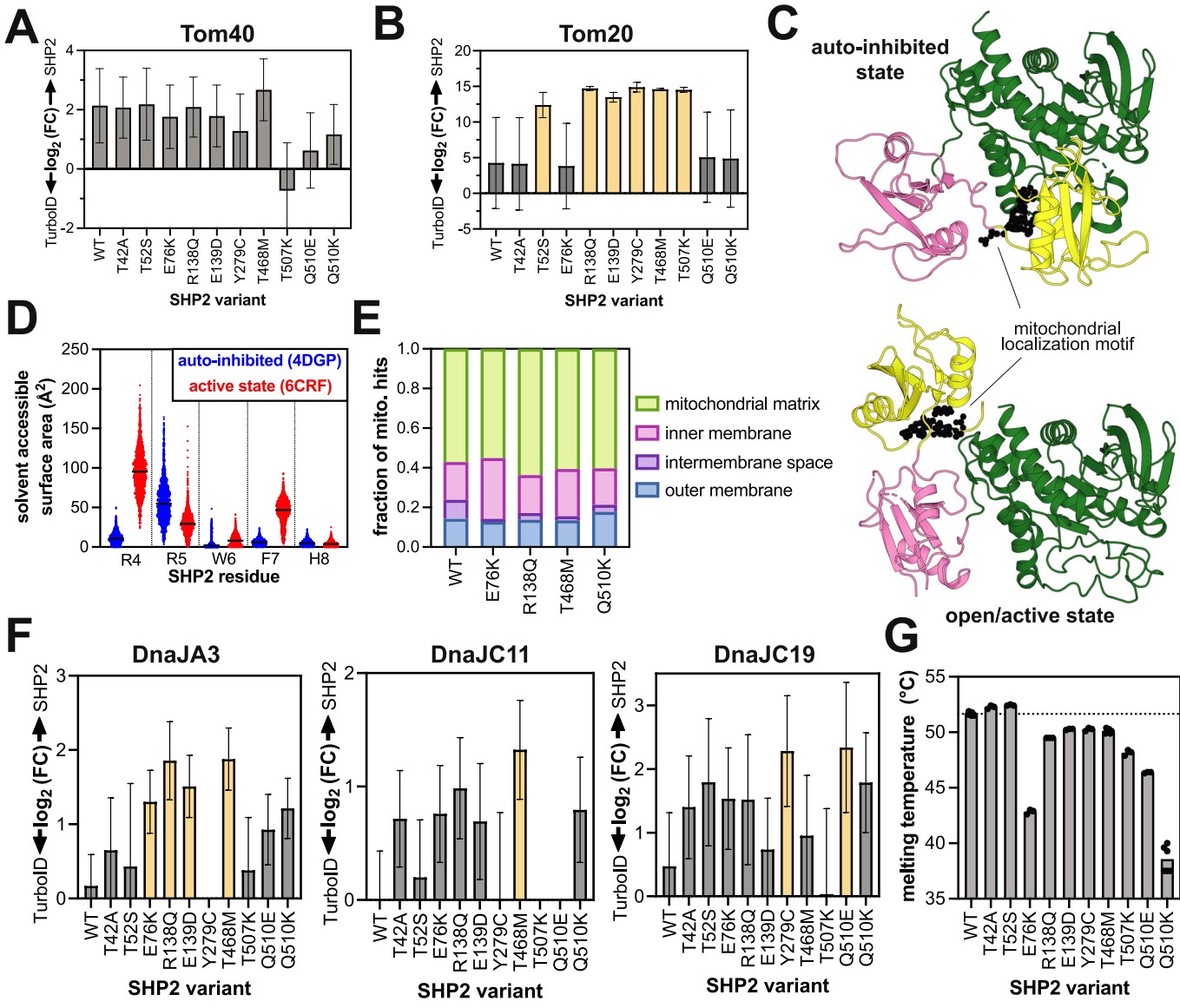

**Figure 6. Protein interactions and structural features impacting SHP2 mitochondrial import.**

Proximity labeling of (**A**) Tom40 and (**B**) Tom20 outer mitochondrial import machinery ($n = 3$ biological replicates). Variants in yellow are significantly enriched over the TurboID control (fold-change >2, $p$ value <0.05): $p = 0.0099$ (T52S), $p = 2.9522 \times 10^{-5}$ (R138Q), $p = 0.0010$ (E139D), $p = 0.0273$ (Y279C), $p = 9.9211 \times 10^{-9}$ (T468M), and $p = 0.0001$ (T507K). Bar height indicates the ratio of the mean value for SHP2 samples relative to the mean value for control samples, and error bars represent standard deviation after error propagation. (**C**) The SHP2 mitochondrial localization motif (R4 to H8) highlighted as black spheres on auto-inhibited (left, PDB code 4DGP) and open (right, PDB code 6CRF) structures. (**D**) Differences in solvent accessibility of the RRWFH motif in molecular dynamics simulations using auto-inhibited conformation (starting structure PDB code 4DGP) or open conformation (starting structure PDB code 6CRF). (**E**) Submitochondrial localization of SHP2 variants inferred from proximity-labeling patterns. (**F**) Proximity-labeling signal for mitochondrial Hsp40/DnaJ proteins, DnaJA3, DnaJC11, and DnaJC19 ($n = 3$ biological replicates). Variants in yellow are significantly enriched over the TurboID control (fold-change >2, $p$ value <0.05): For DNAJA3, $p = 0.0354$ (E76K), $p = 0.0082$ (R138Q), $p = 0.0275$ (E139D), and $p = 0.0169$ (T468M). For DNAJC11, $p = 0.0222$ (T468M). For DNAJC19, $p = 0.0434$ (Y279C) and $p = 0.0328$ (Q510E). Bar height indicates the ratio of the mean value for SHP2 samples relative to the mean value for control samples, and error bars represent standard deviation after error propagation. (**G**) Melting temperatures of SHP2 variants measured by differential scanning fluorimetry. Source data are available online for this figure.

SHP2 was identified as a proximal protein (preprint: Akram et al, 2024). Notably, in our datasets, Tom20 is an enriched protein for several SHP2 mutants but not SHP2[WT], as are other components of the Tom20/Tom40 complex (Figs. 6B and EV5A). We hypothesized that some SHP2 mutations might enhance exposure of the mitochondrial localization signal due to changes in protein conformation (Fig. 6C). We analyzed our previously reported molecular dynamics simulations of SHP2 in both the auto-inhibited state and an open state that is likely accessed by some disease mutants. These simulations showed an increase in solvent accessibility for R4 and F7 in the open conformation, but not for R5 (Fig. 6D) (Jiang et al, 2025). Based on this analysis, it is plausible

but not conclusive that SHP2 mutations alter Tom20 access to the mitochondrial localization sequence.

SHP2 has been observed both in the intermembrane space and the mitochondrial matrix, but most reported mitochondrial functions involve matrix proteins (Salvi et al, 2004; Zheng et al, 2013; Guo et al, 2017; Olou et al, 2023; Kan et al, 2024). Indeed, of the mitochondrial proteins labeled by SHP2 in our TurboID experiments, most were matrix and inner membrane proteins, although SHP2$^{WT}$ also disproportionately labeled intermembrane space proteins more than the mutants (Fig. 6E). This suggests that SHP2$^{WT}$ accumulates in the intermembrane space, or that some mutants experience more efficient transport through the inner mitochondrial membrane. Transport of soluble proteins through the inner membrane is typically mediated by the Tim23 complex. While components of this complex are present in our datasets, most do not show a strong signal, other than Tim17B (Fig. EV5B).

Mitochondrial transport and protein maturation in the mitochondrial matrix are highly dependent on chaperone proteins, most notably the chaperonin Hsp60 and the Hsp70/40 system (Cheng et al, 1989; Tang et al, 2022; Pfanner et al, 2019). Hsp60 (HSPD1) has previously been shown to interact with SHP2 via immunopurification and co-fractionation mass spectrometry (Shen et al, 2009; Havugimana et al, 2022), and we observe significant proximity labeling of this mitochondrial chaperone by multiple SHP2 mutants (Fig. EV5C; Dataset EV5). SHP2 is also a known Hsp70 client (Yoo and Hayman, 2006), however, neither cytoplasmic Hsp70 (HSPA1A/HSPA1B), nor its mitochondrial counterpart mortalin (HSPA9) were significantly enriched for any SHP2 variant in our datasets (Fig. EV5C). Nonetheless, multiple Hsp40/DnaJ proteins showed enhanced proximity-labeling signal for SHP2 mutants, most notably DnaJA3, DnaJC11, and DnaJC19, all of which play critical roles in mitochondria (Fig. 6F) (Qiu et al, 2006). DnaJC19 coordinates with Tim23 to facilitate protein translocation through the inner mitochondrial membrane (Waingankar and D'Silva, 2021), DnaJA3 is critical for mitochondrial matrix proteostasis (Wang et al, 2020), and DnaJC11 is involved in cristae formation and protein import (Violitzi et al, 2019). It is noteworthy that, like SHP2, these proteins have been connected to significant developmental disorders and cardiomyopathies (Hayashi et al, 2006; Ojala et al, 2012). Our observations suggest that interactions with Hsp40 proteins could modulate the mitochondrial activity of SHP2 mutants.

SHP2 likely interacts with Hsp40/DnaJ proteins to facilitate its unfolding and refolding as it traverses the mitochondrial membranes, and these interactions presumably depend on the stability of SHP2. The auto-inhibited state of SHP2 has extensive interdomain interactions, likely making it more thermally stable than the SHP2 open state. Conversely, SHP2 mutants that adopt a more open conformation tend to have lower melting temperatures by differential scanning fluorimetry (van Vlimmeren et al, 2024; Jiang et al, 2025; Serbina & Bishop, 2023; Kim et al, 2024). We previously showed that SHP2$^{R138Q}$ and SHP2$^{T468M}$ have lower melting temperatures than SHP2$^{WT}$ and show here that SHP2$^{Q510K}$ also has a decreased melting temperature (Fig. 6G) (van Vlimmeren et al, 2024; Jiang et al, 2025), suggesting that these mutants have a destabilized auto-inhibited state and/or decreased overall thermal stability. Collectively, our proximity-labeling data and stability measurements paint a plausible picture for mutation-dependent

SHP2 translocation to the mitochondrial matrix, dictated by changes in protein stability and association with mitochondrial chaperones.

## Allosteric inhibition remodels the SHP2 interactome by stabilizing the auto-inhibited state

Our results illustrate how SHP2 conformation and stability can affect its interactome and localization. Several allosteric inhibitors of SHP2 are being pursued as cancer therapies, and many of these molecules bind between the C-SH2 and PTP domains to stabilize the auto-inhibited state (Fig. 7A) (Yuan et al, 2020; Guo et al, 2024). One such inhibitor, TNO155 (batoprotafib), is currently in clinical trials (LaMarche et al, 2020). Consistent with its known mechanism of binding, incubation with TNO155 increased the melting temperatures of SHP2$^{WT}$ and several mutants, albeit to varying extents (Fig. 7B). This variability likely reflects the extent to which SHP2 adopts an open conformation that lacks the inhibitor binding site. Indeed, SHP2$^{E76K}$, which primarily inhabits an open conformation, showed the smallest increase (~2 °C) (Pádua et al, 2018; Chen et al, 2016). By contrast, SHP2$^{Q510K}$ showed a dramatic increase (13 °C), but the molecular basis for this large effect is not immediately obvious.

Given that TNO155 stabilizes the auto-inhibited state of SHP2, we hypothesized that it could remodel the SHP2 interactome. To test this, we preincubated SHP2$^{WT}$-TurboID-transfected cells for 4 h with 1 µM TNO155 prior to conducting proximity-labeling proteomics experiments. Direct comparison of TNO155-treated and untreated cells showed that TurboID labeling skewed toward the untreated cells, both without and with EGF stimulation (Fig. 7C,D; Dataset EV2). These results are consistent with auto-inhibited SHP2, stabilized by TNO155 binding, being less capable of engaging in protein–protein interactions. Among the proteins that showed reduced labeling by SHP2$^{WT}$-TurboID upon TNO155 treatment were Gab1, as well as ATPAF1, the mitochondrial ATP synthase assembly factor described earlier (Fig. 7E, left). Remarkably, many SHP2$^{WT}$ interactors were still significantly enriched upon TNO155 treatment (Fig. 7E, right). Some of these proteins could be co-localized proteins rather than direct interactors; however, multiple components of the Paf1 complex, which is known to bind SHP2 (Takahashi et al, 2011), were not impacted by TNO155 treatment. Thus, some of these proteins could be interactors that engage SHP2 at inhibitor-insensitive interfaces, such as the disordered C-terminal tail. Alternatively, they might bind SHP2 with very high affinity in the open state, rendering it unable to bind TNO155, as has been shown for other SHP2 inhibitors and interactors (LaRochelle et al, 2018). We were intrigued to find that several mitochondrial SHP2 interactors were observed both in the TNO155-sensitive and TNO155-insensitive categories, suggesting that TNO155 may not fully disrupt SHP2 mitochondrial import, despite stabilizing the auto-inhibited state (Fig. 7E, proteins marked with asterisks). These different mechanisms warrant further investigation, as they could have implications for inhibitor efficacy. Collectively, these results show that the allosteric inhibitor TNO155 alters SHP2 interactions. These results also highlight the exquisite sensitivity and broad applicability of TurboID proximity labeling to examine state-dependent changes in SHP2 function.

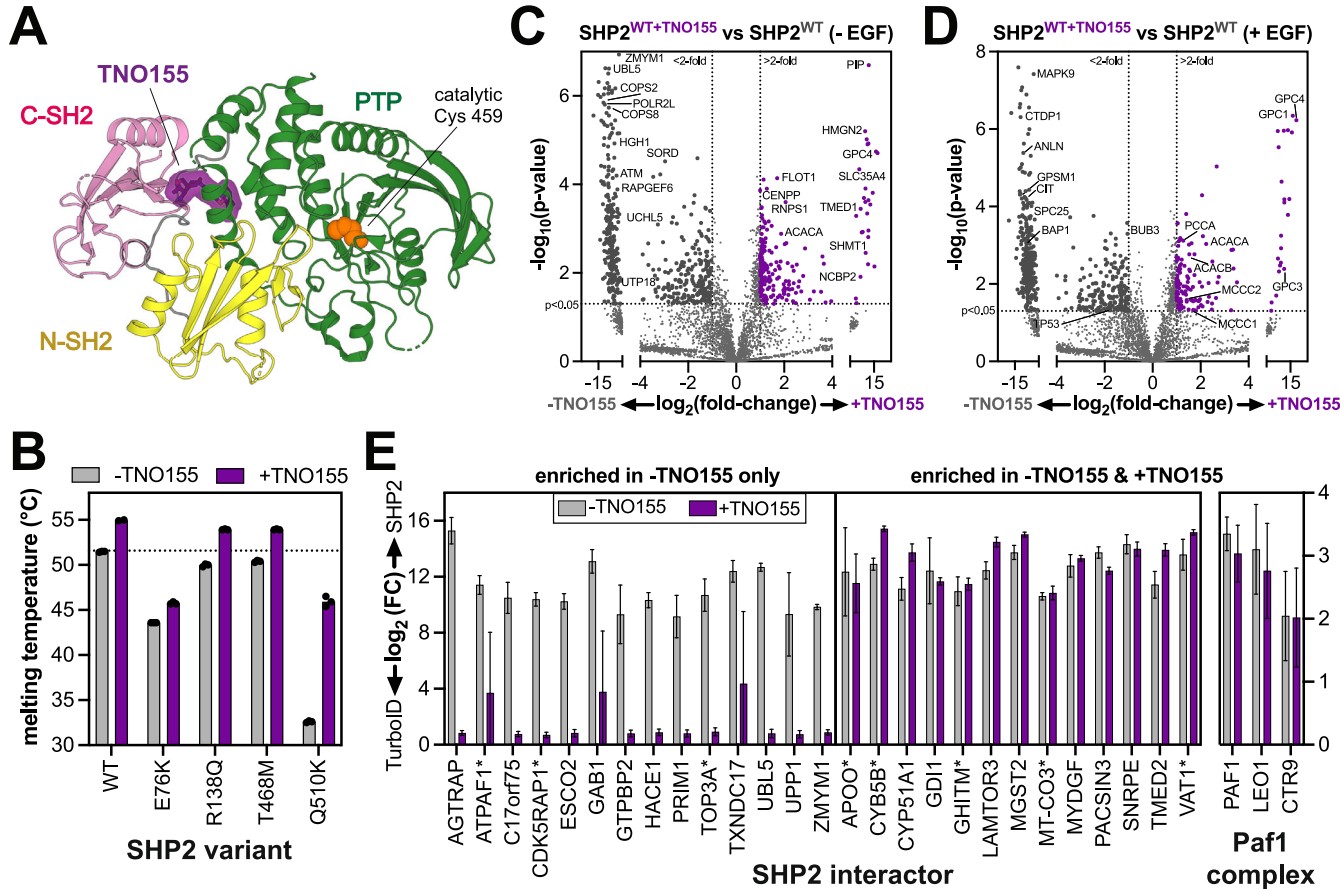

**Figure 7. Alteration of SHP2 interactions by the allosteric inhibitor TNO155.**

(A) Structure of SHP2 with allosteric inhibitor TNO155 docked in between the C-SH2 domain and the PTP domain (PDB code 7JVM). (B) Melting temperatures of SHP2 variants measured by differential scanning fluorimetry in the absence and presence of 10 μM TNO155. (C) Volcano plot highlighting enriched proteins in SHP2$^{WT}$, compared to SHP2$^{WT+TNO155}$ in the absence of EGF stimulation ($n = 3$ biological replicates). 210 proteins were significantly enriched in inhibitor-treated samples over untreated samples, and 480 proteins were significantly enriched in untreated samples over inhibitor-treated ones. $P$ values were determined using a heteroscedastic, two-tailed $t$-test. (D) Volcano plot highlighting enriched proteins in SHP2$^{WT}$, compared to SHP2$^{WT+TNO155}$ in the presence of EGF stimulation ($n = 3$ biological replicates). $P$ values were determined using a heteroscedastic, two-tailed $t$-test. (E) Proximity-labeling signal for representative SHP2$^{WT}$-TurboID interactors relative to the TurboID-only control, comparing enrichments without (gray) and with (purple) TNO155 treatment ($n = 3$ biological replicates). Bar height indicates the ratio of the mean value for SHP2 samples relative to the mean value for control samples, and error bars represent standard deviation after error propagation. Interactor names marked with an asterisk correspond to mitochondrial proteins. Source data are available online for this figure.

# Discussion

Disease-driving mutations in *PTPN11*/SHP2 are almost exclusively missense mutations, many of which disrupt auto-inhibition and hyperactivate SHP2 (Grossmann et al, 2010). Characterization of these mutations has broadened our understanding of the protein-intrinsic regulation of SHP2 and shed light on how dysregulation of SHP2 can cause disease. While SHP2 functions in many signaling pathways, its role in disease has largely been linked to dysregulation of the Ras/MAPK pathway (Asmamaw et al, 2022; Agazie and Hayman, 2003b; Montagner et al, 2005; Bunda et al, 2015; Dance et al, 2008). Recent studies have started to reveal alternate mechanisms of SHP2 dysregulation and pathogenicity. For example, some catalytically inactive NSML mutants drive proliferative signaling through increased adoption of an open conformation, which enhances SH2-phosphoprotein interactions

(Yu et al, 2014; Kontaridis et al, 2006). Other mutations alter substrate- or ligand-binding preferences and rewire signaling interactions (van Vlimmeren et al, 2024; Zhang et al, 2020). This range of mutational effects motivated us to systematically map the cellular interactomes of diverse SHP2 mutants to gain insights into the resulting aberrant cell signaling.

In this study, we show that TurboID proximity labeling allows for in-depth comparison of mutation- and drug-induced changes in SHP2 protein–protein interactions and localization. Our results show that individual SHP2 mutants have overlapping but partly distinct interactomes that reveal both divergent and convergent modes of dysregulation. For example, the melanoma mutant SHP2$^{R138Q}$ has a dysfunctional C-SH2 domain, and interactions with this mutant are less dependent on tyrosine-phosphorylated proteins than SHP2$^{WT}$. The Noonan Syndrome mutant SHP2$^{T42A}$ enhances intrinsic phosphoprotein binding affinity and alters

binding specificity, increasing its interaction with the cell surface receptor MPZL1. SHP2[T42A] functionally phenocopies NSML mutants SHP2[T468M] and SHP2[Q510E], which adopt an open conformation that also stabilizes their interaction with MPZL1. This observation highlights how mechanistically distinct mutations can yield similar cellular and clinical phenotypes.

A key aspect of our study is the juxtaposition of SHP2 proximity-labeling data with predictions of SH2 binding sites, revealing a number of tyrosine phosphorylation sites that could recruit and modulate SHP2 function. The PTP domain of SHP2 also interacts with phosphosites, to dephosphorylate them. From our proximity-labeling datasets alone, we cannot differentiate binders and co-localized proteins from direct substrates, and the SHP PTP domain likely does not have strong sequence preferences that could be used to predict substrates (Zhu et al, 2022). Recently, a substrate-trapping mutation in PTP1B, another phosphatase, was successfully combined with proximity labeling to selectively enhance the signal for substrates over other proximal proteins (Bonham et al, 2023). However, certain SHP2 substrate-trapping mutations, most notably C459S, cause dramatic structural rearrangements that will have collateral effects on SHP2 localization and the measured interactome (Pádua et al, 2018; Sha et al, 2023). Thus, other substrate-trapping mutations such as D425A or T466A should be considered (Agazie & Hayman, 2003a; Merritt et al, 2006), but careful assessment of the structural effects of any substrate-trapping mutants will be required.

Beyond the inability of our approach to directly identify SHP2 substrates, our study has other limitations that could be addressed in follow-up investigations. For example, for this work, we chose to transiently transfect SHP2-TurboID fusion constructs expressed from a constitutive promoter. Although our expression levels were similar to endogenous SHP2 levels on average (Fig. EV1F), there is likely cell-to-cell variation that could introduce noise into the datasets. Other phosphatase proximity-labeling studies have used stable isogenic cell lines and inducible promoters to mitigate these issues (St-Denis et al, 2016). Likewise, the choice of proximity-labeling platform (e.g., BioID, TurboID, APEX), as well as orientation of the fusion protein (N- or C-terminal fusion) could also impact signal-to-noise or the signaling and localization properties of the protein of interest. Furthermore, most of the candidate interactors that we report here have not yet been validated by orthogonal methods, such as antibody-based proximity labeling or co-immunopurification. Therefore, some of these hits may be reflective of co-localization, indirect interactions, or even methodological artifacts (e.g., due to TurboID fusion). As such, the SHP2 interactions and localization reported here should be interpreted with caution and verified in future work.

Beyond the aforementioned technical considerations, we also note that our study was confined to SHP2 knock-out HEK 293 cells with EGF stimulation at a single timepoint. Wild-type SHP2 and SHP2 mutants operate downstream of a variety of receptors and have functional roles in a range of cell types (Chen et al, 2016; Choi et al, 2019; Marasco et al, 2020; Yi et al, 2020; Olou et al, 2023). Building on the framework described here, future studies should focus on exploring the SHP2 interactome in other signaling contexts and cell states. Importantly, despite these limitations, our proximity-labeling study identified several previously observed SHP2 interactions that have been detected in a variety of cell lines using a spectrum of approaches, including affinity-purification mass spectrometry, other proximity-labeling modalities, and two-hybrid assays. Thus, our approach is likely capturing many biologically important SHP2 interactions and functions, and as such, our study provides a valuable resource for defining SHP2 interaction networks and their dysregulation by disease mutations.

A notable feature of our proximity-labeling datasets is that we observed a strong mitochondrial signal for SHP2[WT] and several mutants. While these results will require further validation and corroboration, they point to an underexplored area of SHP2 biology. The role of SHP2 in mitochondrial biology is not well-understood, but some studies have begun to reveal how SHP2 might modulate mitochondrial function. Hyperactive SHP2 variants can dysregulate mitochondrial function, causing oxidative stress in cells (Zhu et al, 2020; Xu et al, 2013; Zheng et al, 2013). SHP2[WT] regulates the inflammasome through interaction with ANT1, and SHP2[E76K] can enhance mitochondrial metabolism through excessive dephosphorylation of STAT3 (Guo et al, 2017; Kan et al, 2024). Here, we show that several SHP2 variants co-localize with components of the electron transport chain. Furthermore, we show that SHP2[R138Q], SHP2[T468M] and SHP2[Q510K], and SHP2[E76K] have increased mitochondrial localization relative to SHP2[WT], and some of these mutants can alter mitochondrial respiration. Our data suggest that some mutants have enhanced interactions with mitochondrial import machinery or associated chaperones, due to a change in protein conformation or stability. However, we note that some SHP2 mutants with decreased thermal stability do not appear to have increased mitochondrial localization, suggesting that there is nuance to the molecular mechanism of SHP2 mitochondrial import. These observations motivate further studies probing SHP2 mislocalization and mitochondrial dysregulation. These follow-up studies should be carried out using complementary approaches to proximity labeling, in disease-relevant cell lines with endogenously expressed wild-type SHP2 and SHP2 mutants. More broadly, our work lays the foundation for new avenues of investigation into SHP2 signaling and pathogenicity. Ultimately, we envision that the experimental framework laid out in this study will be useful for the unbiased dissection of mutational effects in other disease-relevant proteins.

## Methods

### Reagents and tools table

| Reagent/resource | Reference or source | Identifier or catalog number |
| --- | --- | --- |
| **Experimental models** | | |
| HEK 293 SHP2 knock-out cells | (Zhu et al, 2022) | - |
| U-2OS cells | Gaublomme Lab | ATCC - U-2 OS HTB-96 ™ |
| **Recombinant DNA** | | |
| pGEX-4TI SHP2 WT | Addgene | #8322 |
| V5-TurboID-NES_pCDNA3 | Addgene | #107169 |
| pCGN N-Ras wt | Addgene | #14723 |
| pCMV5 mouse Src | Addgene | #13663 |
| pCMV5 mouse Src (Δ528–535) | (Jiang et al, 2025) | - |
| pCDNA3-hPZR-WT | Bennett Lab | - |

| Reagent/resource | Reference or source | Identifier or catalog number |
|---|---|---|
| pET-SUMO-SHP2-N-SH2(WT)-Avi | (van Vlimmeren et al, 2024) | - |
| pET-SUMO-SHP2-C-SH2(WT)-Avi | (Li et al, 2023) | - |
| X5-Y-X5 library | (Li et al, 2023) | - |
| pEF myc-SHP2(WT&mutants) | This study | - |
| pEF myc-SHP2(WT&mutants)-TurboID | This study | - |
| pEF-3xFLAG-ATPAF1 | This study, gene synthesized by Twist Biosciences | - |
| **Antibodies** | | |
| Myc (Clone 9E10) | Invitrogen | R95025 (1:5000) |
| c-Src | CST | 2123S (1:1000) |
| B-actin | Sigma | A5441 (1:5000) |
| Phospho-Tyrosine (P-Tyr-1000) MultiMab | CST | 8954S (1:2000) |
| Pzr | CST | 9893S (1:1000) |
| FLAG | MPBio | 08L100031 (1:1000) |
| Vinculin | CST | 13901S (1:1000) |
| HA (TCL) | Sigma | SAB5600116 (1:1000) |
| HA (IP) | Sigma | 11867423001 (1:1000) |
| IRDye® 680RD Goat anti-Rabbit IgG | LiCor | 926-68071 (1:10,000) |
| IRDye® 800CW Goat anti-Mouse IgG | LiCor | 926-32210 (1:10,000) |
| IRDye 800CW Goat anti-Rat IgG | LiCor | 926-32219 (1:10,000) |
| PY20-PerCP-eFluor 710 pan-phosphotyrosine | Thermo Fisher | 46-5001-41 (1:50) |
| SHP2 (D50F2) | CST | 3397 (1:100) |
| Anti-rabbit IgG (H + L), F(ab')2 Fragment (Alexa Fluor® 488 Conjugate) | CST | 4412 (1:2000) |
| Alexa Fluor® 647 Phalloidin | CST | 8940 (1:100) |
| TOM20 Polyclonal Antibody, CoraLite® 555 | Thermo Fisher | CL55511802100UL (1:100) |
| MYC-tag Monoclonal Antibody (1A5A2), CoraLite® Plus 750 | ProteinTech | CL750-60003 (1:100) |
| 4',6-diamidino-2-phenylindole (DAPI) | Fisher | D1306 (1 µg/mL) |
| **Oligonucleotides and other sequence-based reagents** | | |
| TruSeq-eCPX-Fwd | (Li et al, 2023) | - |
| TruSeq-eCPX-Rev | (Li et al, 2023) | - |
| D701-D712 indexing primers | Illumina | - |
| D501-D508 indexing primers | Illumina | - |
| **Chemicals, Enzymes and other reagents** | | |
| DMEM Cell Culture Medium | Gibco | 10566-024 |

| Reagent/resource | Reference or source | Identifier or catalog number |
|---|---|---|
| Gibco Fetal Bovine Serum | Fisher | A5256701 |
| Anti-Anti (100x) (100 mL) | Gibco | 15240-062 |
| 0.05% Trypsin-EDTA (1x) (100 mL) | Gibco | 25300-054 |
| PEI MAX - Transfection Grade Linear Polyethylenimine Hydrochloride (MW 40,000) | Fisher | NC1038561 |
| Human Epidermal Growth Factor | Sigma | #E9644 |
| Gibco™ DPBS, no calcium, no magnesium | Fisher | 14-190-250 |
| Biotin 98.0 + %, TCI America™ | Fisher | B04635G |
| Pierce™ Anti-c-Myc Magnetic Beads | Fisher | PI88842 |
| Thermo Scientific™ Pierce™ Streptavidin Magnetic Beads (5 mL) | Fisher | PI88817 |
| Promega Sequencing Grade Modified Trypsin, Frozen | Fisher | PR-V5113 |
| Eppendorf™ DNA LoBind Microcentrifuge Tubes | Fisher | 13-698-791 |
| Lucigen Corporation Overexpress C43(DE3) Chemically Competent Cells | Fisher | NC9581214 |
| BL21(DE3) Competent E. coli | Fisher | C2527H |
| MC1061 F- Electrocompetent Cells (≥ 4 × 10^10 cfu/µg, 24 rxns DUOs) | Lucigen | 60514-2 |
| Dynabeads™ FlowComp™ Flexi Kit | Fisher | 11061D |
| Ubpbio IPTG, 100 G | Fisher | 50-112-6936 |
| Invitrogen™ Molecular Probes™ DiFMUP (6,8-Difluoro-4-Methylumbelliferyl Phosphate) 5 mg | Fisher | D6567 |
| Pierce™ Anti-HA Magnetic Beads | Fisher | 88836 |
| Pierce™ Anti-DYKDDDDK Anti-flag Magnetic Agarose | VWR | A36797 |
| 16% Paraformaldehyde | Fisher | 50-980-487 |
| SYPRO Orange Protein Gel Stain | Fisher | S-6650 |
| MitoSOX™ Mitochondrial Superoxide Indicators for live-cell imaging | Invitrogen | M36006 |
| Mitochondria/Cytosol Fractionation Kit | Fisher | 89874 |
| Recombinant purified kinase cocktail (c-Src, c-Abl, AncSZ, Eph1B) | (Li et al, 2023) | - |
| Purified N-SH2(WT)-AviTag | This study | - |
| Purified C-SH2(WT)-AviTag | This study | - |
| Purified SHP2(WT)-TurboID | This study | - |
| Purified SHP2(E76K)-TurboID | This study | - |
| **Software** | | |

| Reagent/resource | Reference or source | Identifier or catalog number |
|---|---|---|
| SpectroMine software version 4.2.230428 | https://biognosys.com/software/spectromine/ | - |
| GraphPad Prism v10 | https://www.graphpad.com/ | - |
| Spectronaut software version 18.6 | https://biognosys.com/software/spectronaut/ | - |
| Image Studio Lite Version 5.2 | https://www.licorbio.com/image-studio-lite | - |
| ImageJ/Fiji v1.54 g | https://fiji.sc/?Downloads | - |
| **Other** | | |
| Orbitrap Q Exactive HF mass spectrometer | Thermo Fisher Scientific | - |
| Andor Dragonfly | Oxford Systems | - |
| Synergy Neo2 multi-mode reader | BioTek | - |
| Odyssey Fc | LiCor | - |
| NextSeq 500 | Illumina | - |
| Step-One Plus RT-PCR thermocycler | Applied BioSciences | - |
| Attune NxT Flow Cytometer | Thermo Fisher Scientific | - |
| Trans-Blot Turbo | Bio-Rad | |
| Fisherbrand Sonic Dismembrator | Thermo Fisher Scientific | - |
| AKTA Pure | GE | - |
| HisTrap HP, 5 ml | Fisher | 17524802 |
| HITRAP CAPTO Q, 5 ML | Cytiva | 11001303 |
| Superdex 75 16/600 gel filtration column | GE | 28989333 |
| ZymoPURE II Plasmid Maxiprep Kit | Zymo | #D4203 |
| Thermo Scientific™ Pierce™ BCA Protein Assay Kit | Fisher | PI23225 |
| QuantiFluor® dsDNA System (Promega) | Fisher | PRE2670 |
| NextSeq 500 Mid-Output v2 Kit (150 cycles) | Illumina | FC-404-2001 |

All new materials and reagents generated through this study will be made freely available upon request, and there are no restrictions on the use of any materials and reagents generated through this work.

## Methods and protocols

### Cell culture
All cell lines were cultured in a 37 °C tissue culture incubator with 5% $CO_2$. Cells were discarded by passage 25 and tested for mycoplasma every 3 months. HEK 293 cells were grown in Dulbecco's Modified Eagle Medium (DMEM) with 10% Fetal Bovine Serum (FBS) and 1% penicillin/streptomycin. HEK 293 SHP2 knock-out and U-2 OS cells, reported previously (Zhu et al, 2022), were grown in DMEM with 10% FBS. Human epidermal growth factor was purchased in lyophilized form (#E9644, Sigma) and reconstituted in 10 mM acetic acid.

### Mammalian expression DNA constructs
The SHP2 gene was cloned from the pGEX-4TI SHP2 WT plasmid from Ben Neel (Addgene plasmid #8322) (O'Reilly et al, 2000). All SHP2 genes for mammalian expression and proteomics experiments, both with and without C-terminal TurboID fusion, were cloned into a pEF vector with an N-terminal myc-tag. TurboID was cloned from the V5-TurboID-NES_pCDNA3 plasmid, which was a gift from Alice Ting (Addgene, #107169) (Branon et al, 2018). The mouse c-Src gene was expressed from the pCMV5 mouse Src plasmid, a gift from Joan Brugge and Peter Howley (Addgene plasmid #13663). The C-terminal regulatory tail was deleted to make a constitutively active c-Src construct. The pCGN N-Ras wt plasmid was a gift from Adrienne Cox (Addgene, # 14723) (Fiordalisi et al, 2001). The pCDNA3-hPZR-WT plasmid was a gift from Anton Bennett. The coding sequence for human ATPAF1 was synthesized by Twist Biosciences and cloned into a mammalian expression vector with an N-terminal 3xFLAG-tag.

### Myc-tag affinity-purification mass spectrometry proteomics
Replicates consist of separate transfections and downstream processing. $10 \times 10^6$ HEK 293 SHP2 KO cells were seeded in a 15 cm plate. The next day, cells were transfected overnight with 25 µg DNA (Myc-tagged $SHP2^{WT}$, $SHP2^{T42A}$ or empty pEF vector) in 2.5 mL empty DMEM using 75 µg of polyethylenimine (PEI). The next morning, the transfection medium was replaced by pre-warmed empty DMEM. 36 h after transfection, cells were harvested by scraping and washed twice in 1 mL phosphate-buffered saline (PBS) pre-warmed to 37 °C. Half of the samples were resuspended in 1 mL PBS with 100 ng/mL epidermal growth factor (EGF), the other half was resuspended in 1 mL PBS. Samples were placed in a 37 °C heat block for 10 min and then placed on ice to stop the reaction. Cells were spun down in a refrigerated tabletop centrifuge. Cells were lysed in non-denaturing lysis buffer (20 mM Tris-HCl, pH 8.0, 137 mM NaCl, 10% glycerol, 1% Nonidet P-40 (NP-40), 2 mM EDTA, with protease inhibitors and phosphatase inhibitors added fresh) for 25 min at 4 °C while rotating, then spun down in a refrigerated tabletop centrifuge at $17,000 \times g$ for 15 min. Protein concentration was determined using a bicinchoninic acid (BCA) assay. About 1500 µg of protein was used in an immuno-precipitation using 75 µL of magnetic anti-Myc beads. Samples were left overnight at 4 °C while rotating. The next day, samples were prepared for proteomics.

### Preparation of TurboID samples
Replicates consist of separate transfections and downstream processing. $10 \times 10^6$ HEK 293 $SHP2^{KO}$ cells were seeded in a 15 cm plate. The next day, cells were transfected overnight with 25 µg of SHP2-TurboID DNA in 2.5 mL of empty DMEM using 75 µg of PEI. The next morning, the transfection medium was replaced by pre-warmed empty DMEM with 2.5% FBS. 36 h after transfection, the media was aspirated and replaced by one of the following: empty DMEM (unstimulated control), empty DMEM with 100 ng/mL EGF (stimulated control), 100 µM biotin in DMEM

(unstimulated sample), or 100 µM biotin and 100 ng/mL EGF in DMEM (stimulated sample). Cells were placed at 37 °C for 10 min and then promptly placed on ice. Cells were harvested in 10 mL of ice-cold PBS by scraping and washed twice in 5 mL cold PBS. Cells were lysed in non-denaturing lysis buffer (20 mM Tris-HCl, pH 8.0, 137 mM NaCl, 10% glycerol, 1% Nonidet P-40 (NP-40), 2 mM EDTA, with protease inhibitors and phosphatase inhibitors added fresh) for 25 min while rotating at 4 °C, then spun down in a refrigerated tabletop centrifuge at $17,000 \times g$ for 15 min. Protein concentration was determined using a BCA assay. About 1500 µg of protein was used in an immunopurification step with 75 µL of magnetic anti-Streptavidin beads. Samples were left overnight at 4 °C while rotating. The next day, samples were prepared for proteomics.

### Preparation of proteomic IP-samples

Samples were prepared according to a previously reported protocol (Cho et al, 2020). Beads were washed twice in 1 mL 50 mM Tris-HCl (pH 8.0), then twice more in 200 µL 2 M urea in 50 mM Tris (pH 8.0). Beads were resuspended in 80 µL of 2 M urea in 50 mM Tris-HCl (pH 8.0), containing 1 mM DTT and 0.4 µg trypsin at room temperature while shaking moderately. After 1 h, the supernatant was transferred to a new tube. Beads were washed twice with 60 µL of 2 M urea in 50 mM Tris (pH 8.0). The washes were combined with the on-bead digest 80 µL supernatant to a total volume of 200 µL. Dithiothreitol (DTT) was added to a final concentration of 4 mM, and incubated at room temperature while shaking moderately. After 30 min, iodoacetamide (IAA) was added to a final concentration of 10 mM and incubated at room temperature in the dark while shaking moderately. After 45 min, an additional 0.5 µg of trypsin was added to each sample. Digestion proceeded overnight at room temperature while shaking. After overnight digestion, samples were acidified using formic acid (FA) to ~1% (vol/vol). Samples were desalted using C18 stagetips. Briefly, tips were conditioned with 100 µL of 100% MeOH, 100 µL of 0.2% FA/60% Acetonitrile (ACN), and twice with 100 µL of 0.2% FA. Acidified peptides were loaded onto the stagetips and washed twice with 100 µL of 0.2% FA. Peptides were eluted with 50 µL of 0.2% FA/60% ACN, and dried in a vacuum centrifuge at room temperature. Peptides were stored at −80 °C until injection.

### Preparation of lysate proteomic samples

About 50 µg of protein was used as input for the lysate samples and brought up to a total volume of 200 µL in 50 mM Tris-HCl (pH 8.0). Dithiothreitol (DTT) was added to a final concentration of 5 mM. Samples were incubated for 45 min at room temperature while shaking moderately. Iodoacetamide (IAA) was added to a final concentration of 10 mM and shaken for another 45 min in the dark at room temperature. About 10 µL of magnetic SP3 beads at a concentration of 50 mg/mL were added to the samples. About 250 µL of EtOH was added, and samples were placed on a magnetic rack. Supernatant was aspirated, and beads were washed three times for 2 min with 1 mL 80% EtOH. Samples were reconstituted in 200 µL fresh 100 mM ammonium bicarbonate (ABC), pH 7.7. Trypsin was added in a 1:50 ratio, and samples were digested overnight while lightly shaking. In the morning, the supernatant was transferred to a new tube and spun down at $17,000 \times g$ for 5 min. About 175 µL of supernatant was transferred to another new

tube, and 1% FA (v/v) was added. Samples were dried down in a vacuum centrifuge at room temperature, and dried peptides were stored at −80 °C until injection.

### LC-MS/MS analysis on a Q-exactive HF for immunoprecipitated samples

Samples were resuspended in 13 µL 3% ACN/0.2% FA. About 6 µL was injected and analyzed on a Waters M-Class UPLC using a 25 cm Ionopticks Aurora column coupled to a benchtop Thermo Fisher Scientific Orbitrap Q Exactive HF mass spectrometer. Peptides were separated at a flow rate of 400 nL/min with a 100 min gradient, including sample loading and column equilibration times. Data were acquired in data-dependent mode using Xcalibur 4.1 software. MS1 Spectra were measured with a resolution of 120,000, an AGC target of 3e6 and a mass range from 300 to 1800 m/z. Up to 12 MS2 spectra per duty cycle were triggered at a resolution of 60,000, an AGC target of 1e5, an isolation window of 0.8 m/z, a normalized collision energy of 28, a scan range of 200 to 2000 m/z, and a fixed first mass of 110 m/z.

### Quantification and statistical analyses of immunoprecipitated samples

All raw data were analyzed with SpectroMine software version 4.2.230428 using a UniProt database (Homo sapiens, UP000005640). Carbamidomethylation on cysteines was set as a fixed modification. Oxidation of methionine and protein N-terminal acetylation were set as variable modifications, with a maximum of five variable modifications. Trypsin/P was set as the digestion enzyme, and up to two missed cleavages were permitted. For identification, we applied a maximum false discovery rate of 1% on the protein and peptide level. We required one or more unique or razor peptides for protein identification. Then, noise was added from the randomly sampled lower range of the limit of detection to all raw intensities. Protein groups with <4.99 average MS/MS counts were removed from further analysis. Protein group intensities were normalized for the total intensity of all observable protein groups in that sample. Normalized protein group intensities were log2-transformed and averaged, from which fold-changes were calculated. $P$ values were calculated using a two-tailed, heteroscedastic $t$-test. Gene Ontology analysis was performed using Panther or String. Submitochondrial localization was determined by gene ontology terms for Cellular Component, as well as identified compartments by cross-linked assisted spatial proteomics (Zhu et al, 2024).

### LC-MS/MS analysis of total proteome samples

Whole proteome, label-free MS analysis was performed by data-independent acquisition (DIA). For this type of LC-MS/MS analysis, about 1 µg of total peptides were analyzed on a Waters M-Class UPLC using a 15 cm IonOpticks Aurora Elite column (75-µm inner diameter; 1.7-µm particle size; heated to 45 °C) coupled to a benchtop Thermo Fisher Scientific Orbitrap Q Exactive HF mass spectrometer. Peptides were separated at a flow rate of 400 nL/min with a 90 min gradient, including sample loading and column equilibration times. Data were acquired in data-independent mode using Xcalibur 4.5 software. MS1 Spectra were measured with a resolution of 120,000, an AGC target of 3e6 and a mass range from 350 to 1600 m/z. Per MS1, 29 equally distanced, sequential segments were triggered at a resolution of 30,000, an AGC target

of 3e6, a segment width of 43 m/z, and a fixed first mass of 200 m/z. The stepped collision energies were set to 22.5, 25, and 27.

### Quantification and statistical analyses of total proteome samples

All DIA data were analyzed with Spectronaut software version 18.6 (Bruderer et al, 2015), using directDIA analysis methodology against a UniProt database (Homo sapiens, UP000005640). Carbamidomethylation on cysteines was set as a fixed modification. Oxidation of methionine and protein N-terminal acetylation were set as variable modifications. Trypsin/P was set as the digestion enzyme. Normalization was done per Spectronaut's "automatic normalization". For identification, we applied a maximum false discovery rate of 1% on the protein and peptide level. We required one or more unique or razor peptides for protein identification. Then, noise was added from the randomly sampled lower range of the limit of detection to all raw intensities. Protein groups with <4.99 average MS/MS counts were removed from further analysis. Protein group intensities were further normalized for the total intensity of all observable protein groups in that sample. Normalized protein group intensities were log2-transformed and averaged, from which fold-changes were calculated. P values were calculated using a two-tailed, heteroscedastic *t*-test.

### Purification of SH2 domains (adapted from: van Vlimmeren et al, 2024)

The SHP2 full-length, wild-type gene used as the template for all SHP2 constructs in this study was cloned from the pGEX-4TI SHP2 WT plasmid. SHP2 SH2 domains were cloned into a $His_6$-SUMO-SH2-Avi construct, as described previously (Li et al, 2023). C43(DE3) cells were transformed with plasmids encoding both the respective SH2 domain and the biotin ligase BirA. Cells were grown in LB supplemented with 50 µg/mL kanamycin and 100 µg/mL streptomycin at 37 °C until cells reached an optical density at 600 nm ($OD_{600}$) of 0.5. IPTG (1 mM) and biotin (250 µM) were added to induce protein expression and ensure biotinylation of SH2 domains, respectively. Protein expression was carried out at 18 °C overnight. Cells were centrifuged and subsequently resuspended in lysis buffer (50 mM Tris pH 7.5, 300 mM NaCl, 20 mM imidazole, 10% glycerol, and freshly added 2 mM β-mercaptoethanol). The cells were lysed using sonication (Fisherbrand Sonic Dismembrator), and spun down at 14,000 rpm for 45 min. The supernatant was applied to a 5 mL Ni-NTA column (Cytiva). The resin was washed with ten column volumes of lysis buffer and wash buffer (50 mM Tris, pH 7.5, 50 mM NaCl, 20 mM imidazole, 10% glycerol, and freshly added 2 mM β-mercaptoethanol). The protein was eluted off the Ni-NTA column in elution buffer (50 mM Tris, pH 7.5, 50 mM NaCl, 500 mM imidazole, and 10% glycerol) and brought onto a 5 mL HiTrap Q Anion exchange column (Cytiva). The column was washed using Anion A buffer (50 mM Tris, pH 7.5, 50 mM NaCl, and 1 mM TCEP). Protein elution off the column was induced through a salt gradient between Anion A buffer and Anion B buffer (50 mM Tris, pH 7.5, 1 M NaCl, and 1 mM TCEP). The eluted protein was cleaved at the $His_6$-SUMO tag by the addition of 0.05 mg/mL $His_6$-tagged Ulp1 protease at 4 °C overnight. This cleavage cocktail was flowed through a 2 mL Ni-NTA gravity column (Thermo Fisher) to isolate the cleaved protein away from uncleaved protein and Ulp1. Finally, the cleaved protein was purified by size-exclusion chromatography on a Superdex 75 16/600 gel filtration column (Cytiva) equilibrated with SEC buffer

(20 mM HEPES, pH 7.4, 150 mM NaCl, and 10% glycerol). Pure fractions were pooled and concentrated, and flash frozen in liquid $N_2$ for long-term storage at −80 °C.

### Purification of full-length SHP2 proteins (adapted from: van Vlimmeren et al, 2024)

Full-length SHP2 variants were cloned into a pET28-His-TEV plasmid from the pGEX-4TI SHP2 WT plasmid. BL21(DE3) cells were transformed with the respective plasmids, and were grown in LB supplemented with 100 µg/mL kanamycin at 37 °C until cells reached an $OD_{600}$ of 0.5. IPTG (1 mM) was added to induce protein expression, which was carried out at 18 °C overnight. Cells were centrifuged and subsequently resuspended in lysis buffer (50 mM Tris, pH 7.5, 300 mM NaCl, 20 mM imidazole, 10% glycerol, and freshly added 2 mM β-mercaptoethanol). The cells were lysed using sonication (Fisherbrand Sonic Dismembrator), and spun down at 14,000 rpm for 45 min. The supernatant was applied to a 5 mL Ni-NTA column (Cytiva). The resin was washed with ten column volumes of lysis buffer and wash buffer (50 mM Tris, pH 7.5, 50 mM NaCl, 20 mM imidazole, 10% glycerol, and freshly added 2 mM β-mercaptoethanol). The protein was eluted off the Ni-NTA column in elution buffer (50 mM Tris, pH 7.5, 50 mM NaCl, 500 mM imidazole, and 10% glycerol) and brought onto a 5 mL HiTrap Q Anion exchange column (Cytiva). The column was washed using Anion A buffer (50 mM Tris, pH 7.5, 50 mM NaCl, and 1 mM TCEP). Protein elution off the column was induced through a salt gradient between Anion A buffer and Anion B buffer (50 mM Tris pH 7.5, 1 M NaCl, 1 mM TCEP). The eluted protein was cleaved at the $His_6$-TEV tag by the addition of 0.10 mg/mL of $His_6$-tagged TEV protease at 4 °C overnight. This cleavage cocktail was flowed through a 2 mL Ni-NTA gravity column (Thermo Fisher) to separate the cleaved protein from uncleaved protein and TEV protease. Finally, the cleaved protein was purified by size-exclusion chromatography on a Superdex 200 16/600 gel filtration column (Cytiva) equilibrated with SEC buffer (20 mM HEPES, pH 7.5, 150 mM NaCl, and 10% glycerol). Pure fractions were pooled and concentrated, and flash frozen in liquid $N_2$ for long-term storage at −80 °C.

### SH2 specificity profiling (adapted from: van Vlimmeren et al, 2024)

Electrocompetent MC1061 cells were transformed with ~100 ng of the strep-tagged $X_5$-Y-$X_5$ library (Li et al, 2023). After 1 h recovery in 1 mL LB, cells were further diluted into 250 mL LB + 0.1% chloramphenicol. 1.8 mL of overnight culture was used to inoculate 100 mL LB + 0.1% chloramphenicol, and grown until OD600 reached 0.5. About 20 mL of cell suspension was induced at 25 °C using a final concentration of 0.4% arabinose until the $OD_{600}$ reached ~1 (after about 4 h). The cells were spun down at 4000 rpm for 15 min, and the pellet was resuspended in PBS so that the $OD_{600}$ ~1.5. The cells were stored in the fridge and used within a week.

For each sample, 150 µL of Dynabeads™ FlowComp™ Flexi Kit were washed twice in 1 mL SH2 buffer (50 mM HEPES, pH 7.5, 150 mM NaCl, 1 mM TCEP, and 0.2% BSA) on a magnetic rack. The beads were then resuspended in 150 µL of SH2 buffer. SH2 domains were thawed quickly, and 20 µM of protein was added to the beads. SH2 buffer was added up to 300 µL, and the suspension was incubated for 1 h at 4 °C while rotating. After 1 h, the suspension was placed on a magnetic rack and washed twice with 1 mL SH2 buffer.

About 150 µL of prepared cells per sample were spun down at 4000 rpm for 5 min at 4 °C. Kinase screen buffer was prepared (50 mM Tris, 10 mM magnesium chloride, 150 mM sodium chloride; add 2 mM sodium orthovanadate and 1 mM TCEP fresh), and the cells of each sample were resuspended in 100 µL kinase screen buffer. Kinases c-Src, c-Abl, AncSZ, Eph1B were added to a final concentration of 2.5 µM each, creatine phosphate was added to a final concentration of 5 mM, and phosphokinase was added to a final concentration of 50 µg/mL. The suspension was incubated at 37 °C for 5 min before ATP was added to a final concentration of 1 mM. This mixture was incubated at 37 °C for 3 h. After 3 h, EDTA was added to a final concentration of 25 mM to quench the reaction. The input library control sample was not phosphorylated. These cells were spun down at 4000 rpm for 15 min at 4 °C. The cells were then resuspended in 100 µL of SH2 buffer + 0.1% BSA. Phosphorylation of the cells was confirmed by labeling with the PY20-PerCP-eFluor 710 pan-phosphotyrosine antibody followed by analysis via flow cytometry.

About 100 µL of phosphorylated cells were mixed with 75 µL SH2-beads for 1 h at 4 °C while rotating. After 1 h, samples were placed on a magnetic rack, supernatant was removed, and 1 mL SH2 buffer was added to each sample. This was rotated for 30 min at 4 °C to wash the beads. After this wash, the beads were placed on a magnetic rack, the supernatant was removed, and 50 µL MilliQ was added.

All SH2-selected samples and the input library control were resuspended in 50 µL MilliQ water, vortexed, and boiled for 10 min at 100 °C. The boiled lysate was used as the DNA template in a PCR reaction using the TruSeq-eCPX-Fwd and TruSeq-eCPX-Rev primers. The mixture resulting from this PCR was used directly in a second PCR to append Illumina sequencing adapters and unique 5' and 3' indices to each sample (D700 and D500 series primers). The resulting PCR mixtures were run on a gel, the band of the expected size was extracted and purified, and its concentration was determined using QuantiFluor® dsDNA System (Promega). Samples were pooled at equal molar ratios and sequenced by paired-end Illumina sequencing on a MiSeq or NextSeq instrument using a 150-cycle kit. The number of samples per run, and the loading density on the sequencing chip, were adjusted to obtain at least 1–2 million reads for each index/sample.

### Analysis of data from screens with the $X_5$-Y-$X_5$ library

Deep sequencing data were processed and analyzed as described previously (Li et al, 2023). First, paired-end reads were merged using FLASH (Magoč and Salzberg, 2011). Then, adapter sequences and any constant regions of the library flanking the variable peptide-coding region were removed using Cutadapt (Martin, 2011). Finally, these trimmed files were analyzed using in-house Python scripts in order to count the abundance of each peptide in the library, as described previously (https://github.com/nshahlab/2022_Li-et-al_peptide-display). For each individual amino acid, we counted its occurrence at every position along peptides of the expected length of 11 residues, excluding any sequences containing a stop codon. This generated an $11 \times 20$ counts matrix with each position in the peptide represented by a column (from −5 to +5), and each row represented by an amino acid (ordered by biochemical properties). Frequencies of each amino acid at each position were determined by taking the position-specific count for each amino acid and dividing that by the column total. Frequencies

in a matrix from a selected sample were further normalized against frequencies from an input sample, and the resulting enrichment values were $\log_2$-transformed. Matrices from two independent screens with each SH2 domain were averaged to yield the data in Fig. 2E; Dataset EV4.

### Analysis of human phosphosites using $X_5$-Y-$X_5$ library screens

About 39,235 human phosphosites were downloaded from PhosphoSitePlus (retrieved on October 20, 2022). In order to score each phosphosite using the generated position-weighted count matrices from our $X_5$-Y-$X_5$ library screen, we first calculated the normalized enrichment for each amino acid at each position across the matrices. Then, for each phosphosite, we summed up the $\log_2$-normalized enrichments for each residue according to the enrichment matrix (excluding the central tyrosine), and divided the sum by the number of scored residues (ten in total). Finally, the scores for each screen were normalized such that the minimum and maximum possible theoretical score was set to 0 and 1, respectively. For each SH2 domain, two replicate screens were conducted, and the scoring matrices from each individual replicate were used to score all phosphosites. Then, the scores from each replicate were averaged to yield the data in Dataset EV4.

### Purification of SHP2-TurboID constructs

BL21 (DE3) cells were transformed with SHP2[WT]-TurboID or SHP2[E76K]-TurboID DNA. After heat shock, cells were recovered in 1 mL Luria Broth (LB) for 1 h at 37 °C while shaking. About 100 µL was plated and incubated overnight at 37 °C. The next morning, colonies were resuspended and grown in LB with 100 µg/mL kanamycin at 37 °C until the culture reached an $OD_{600}$ of 0.5. 1 mM Isopropyl β-D-1-thiogalactopyranoside (IPTG) was added to induce protein expression, which proceeded at 18 °C overnight. Cells were pelleted and resuspended in lysis buffer (50 mM Tris pH 7.5, 300 mM NaCl, 20 mM imidazole, 10% glycerol, and freshly added 2 mM β-mercaptoethanol). The cells were lysed by sonication (Fisherbrand Sonic Dismembrator), and centrifuged at 14,000 rpm for 45 min. The supernatant was brought onto a 5 mL Ni-NTA column (Cytiva). The column was washed with ten column volumes of lysis buffer and wash buffer (50 mM Tris, pH 7.5, 50 mM NaCl, 20 mM imidazole, 10% glycerol, and freshly added 2 mM β-mercaptoethanol). The protein was eluted off the Ni-NTA column in elution buffer (50 mM Tris, pH 7.5, 50 mM NaCl, 500 mM imidazole, and 10% glycerol) and applied to a 5 mL HiTrap Q Anion exchange column (Cytiva). The column was then washed using Anion A buffer (50 mM Tris, pH 7.5, 50 mM NaCl, and 1 mM tris(2-carboxyethyl)phosphine (TCEP)). The protein was eluted off the column through a salt gradient between Anion A buffer and Anion B buffer (50 mM Tris, pH 7.5, 1 M NaCl, and 1 mM TCEP). The eluted protein was cleaved at the His6-TEV tag by the addition of 0.10 mg/mL of His6-tagged TEV protease at 4 °C overnight. This cleavage mixture was applied to a 2 mL Ni-NTA gravity column (Thermo Fisher) to separate the cleaved protein from the uncleaved protein and TEV protease. Finally, the cleaved protein was purified by size-exclusion chromatography on a Superdex 200 10/300 gel filtration column (Cytiva) equilibrated with SEC buffer (20 mM HEPES, pH 7.5, 150 mM NaCl, and 10% glycerol). Pure fractions were pooled and concentrated, and flash frozen in liquid N2 for long-term storage at −80 °C.

### DiFMUP dephosphorylation and peptide activation assay

Full-length SHP2 and SHP2-TurboID constructs were diluted in assay buffer (60 mM HEPES, pH 7.2, 75 mM KCl, 75 mM NaCl, 1 mM EDTA, 0.05% Tween-20, with 0.5 mM TCEP freshly added) to a 2X concentration of 0.1 nM. Peptides were serially diluted in assay buffer, with the last point of the concentration series lacking peptide. About 15 µL of peptide dilution mix and 15 µL of 800 µM DiFMUP was added to a black 96-well half-area plate. About 30 µL SHP2 was added to each well to a final concentration of 0.05 nM immediately before starting the plate reader assay. Kinetic measurements (Ex 358/Em 355) were taken on a BioTek Neo2 plate reader every 25 s for 6 min.

### Co-immunopurification experiments

About $1 \times 10^6$ SHP2 KO HEK 293 cells were seeded in a 6-cm plate. The next day, cells were transfected using 2 µg of each plasmid (myc-SHP2, c-Src, and MPZL1 or FLAG-ATPAF1), and 18 µg PEI in 600 µL DMEM. The transfection medium was refreshed after 16 h and replaced with complete medium. After 48 h, cells were harvested by scraping in PBS. Cells were washed three times in 1 mL PBS, and lysed in 100 µL lysis buffer (20 mM Tris-HCl, pH 8.0, 137 mM NaCl, 2 mM EDTA, 10% glycerol, and 0.5% NP-40 + protease inhibitors + phosphatase inhibitors) for 30 min while rotating at 4 °C. Cells were spun at 17.7 rpm for 15 min at 4 °C. Supernatant was transferred to a clean Eppendorf tube and stored at −20 °C.

Protein concentration was determined using a BCA assay, and absorbance was measured at 562 nm using a BioTek Synergy Neo2 multi-mode reader. 300 µg of protein in a total volume of 380 µL was incubated overnight while rotating at 4 °C with 30 µL of magnetic Myc-beads, for MPZL1 co-IP by SHP2. ATPAF1 samples were incubated with FLAG-beads while rotating for 30 min at room temperature to co-IP SHP2. The beads were washed 3 times on a magnetic rack using 1 mL lysis buffer. Then, 65 uL 1x Laemmli buffer was added, and beads were boiled at 100 °C for 8 min. For whole cell lysates, 15 µg protein was loaded onto a gel. For IP samples, 15 µL of boiled supernatant was used. Gel was transferred onto a nitrocellulose membrane using TurboBlot (Bio-Rad), and the membrane was blocked using 5% bovine serum albumin (BSA) in Tris-buffered saline (TBS) for 1 h at room temperature. Membranes were rinsed with TBS with 0.1% Tween-20 (TBST) and incubated with primary antibodies in TBST + 5% BSA overnight at 4 °C (Src 1:1000, β-actin 1:5000, Myc 1:5000, FLAG 1:5000, PD-1 1:1000, p-Tyr 1:2000). Co-immunopurification of the protein of interest was detected using an α-FLAG antibody. Membranes were washed and incubated with secondary antibodies (IRDye 680 and 800). Blots were imaged on a LiCor Odyssey. Band intensities were quantified using Image Studio Lite (Version 5.2) using the Median Background setting. IP intensities were divided by corresponding intensities in the total cell lysate. $P$ values were calculated in GraphPad Prism (Version 10.1.0) using a paired, two-sided $T$-test.

### Ras dephosphorylation assay (Rassay)

About $0.8 \times 10^6$ SHP2$^{KO}$ HEK 293 cells were seeded in a 6 cm plate. The next day, cells were transfected with Ras, Ras and Src, or Ras, Src and SHP2; to a total of 3 µg in 300 µL DMEM with 9 µg of polyethyleneimine. The transfection medium was refreshed the next morning and replaced with warm DMEM with 10% FBS.

Approximately 36 h after transfection, the cells were harvested by scraping and washed three times in 1 mL cold PBS. Cells were lysed in 150 µL lysis buffer (20 mM Tris, pH 8.0, 137 mM NaCl, 2 mM EDTA, 10% glycerol, 0.5% NP-40, with freshly added phosphatase- and protease inhibitors) for 25 min on ice. Lysates were spun down for 15 min at $17,000 \times g$ in a 4 °C tabletop centrifuge. Supernatant was used in a BCA assay to determine protein concentration. About 80 µg of protein was used in an immuno-precipitation (IP) with 5 µg of packed Pierce anti-HA magnetic beads (Fisher, #88836) in a total volume of 350 µL lysis buffer. Samples were incubated at 4 °C overnight while rotating. The next morning, beads were washed three times on a magnetic rack using 1 mL of lysis buffer, and resuspended in 65 µL of 1x Laemmli buffer. All samples were boiled at 100 °C for 8 min, and 15 µg of total protein (total cell lysate) or 15 µL of each sample (immuno-precipitation) was loaded onto a 12% acrylamide gel. Proteins were transferred to a 0.45-µm nitrocellulose membrane using the StandardSD protocol on the Bio-Rad Trans-Blot Turbo. Membranes were blocked for 1 h at room temperature using 5% BSA in TBS. Primary antibodies were stained for 2 h at room temperature in 5% BSA in TBST. Membranes were washed three times in 5 mL TBST for 5 min each. Secondary antibodies were incubated in 5% BSA in TBST for 1 h at room temperature. Membranes were imaged on a LiCor Odyssey, and bands were quantified using Image Studio.

### Preparing endogenous SHP2 samples for imaging

HEK 293 cells and SHP2 knock-out HEK 293 cells were seeded in a black, glass-bottom plate. HEK 293 cells were stimulated with 100 ng/mL EGF for 10 min, or left unstimulated. SHP2 knock-out HEK 293 cells, which were used as a negative antibody control, were left unstimulated. Then, half the culture media was replaced with 4% paraformaldehyde (PFA) + 4% sucrose in PBS, pre-warmed to 37 °C. This was incubated for 1 min at room temperature, then the entire well-volume was replaced with 4% PFA + 4% sucrose in PBS. The plate was incubated for 10 min at 37 ˚C. Wells were rinsed twice with 1 mL PBS and permeabilized with 0.1% Triton X-100 in PBS for 10 min on ice. Wells were rinsed twice with 1 mL PBS. Blocking was performed by the addition of 5% BSA in PBS, which was incubated for 1 h at room temperature. Wells were rinsed twice with 1 mL PBS, after which primary antibodies were applied in PBS + 0.05% Tween-20 for 1 h at room temperature. After incubation, wells were rinsed three times with 1 mL PBS. Secondary antibodies were applied in PBS for 30 min at room temperature in the dark. Finally, wells were rinsed three times with 1 mL PBS, after which DAPI was added, and cells were imaged. If restaining was necessary, fluorophores were bleached by the addition of LiBH$_4$ (0.5 mg/mL) for 30 min at room temperature, after which cells were reblocked and restained.

### Preparing SHP2-transfected samples for imaging

SHP2 knock-out HEK 293 cells were seeded in a black, glass-bottom plate. The next day, cells were transfected with 0.5 µg of myc-SHP2 DNA in empty DMEM overnight. Transfection medium was refreshed with DMEM + 10% FBS the next morning. The following day, cells treated with TNO155 were incubated in 1 µM TNO155 for 4 h before proceeding with fixation and staining. Cells were stimulated with 100 ng/mL EGF for 10 min, or left unstimulated. Fixation and staining followed the same protocol as for the endogenous SHP2 samples.

### Acquisition and analysis of fluorescence images

All images were acquired under oil immersion 60x magnification (Nikon, MRD71670) using a confocal spinning disk microscope (Andor Dragonfly) coupled to a Nikon Ti-2 inverted epifluorescence microscope with automated stage control, Nikon Perfect Focus System and a Zyla PLUS 4.2-megapixel USB3 camera. Illumination was done with 100 mW 405 nm, 50 mW 488 nm, 50 mW 561 nm, 140 mW 640 nm, and 100 mW 785 nm solid-state lasers. All hardware was controlled using Andor Fusion software. Lasers, laser powers, exposure times, objectives and experiment-specific acquisition parameters are 100% power, 100 ms exposure for all the images. Images were acquired with 11 z-slices at 2.0-μm intervals (Total scan size 20 μm). Briefly, to analyze SHP2 localization, the images were converted to zarr format. The images were then stitched using ashlar (https://github.com/labsyspharm/ashlar). Images were then preprocessed to replace any 0-padded values from stitching with background values. For images with two rounds of imaging (i.e., the endogenous samples) we performed registration using Affine transformation. Once we generated the final image stack, cell masks were obtained using the SHP2 image, and segmentation was performed using Cellpose3 (Stringer and Pachitariu, 2025). Finally, we set up signal cut-offs for both the SHP2 signal and mitochondrial signal (as measured by Tom20). We then calculated the number of SHP2-positive pixels co-localizing with mitochondria-positive pixels per cell (code available with Fig. EV4 Source Data). We further calculated the average SHP2 intensity per SHP2-positive cell. Visualization of images was done using ImageJ/Fiji. For SHP2 and Tom20 channels, a binary mask was created using default settings. Overlapping pixels were identified using the Image Calculator's "AND" operator.

### Mitochondrial isolations

About $10 \times 10^6$ SHP2$^{KO}$ HEK 293 cells were seeded in a 15-cm plate. Cells were transfected with 25 μg DNA in DMEM with 75 μg of PEI. Transfection medium was aspirated and replaced with DMEM + 10% FBS. Thirty-six hours after transfection, cells were harvested by scraping and washed three times in 1 mL PBS. Cell pellets were processed using a Mitochondria/Cytosol Fractionation Kit (Fisher, #89874) according to the manufacturer's instructions. Mitochondrial pellet was lysed in 30–50 μL lysis buffer (20 mM Tris-HCl, pH 8.0, 137 mM NaCl, 10% glycerol, 1% Nonidet P-40 (NP-40), 2 mM EDTA, with protease inhibitors and phosphatase inhibitors added fresh). Protein concentration of cytosolic and mitochondrial fractions was determined using a BCA assay, and samples were further analyzed by Western blot.

### Quantification of reactive oxygen species

Reactive oxygen species (ROS) were profiled using MitoSOX™ Mitochondrial Superoxide Indicators for live-cell imaging (Invitrogen, # M36006) according to the manufacturer's instructions. Briefly, $0.8 \times 10^6$ SHP2$^{KO}$ HEK 293 cells were seeded in a 6 cm plate, and transfected the next day with 5 μg of SHP2 DNA in empty DMEM. Transfection medium was replaced the following morning with DMEM + 10% fetal bovine serum (FBS). Cells were harvested 36 h post-transfection by scraping in PBS, and washed two more times in 1 mL PBS. MitoSox Green reagent was diluted to 1 mM in dimethylformamide (DMF), and further diluted to a working concentration of 2.5 μM in PBS. Cells were resuspended in 200 μL of MitoSox Green working solution for 1 h at 37 °C; + ATP samples were

incubated with 5 mM ATP and MitoSox Green simultaneously. Cells were washed three times in 500 μL PBS and analyzed on an Attune NxT using 488 nm excitation and emission filter 530/30.

### Differential scanning fluorimetry

Purified protein stocks were thawed and diluted in DSF buffer (20 mM HEPES, pH 7.5, 50 mM NaCl, and 0.4% DMSO). About 19 μL of buffer was added to a MicroAmp Fast Optical 96-well Reaction plate (Applied Biosystems, # 4346906). About 1 μL of 500x SYPRO Orange Protein Gel Stain (Thermo Fisher, catalog no. S-6650) was added to a final protein concentration of 10 μM and 25x SYPRO Orange. Melting curves were performed in an Applied Biosystems Step-One Plus RT-PCR thermocycler. Temperature measurements started at 15 °C, and the temperature was raised by 0.5 °C every minute with continuous measurements of fluorescence (excitation: 472 nm; emission: 570 nm). Raw fluorescence values along with corresponding temperatures were analyzed using DSFworld, and $T_m$ values were calculated using dRFU (Wu et al, 2024).

### Analysis of solvent exposure in molecular dynamics simulations

Molecular dynamics simulations were performed and reported previously (Jiang et al, 2025). In total, twelve 2.5 μs simulations were analyzed: three simulations of SHP2$^{WT}$ and three simulations of SHP2$^{E76K}$, starting from the conformation seen in PDB code 4DGP, and three simulations of SHP2$^{WT}$ and three simulations of SHP2E76K, starting from the conformation seen in PDB code 6CRF. MD trajectories were compiled from the raw data using the CPPTRAJ module of AmberTools22 (Case et al, 2022). Structures were extracted from the trajectories in 10 ns increments for analysis. Solvent-accessible surface area (SASA) of each residue was calculated at each extracted timepoint using the PDB module in Biopython (Cock et al, 2009). SASA values for all simulations starting from 4DGP or 6CRF were combined to generate the distributions shown in Fig. 6D.

### AlphaFold 3 modeling of the SHP2-PPIF interaction

Full-length human SHP2 (Uniprot Q06124) and full-length (Uniprot P30405) were modeled as a 1:1 complex using AlphaFold 3 via the AlphaFold Server (https://alphafoldserver.com/) (Abramson et al, 2024). The default parameters were used, with no post-translational modifications on either chain. Models were visualized using PyMOL (Schrödinger and Delano, 2020).

## Data availability

There are no restrictions on the use of any data generated in this study. The datasets produced in this study are available in the following databases: Affinity-purification mass spectrometry data: PRIDE PXD067419 (https://www.ebi.ac.uk/pride/archive/projects/PXD067419). TurboID mass spectrometry data: PRIDE PXD067905 (https://www.ebi.ac.uk/pride/archive/projects/PXD067905). Total proteome mass spectrometry data: PRIDE PXD069201 (https://www.ebi.ac.uk/pride/archive/projects/PXD069201). Microscopy data: Bio-Image Archive S-BIAD2267 (https://doi.org/10.6019/S-BIAD2267). SH2 profiling deep sequencing data: Dryad (https://doi.org/10.5061/dryad.wstqjq2xv). AlphaFold 3 models: Zenodo (https://doi.org/10.5281/zenodo.17020308).

The source data of this paper are collected in the following database record: biostudies:S-SCDT-10_1038-S44319-025-00674-4.

## Peer review information

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

## Acknowledgements

We would like to thank the members of the Shah and Jovanovic labs for their scientific insights and helpful discussions. This research was funded by NIH National Institute of General Medical Sciences grant R35GM138014 to NHS. MJ was supported by NIH National Institute of General Medical Sciences grant R35GM152258. JTG was funded by NIH National Cancer Institute grant DP2CA281605. LCT was supported by an National Science Foundation Graduate Research Fellowship (award # 2036197).

## Author contributions

**Anne E van Vlimmeren**: Conceptualization; Data curation; Formal analysis; Investigation; Visualization; Methodology; Writing—original draft; Writing—review and editing. **Lauren C Tang**: Conceptualization; Formal analysis; Methodology; Writing—review and editing. **Ziyuan Jiang**: Investigation; Writing—review and editing. **Abhishek Iyer**: Formal analysis; Investigation; Methodology; Writing—review and editing. **Rashmi Voleti**: Investigation; Methodology; Writing—review and editing. **Konstantin Krismer**: Formal analysis; Writing—review and editing. **Jellert T Gaublomme**: Resources; Supervision; Funding acquisition; Writing—review and editing. **Marko Jovanovic**: Conceptualization; Resources; Supervision; Funding acquisition; Writing—review and editing. **Neel H Shah**: Conceptualization; Resources; Data curation; Formal analysis; Supervision; Funding acquisition; Visualization; Writing—original draft; Project administration; Writing—review and editing.

Source data underlying figure panels in this paper may have individual authorship assigned. Where available, figure panel/source data authorship is listed in the following database record: biostudies:S-SCDT-10_1038-S44319-025-00674-4.

## Disclosure and competing interests statement

The authors declare no competing interests.

# Expanded View Figures

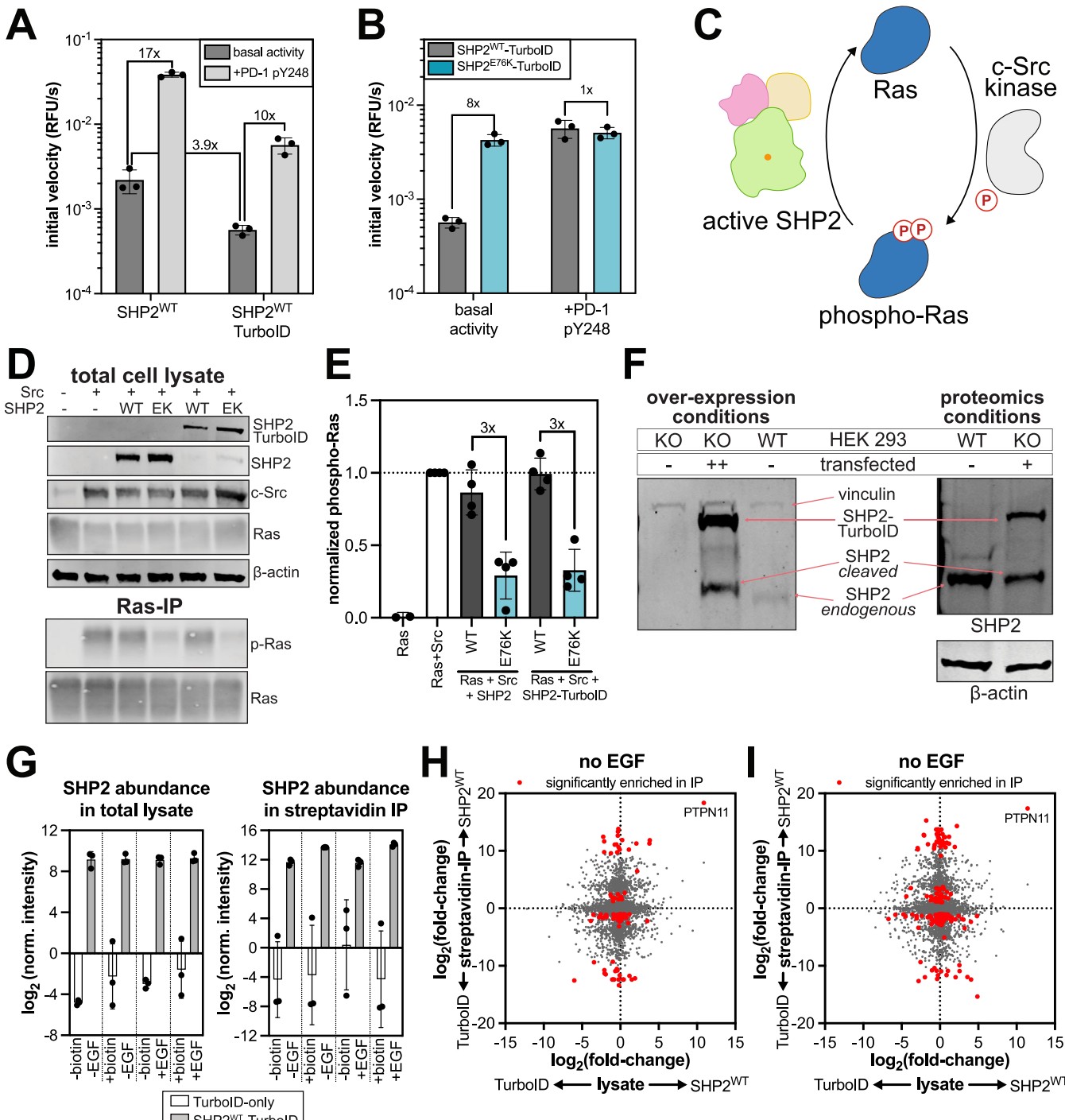

**Figure EV1. Quality control and validation of the SHP2-TurboID system.**

(A) In vitro activity measurements showing that SHP2-TurboID can be activated by a phosphopeptide (PD-1 pY248) ($n = 3$). Bar heights indicate the mean, and error bars indicate standard deviation. (B) In vitro activity measurements showing that E76K hyperactivation is preserved in a TurboID-fusion context, and only SHP2$^{WT}$-TurboID, but not SHP2$^{E76K}$-TurboID can be further activated by PD-1 pY248 ($n = 3$). Bar heights indicate the mean, and error bars indicate standard deviation. (C) Schematic of the Ras dephosphorylation assay in HEK 293 cells. (D) Representative western blots showing dephosphorylation of N-Ras by SHP2-TurboID proteins. WT = SHP2$^{WT}$ EK = SHP2$^{E76K}$. (E) Quantification of Ras dephosphorylation assays ($n = 4$). Bar heights indicate the mean, and error bars indicate standard deviation. (F) The left blot compares SHP2 levels in WT and SHP2$^{KO}$ HEK 293 cells, and shows over-expression of SHP2$^{WT}$-TurboID in SHP2$^{KO}$ cells, highlighting partial cleavage of SHP2-TurboID. The right blot compares endogenous SHP levels in WT HEK 293 cells with SHP2$^{WT}$-TurboID levels in SHP2$^{KO}$ HEK 293 cells, transfected under the same conditions used for proteomics experiments. (G) Further confirmation of SHP2$^{KO}$ cells used for proteomics experiments, comparing SHP2 levels detected by mass spectrometry in negative control TurboID-only samples and SHP2$^{WT}$-TurboID samples. Bar heights indicate the mean, and error bars indicate standard deviation. (H) Fold-change between SHP2$^{WT}$-TurboID and TurboID-only for total lysates, measuring protein abundance (x-axis), and streptavidin-IP, measuring proximity labeling (y-axis), in unstimulated cells. (I) Same as (H), but for cells stimulated with 100 ng/mL EGF ($n = 3$ for total lysate and TurboID datasets). Source data are available online for this figure.

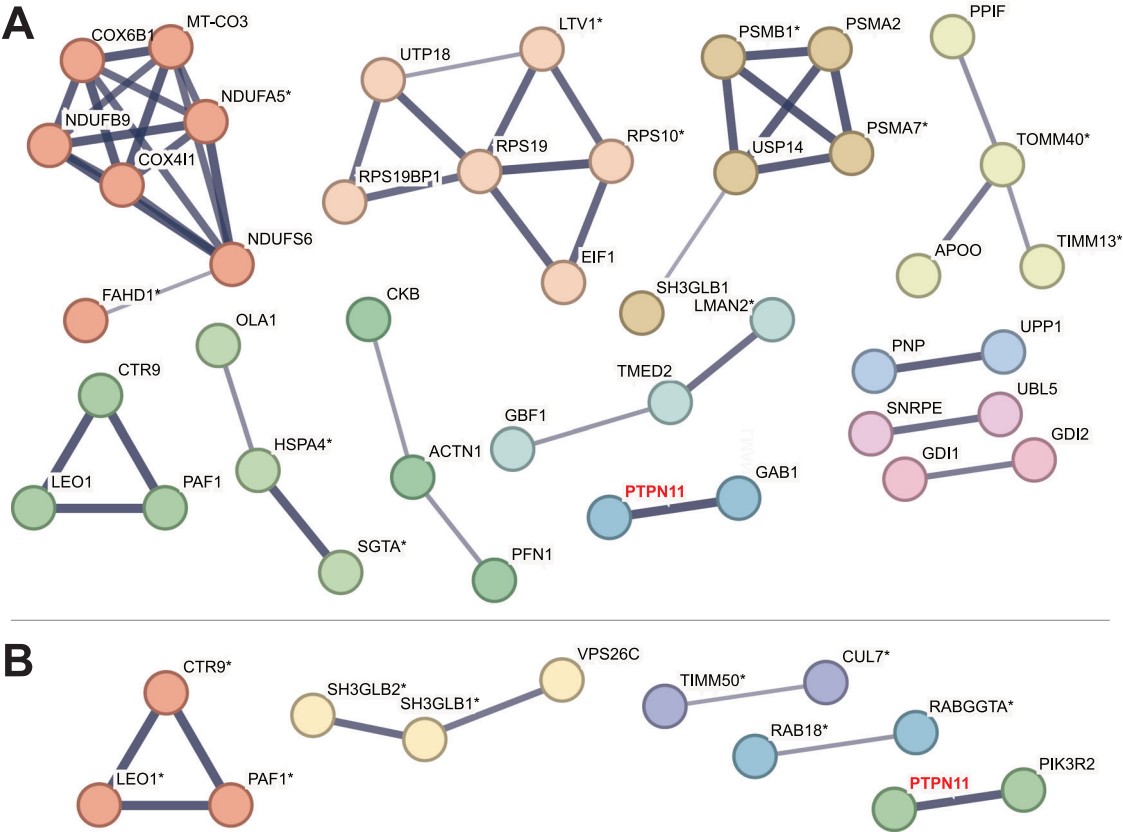

**Figure EV2. STRING interaction networks for SHP2WT-TurboID hits.**

(A) STRING interaction network for proteins enriched by SHP2WT-TurboID over the TurboID control by at least twofold with a p value <0.1 (heteroscedastic, unpaired t-test), in the absence of EGF stimulation. (B) Same as in (A) but with EGF stimulation. Proteins with a p value between 0.05 and 0.1 are marked with an asterisk. All other proteins have a p value <0.05. For both panels, solid lines between proteins indicate a known physical interaction, as documented in the STRING database. Only proteins that have a physical interaction with at least one other protein in our interactomes and have an edge confidence score of at least 0.4 are shown. Edge thickness represents edge confidence: thin = 0.4, medium = 0.7, thick = 0.9. Clusters were identified by Markov Clustering (MCL). Source data are available online for this figure.

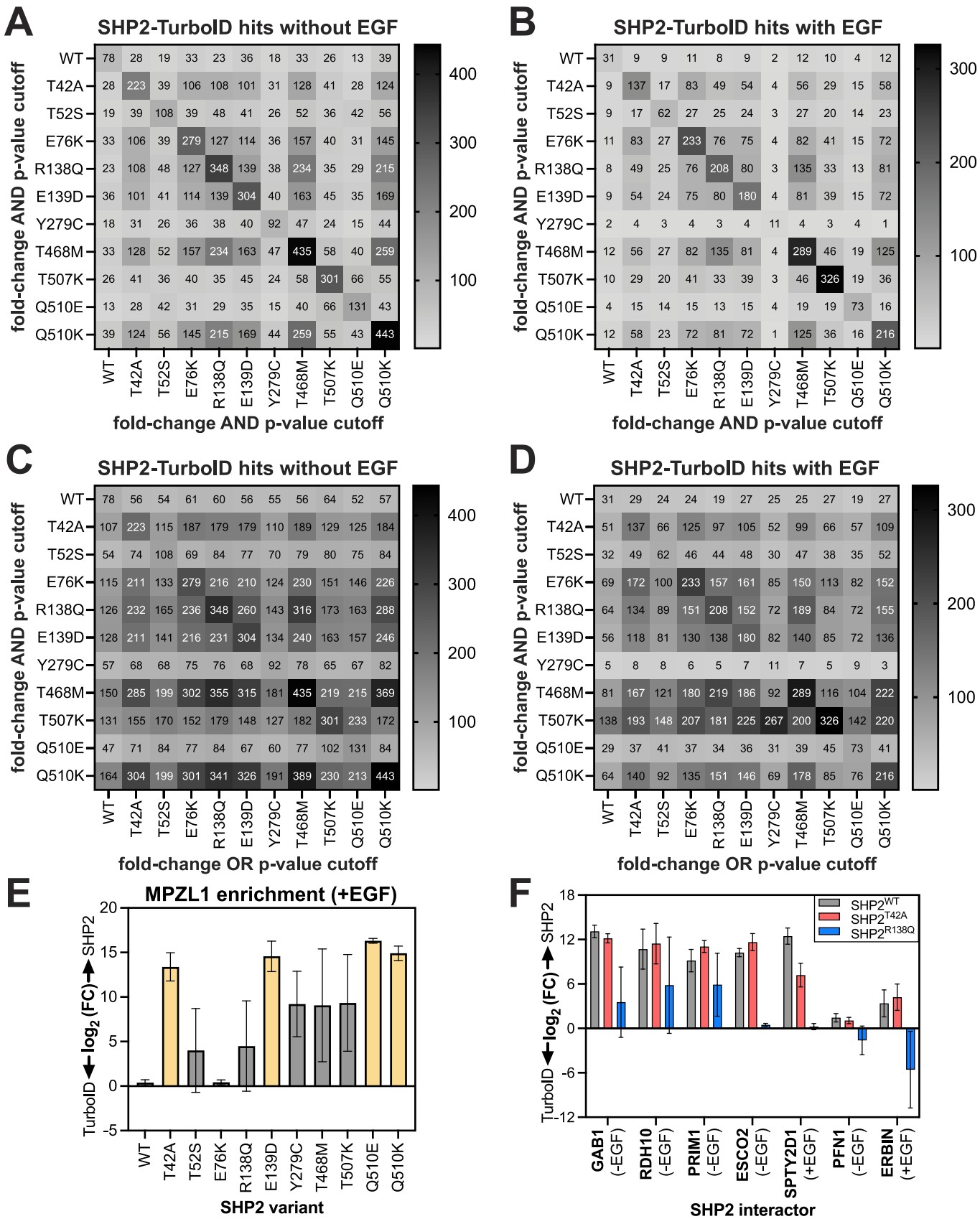

◀ **Figure EV3. Mutation- and EGF-dependent changes in SHP2-TurboID proximity labeling.**

(A) Stringent pairwise overlap in hits for all SHP2 variants relative to the TurboID control in unstimulated samples. Hits are defined as proteins with >2-fold enrichment and a p value <0.05 (heteroscedastic, unpaired t-test), and overlap indicates the number of proteins that meet these criteria for both SHP2 variants in the comparison. (B) Same as in (A) but with EGF stimulation. (C) Relaxed pairwise overlap in hits for all SHP2 variants relative to TurboID control in unstimulated samples. For SHP2 variants on the y-axis, a hit was defined as proteins with both >2-fold enrichment and a p value <0.05. For SHP2 variants on the x-axis, a hit was defined as proteins with either >2-fold enrichment or a p value <0.05. (D) Same as (C), but with EGF stimulation. Note that in panels (A–D), the smaller numbers for SHP2$^{Y279C}$ reflect the use of only two replicates for this mutant, due to an instrument error and sample loss. (E) Enrichment of MPZL1 in our dataset for all SHP2 variants relative to the TurboID-only control (+EGF), highlighting significant proximity labeling with some mutants ($n = 3$ biological replicates for all samples except SHP2$^{Y279C}$, which had two biological replicates). Yellow indicates a significant difference from the TurboID-only control (fold-change >2, p value <0.05 from a heteroscedastic, unpaired t-test): $p = 0.005656$ (T42A), $p = 0.00580$ (E139D), $p = 0.00003$ (Q510E), and $p = 0.00049$ (Q510K). (F) Enrichment of proteins with predicted C-SH2 binding sites (top ~30th percentile) by SHP2$^{WT}$, SHP2$^{T42A}$, and SHP2$^{R138Q}$, showing loss of signal for C-SH2-dead SHP2$^{R138Q}$ ($n = 3$ biological replicates). In panels (E, F), bar heights indicate the mean, and error bars indicate standard deviation. Source data are available online for this figure.

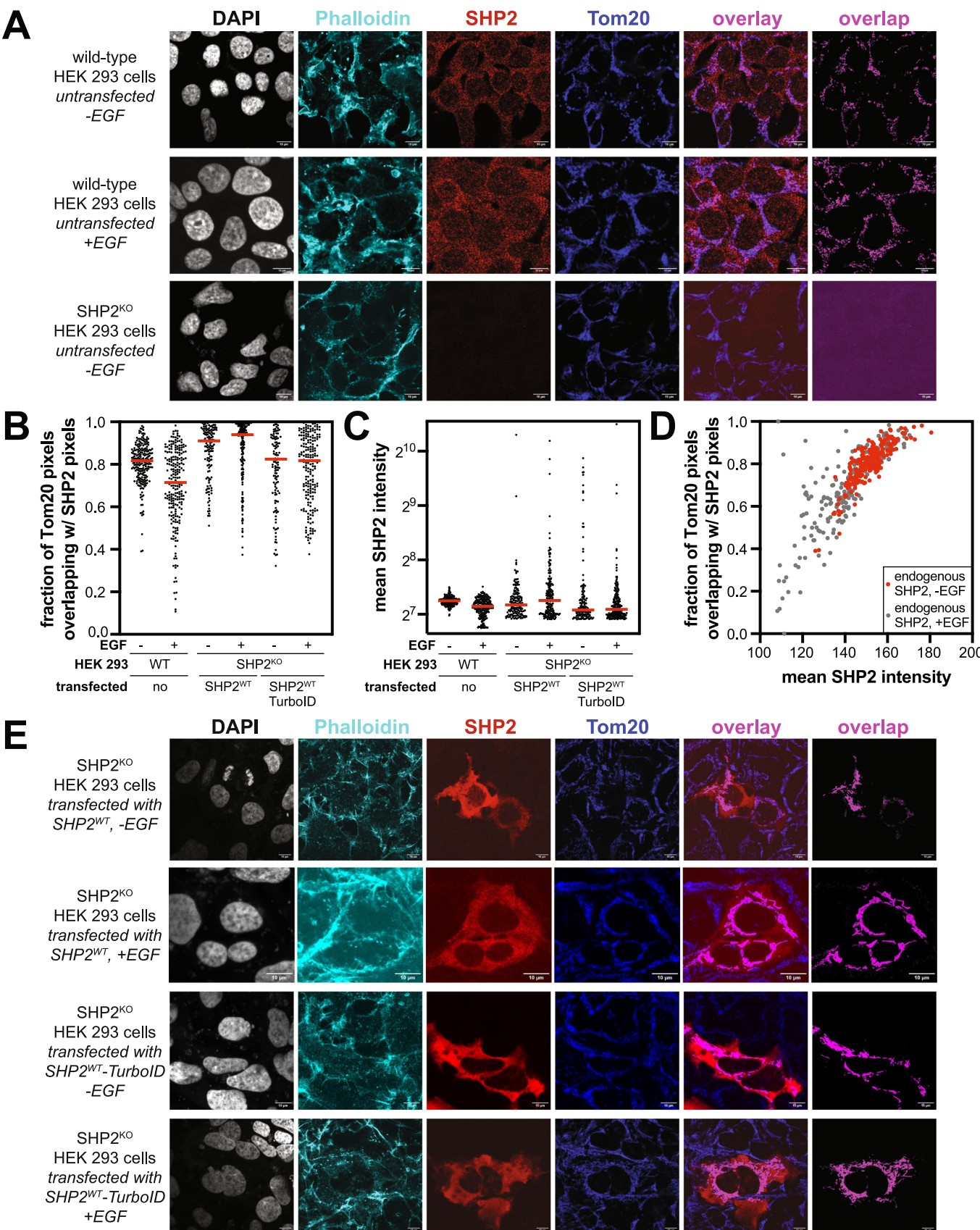

◀  **Figure EV4.  Analysis of SHP2 localization by fluorescence microscopy.**

(A) Representative images of endogenous SHP2 localization by confocal fluorescence microscopy showing co-localization of SHP2 with mitochondria in wild-type HEK 293 cells, and absence of SHP2 in SHP2$^{KO}$ HEK 293 cells. (B) Fraction of Tom20-positive pixels that overlap with SHP2-positive pixels. (C) Mean fluorescence intensity of all SHP2-positive pixels in cells used to assess the mitochondrial overlap in panel (B). In panels (B, C), the red lines are median values, $N = 129$–234 cells, and the SHP2$^{KO}$ cells $-/+$ EGF have a statistically significant difference in both co-localization and mean SHP2 intensity ($p < 0.0001$, Welch's unpaired $t$-test). (D) Correlation between mean SHP2 intensity in cells and SHP2 pixel overlap with Tom20. (E) Representative images of SHP2 localization in SHP2$^{KO}$ HEK 293 cells transfected with different SHP2 constructs, unstimulated or stimulated with EGF. Source data are available online for this figure.

                                        

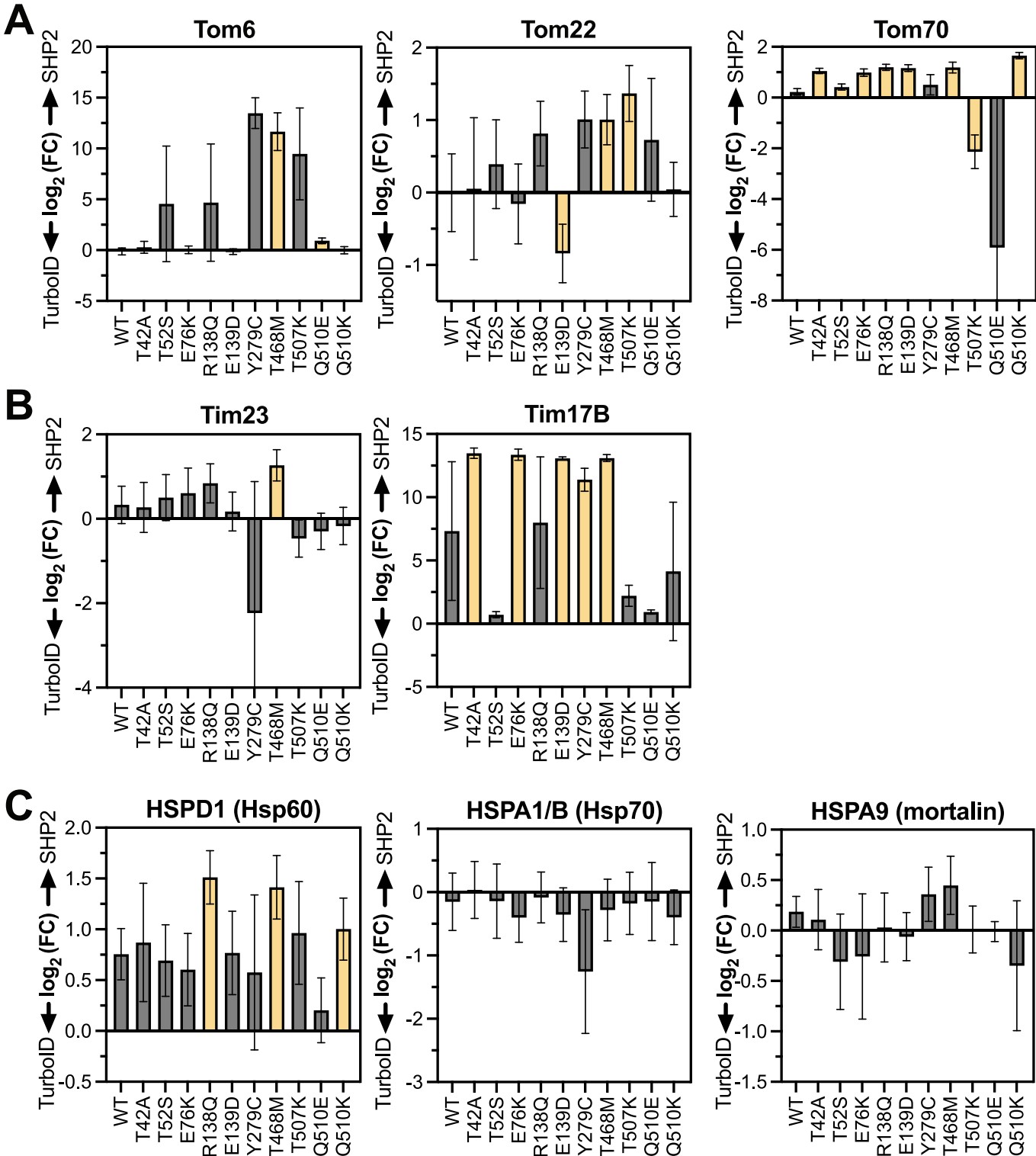

◄ **Figure EV5. Mutant-specific labeling of mitochondrial import proteins and chaperones.**

In all panels, yellow bars indicate a significant difference from the TurboID-only control (fold-change >2, $p$ value <0.05 from a heteroscedastic, unpaired $t$-test). Bar heights indicate the mean, and error bars indicate standard deviation. (**A**) Enrichment or depletion of select Tom proteins in our dataset for all SHP2 variants relative to the TurboID-only control. For Tom6, $p = 0.01094$ (T468M) and $p = 0.02227$ (Q510E). For Tom22, $p = 0.04200$ (E139D), $p = 0.01760$ (T468M), and $p = 0.00753$ (T507K). For Tom70, $p = 0.00051$ (T42A), $p = 0.00860$ (T52S), $p = 0.00092$ (E76K), $p = 0.00017$ (R138Q), $p = 0.00025$ (E139D), $p = 0.00537$ (T468M), $p = 0.04238$ (T507K), and $p = 5.06401 \times 10^{-5}$ (Q510K). (**B**) Same as (**A**), but for two components of the Tim23 complex in our dataset. For Tim23, $p = 0.02098$ (T468M). For Tim17B, $p = 0.00019$ (T42A), $p = 0.00026$ (E76K), $p = 1.28078 \times 10^{-5}$ (E139D), $p = 0.04912$ (Y279C), $p = 1.9238 \times 10^{-5}$ (T468M). (**C**) Same as (**A**) but for the mitochondrial chaperonin HSPD1/Hsp60, HSPA1A/HSPA1B (cytosolic Hsp70), and HSPA9/mortalin (mitochondrial Hsp70). For HSPD1/Hsp60, $p = 0.00509$ (R138Q), $p = 0.00339$ (T468M), and $p = 0.01100$ (Q510K). Source data are available online for this figure.

