## [Peer Review File · EMBO Reports]

Proximity-labeling proteomics reveals remodeled interactomes and altered localization of pathogenic SHP2 variants

Anne van Vlimmeren, Lauren Tang, Ziyuan Jiang, Abhishek Iyer, Rashmi Voleti, Konstantin Krismer, Jellert Gaublomme, Marko Jovanovic, and Neel Shah

Corresponding author(s): Neel Shah (neel.shah@columbia.edu)

Review Timeline:

Submission Date:	3rd Apr 25
Editorial Decision:	6th May 25
Revision Received:	11th Sep 25
Editorial Decision:	5th Nov 25
Revision Received:	12th Nov 25
Accepted:	1st Dec 25

Editor: Deniz Senyilmaz Tiebe / Martina Rembold

Transaction Report:

Dear Dr. Shah,

Thank you for transferring your manuscript to EMBO Reports, which was now seen by three referees, whose reports are copied below.

Referees express interest in the provided resource reporting the interactome of wild type and pathogenic mutants of SHP2 utilizing proximity-labeling mediated proteomics. However, they also raise some concerns that need to be addressed to consider publication here. In particular, referees find that additional evidence supporting the proposed mitochondrial localization of SHP2 variants (all referees). Referee Moreover, referees #1 and #3 find that the proximity labelling approach needs further validation. Please contact me if you have questions or comments regarding the revision for further discussion (also by video chat).

Given these recommendations, we would like to invite you to revise your manuscript with the understanding that the referee concerns (as in their reports) must be fully addressed and their suggestions taken on board. Please address all referee concerns in a complete point-by-point response. Acceptance of the manuscript will depend on a positive outcome of a second round of review. It is EMBO reports policy to allow a single round of major experimental revision only and acceptance or rejection of the manuscript will therefore depend on the completeness of your responses included in the next, final version of the manuscript.

We realize that it is difficult to revise to a specific deadline. In the interest of protecting the conceptual advance provided by the work, we recommend a revision within 3 months. Please discuss the revision progress ahead of this time with me if you require more time to complete the revisions, or if you have questions or comments regarding the revision (also by video chat).

1. A data availability section providing access to data deposited in public databases is missing (where applicable).
2. Your manuscript contains statistics and error bars based on $n=2$. Please use scatter plots in these cases.

You can submit the revision either as a Scientific Report or as a Research Article. For Scientific Reports, the revised manuscript can contain up to 5 main figures and 5 Expanded View figures, and it should not exceed 27000 characters. If the revision leads to a manuscript with more than 5 main figures it will be published as a Research Article. In this case the Results and Discussion section should be separate. If a Scientific Report is submitted, these sections have to be combined. This will help to shorten the manuscript text by eliminating some redundancy that is inevitable when discussing the same experiments twice. In either case, all materials and methods should be included in the main manuscript file.

4) a .docx formatted letter INCLUDING the reviewers' reports and your detailed point-by-point responses to their comments. As part of the EMBO publication's Transparent Editorial Process, EMBO reports publishes online a Review Process File (RPF) to accompany accepted manuscripts. This File will be published in conjunction with your paper and will include the referee reports, your point-by-point response and all pertinent correspondence relating to the manuscript.

<https://www.embopress.org/page/journal/14693178/authorguide#transparentprocess>

5) a complete author checklist, which you can download from our author guidelines <https://www.embopress.org/page/journal/14693178/authorguide>. Please insert information in the checklist that is also reflected in the manuscript. The completed author checklist will also be part of the RPF.

6) Please note that all corresponding authors are required to supply an ORCID ID for their name upon submission of a revised manuscript (<<https://orcid.org/>>). Please find instructions on how to link your ORCID ID to your account in our manuscript tracking system in our Author guidelines <<https://www.embopress.org/page/journal/14693178/authorguide#authorshipguidelines>>

Additional information on source data and instruction on how to label the files are available: <https://www.embopress.org/page/journal/14693178/authorguide#sourcedata>

9) Our journal encourages inclusion of *data citations in the reference list* to directly cite datasets that were re-used and obtained from public databases. Data citations in the article text are distinct from normal bibliographical citations and should directly link to the database records from which the data can be accessed. In the main text, data citations are formatted as follows: "Data ref: Smith et al, 2001" or "Data ref: NCBI Sequence Read Archive PRJNA342805, 2017". In the Reference list, data citations must be labeled with "[DATASET]". A data reference must provide the database name, accession number/identifiers and a resolvable link to the landing page from which the data can be accessed at the end of the reference. Further instructions are available at <http://www.embopress.org/page/journal/14693178/authorguide#referencesformat>

- the name of the statistical test used to generate error bars and P values,
- the number (n) of independent experiments (please specify technical or biological replicates) underlying each data point,
- the nature of the bars and error bars (s.d., s.e.m.),
- If the data are obtained from n Program fragment delivered error ``Can't locate object method "less" via package "than" (perhaps you forgot to load "than"?) at //ejpvfs23/sites23b/embo/www/letters/embo_decision_revise_and_review.txt line 56.' 2, use scatter blots showing the individual data points.

12) Please also note our reference format:

13) All Materials and Methods need to be described in the main text using our 'Structured Methods' format, which is required for all research articles. According to this format, the Methods section includes a Reagents and Tools Table (listing key reagents, experimental models, software and relevant equipment and including their sources and relevant identifiers) followed by a Methods and Protocols section describing the methods using a step-by-step protocol format. The aim is to facilitate adoption of the methodologies across labs. More information on how to adhere to this format as well as a downloadable template (.docx) for the Reagents and Tools Table can be found in our author guidelines:

I look forward to seeing a revised version of your manuscript when it is ready. Please let me know if you have questions or comments regarding the revision.

Kind regards,

Deniz Senyilmaz Tiebe

Deniz Senyilmaz Tiebe, PhD
Senior Scientific Editor
EMBO Reports

Referee #1:

Here, the authors have used TurboID-based proximity proteomics in living cells to catalogue the proteins that interact transiently with the PTPN11/SHP2 protein-tyrosine phosphatase, comparing the interactors observed for WT SHP2 with those observed with 10 different mutant SHP2 proteins, representative of both GOF and LOF mutations that are causally implicated in human disease. For this purpose, they fused TurboID to the C-terminus of SHP2, the activated E76K SHP2 mutant and 9 other mutants. These were transiently expressed in SHP2 KO HEK293 cells, which were exposed to biotin for 10 min, followed by MS-based proteomic analysis of biotinylated proteins, leading to identification of 78 core interactors, including several known SHP2 interactors, such as GAB1, as well as novel interactors, including the GDI1 and GDI2 Rab protein regulators, the UBL5 and SNRP spliceosome components, the PPIF mitochondrial PPIase, and the HER2/ERBB2 RTK. When the transfected HEK293 cells were stimulated with EGF, 31 SHP2 interacting proteins were identified, most of which were unique to the stimulated condition, with another 30 proteins showing reduced interaction upon EGF stimulation. Some of the EGF-dependent interactors have been reported to contain pTyr sites, and to determine whether these sites might be directly recognized by the N- or C-SH2 domain of SHP2, they exploited a degenerate pTyr peptide library approach to define the primary sequence specificities of the N- and C-terminal SH2 domains; this allowed them to identify 26 and 10 predicted tight-binding phosphosites in unstimulated and EGF-stimulated conditions, respectively, including the known SHP2 binding site on GAB1. Using this baseline information, they went on to determine the consequence of disease-causing mutations on SHP2's interaction with cellular proteins. In the absence of EGF stimulation, the SHP2 mutants shared between 13 and 39 hits with WT SHP2, although with more relaxed cutoff criteria between 52 and 61 shared proteins were noted. Out of an average of ~240 core interacting proteins, significant enrichment over the control was found for 4/11 SHP2 variants in unstimulated cells and 3/11 variants in EGF stimulated cells, but different SHP2 mutants also had unique interactors. By analysis of T42A N-SH2 and R138Q C-SH2 domain mutations, known to reduce pTyr-protein recognition, they showed that T42A SHP2 enriched MPZL1 and ACOX1, which in prior work they had predicted to be SHP2 SH2 domain interactors. Although the R138Q mutation weakens pTyr binding to the C-SH2 domain by removing the conserved FLVRES motif Arg, SHP2 R138Q enriched more proteins than WT, but these were less dependent on pTyr. Mitochondrial proteins, including NDUF and COX proteins, constituted a large part of the core SHP2 interactome, with 21 mitochondrial proteins enriched by WT SHP2; several SHP2 mutants showed enhanced interaction with mitochondrial proteins, including ETC proteins. They found increased association with Tom20, the MLS receptor on the mitochondrial outer membrane, for SHP2 mutants that exhibited increased mitochondrial protein interactions, and they proposed that these mutants have a partially unfolded "open" state that exposes a cryptic MLS, that promotes mitochondrial import; consistent with this, they found increased association with the Hsp40/DnaJ family mitochondrial chaperone proteins, which they propose might drive SHP2 mutant protein unfolding and thereby enhance its mitochondrial import. Finally, they tested the effects of the TNO155 (batoprotafib) allosteric SHP2 inhibitor, currently in clinical trials, on the SHP2 interactome. Consistent with its known mechanism of binding, incubation with TNO155 increased the melting temperatures of SHP2 WT and several SHP2 mutants, albeit to varying extents. They also found that TNO155 treatment reduced mitochondrial protein interactions and conclude that this was

due to bound TNO155 altering both SHP2 interactions and its localization by changing its overall conformation.

This is a worthwhile proximity proteomics survey of SHP2 PTP-interacting proteins in control and EGF-treated transfected HEK293 cells that has uncovered new pTyr-dependent and independent SHP2 interactors, whose patterns were changed upon EGF treatment. Concurrent analysis of the effects of 10 SHP2 disease mutations on the SHP2 interactome revealed some disease-specific interactors that might be of functional relevance in the disease they cause. In some ways, the most interesting findings are the mitochondrial SHP2 interactors, where some SHP2 mutants showed elevated association, which could be of significance in altered mitochondrial function in disease. In this regard, the authors imply that SHP2 is imported into mitochondria to exert mitochondrial functions through interactions with mitochondrial proteins, in some cases by pTyr/SH2 domain binding. However, they do not provide any compelling evidence that SHP2 is intramitochondrial or that the observed interactions occur in the mitochondrial matrix. Although in general the proximity proteomics data are convincing, there are some weaknesses, and, overall, these studies lack true functional validation of any of the new SHP2 interactions. Nevertheless, the WT and mutant SHP2 interactome databases should prove to be a valuable resource to the community for elucidating the roles of SHP2 mutations in human disease.

1. Why was the number of SHP2 binders decreased in the EGF-treated cells? The authors argue this could be due to the fact that SHP2 is diffusely localized in untreated cells, whereas it is translocated to major areas of Tyr phosphorylation in EGF-treated cells, such as the plasma membrane, but did not demonstrate such translocation experimentally. An alternative possibility is that this is due to negative feedback phosphorylation of SHP2, or possibly oxidation of the active site Cys in response to EGF-induced H₂O₂ production?
2. The authors went to great lengths to define primary pTyr sequence specificities of the SHP2 N- and C-SH2 domains and then use this information to predict which pTyr residues in interactors were likely to bind to one or other SHP2 SH2 domain, but did not validate any of the predicted pTyr/SH2 interactions by Tyr to Phe mutations. In this connection, it might be possible to carry out pTyr-enrichment proteomics on tryptic digests of the isolated biotinylated proteins that were in close proximity with SHP2 as a means of directly identifying pTyr sites in interactors that might be bound to the SHP2 SH2 domains (which would protect the interacting pTyr residues from dephosphorylation)?
3. The authors partially addressed the concern that overexpression of exogenously expressed SHP2-TurboID protein might result in spurious SHP2 interactions, by estimating that SHP2-TurboID overexpression was only ~4-fold over endogenous. However, because they used transient expression for these experiments, not every cell will have been transfected and in some transfected cells the concentration of SHP2-TurboID could have been significantly higher. As discussed below, it is also not clear that SHP2-TurboID has the same subcellular distribution as endogenous SHP2, and this needs to be ascertained by IF staining.
4. For the TurboID proximity proteomic experiments, the authors used a standard 10-minute biotin treatment of transfected cells initiated at the same time as EGF was added. This protocol will identify early EGFR-dependent SHP2 interactions, but additional SHP2 partner information could be obtained if they carried out a time course, particularly for the EGF-stimulation condition, e.g., by adding biotin at different times after EGF, which could reveal the kinetics of binding and dissociation of SHP2 interactors.
5. The authors explained the observed differences in the SHP2 interactome dependent on either EGF treatment or the individual SHP2 disease mutations in part by differences in the subcellular localization of SHP2, but the evidence that such localization differences exist is actually the case is rather minimal. There is one immunolocalization figure (Figure S6) where partial co-localization of endogenous SHP2 and mitochondria was observed, but this was done in a different cell line, i.e. U2-OS cells. Additional immunolocalization experiments with HEK293 cells are needed, particularly for SHP2-TurboID with and without EGF treatment to assess whether EGF treatment results in re-localization of SHP2-TurboID as they propose. The subcellular localizations of the mutant SHP2-TurboID proteins with and without EGF treatment also need to be determined to assess whether the SHP2 mutants exhibit increased mitochondrial association or nuclear localization, as is suggested.
6. The large number of mitochondrial proteins found in the SHP2 interactome data are consistent with a mitochondrial localization for SHP2. However, even though there have been some reports of SHP2 interaction with mitochondrial proteins, SHP2/PTPN11 is not listed as a mitochondrial protein in MitoCarta 3.0, the best validated mitochondrial protein database, indicating that SHP2 is either an extremely minor mitochondrial protein or a cell type specific visitor. The authors did show in Figure 5C that SHP2 was present in a crude mitochondrial fraction, but it is unclear what fraction of cellular SHP2 was present in the mitochondrial fraction, and this needs to be estimated. As indicated below, further analysis is needed to determine whether the SHP2 associated with mitochondria is in the mitochondrial matrix or is associated with the outer mitochondrial membrane.
7. Some of the HSP40/DnaJ family proteins do have reported pTyr sites, but these were all identified through HTP studies, and it is not clear whether their phosphorylation would regulate their chaperone activity or what tyrosine kinase would phosphorylate these sites - there is little convincing evidence the cytoplasmic tyrosine kinases get into mitochondria.
8. With further regard to the proposed SHP2 mitochondrial localization, the authors suggest that this is due to a cryptic mitochondrial localization signal (MLS) in SHP2, but did not identify/mutate this putative MLS to test this possibility. This raises

the important question of which SHP2-TurboID interactors are mitochondrially anchored on the outer membrane, as opposed to being truly intramitochondrial? Either protease protection experiments with intact purified mitochondria or IEM localization of SHP2/SHP2-TurboID is required. Alternatively, can mitochondrial SHP2 observed by GFP-tagging SHP2? Is it possible that association of SHP2 with a bona fide mitochondrial protein facilitates its import by acting as a carrier. How does the mitochondrial SHP2 interactome differ if a potent N-terminal MLS is appended to SHP2-TurboID?

9. With regard to the observed spliceosome SHP2 interactors, while there have been reports of nuclear SHP2, and Figure S6F shows a few nuclear puncta, both SHP2 itself (68 kDa) and the SHP2-TurboID fusion (~100 kDa = SHP2 68 kDa + TurboID 34 kDa) are too large to enter the nucleus passively. Is there an identified SHP2 NLS (or NES), and could the NLS be mutated to establish that the observed interactions occur within the nucleus?. Based on the N/C-SH2 consensus, are there relevant pTyr-containing protein targets for SHP2 in the nucleus, such as the spliceosome subunit proteins. The alternative to the interactions occurring with intranuclear SHP2 is that there is a population of spliceosome subunits in the cytoplasm that SHP2 could interact. Can a nuclear population of WT SHP2-TurboID be detected by GFP-tagging or IF staining and does this differ for the SHP2 mutants.

10. Have the authors ruled out that some of the observed SHP2 interactions occur post lysis, a potential problem with high activity TurboID -? The authors did use EDTA in the lysis buffer but not denaturing detergents?)

11. How many of the pTyr sites on SHP2 interactors can be dephosphorylated by SHP2, e.g. via catch and release proximity activity?

12. Despite the convincing proximity proteomics data, only a few interactions were validated by additional approaches, and in the end we do not learn how the observed SHP2 interactions might affect the functions of the target proteins or whether any of the novel/unique interactions with disease mutant SHP2s are functionally significant. Specifically, for pTyr-independent interactions, one would need to define the contact sites in SHP2 and the interacting protein, and show that mutation of key contact residues disrupts the TurboID labeling. Admittedly, this would be beyond the scope of this paper, but the authors could exploit AlphaFold 3 modeling to see if contact surfaces between SHP2 and interactors of interest could be identified.

Other points: 1. Page 5: The cited Gingras group's protein phosphatase interactome paper (ref. 35) did not study PTPN11, but BioGrid lists 342 unique protein interactors for WT SHP2, found by IP/MS and proximity labeling, and some comparison of the authors' interactomes with this publicly available dataset would be instructive.

2. Figure 2A: Reasonably the authors decided to use C-terminally fused TurboID to avoid interference with the N-SH2 autoinhibitory interaction, but did they actually test TurboID fused at the SHP2 N-terminus to see if this affected enzyme activity, or whether there was any difference in the repertoire of biotin-tagged proteins, perhaps due to the topology of binder interactions in this configuration?

3. Figure S2A/B: They showed that the pY248 PD-1 peptide-stimulated SHP2-TurboID fusion protein catalytic activity, but the initial velocity values on the y axis are plotted on a log scale, and it looks as though the fusion protein's catalytic activity was in fact significantly lower than that of WT. This deserves comment.

4. Figure 2E: It would be helpful if the N- and C-SH2 domain consensus were provided rather than the reader having to figure this out from the heat maps.

5. Figure 4C: Did the authors determine if the observed SHP2 interactions with MPZL1 and ACOX1 were dependent on these pTyr sites by making Tyr to Phe mutants.

6. Figure 5C: The mitochondrial fractionation experiment shows that there was an increase in mitochondrial association of the three of the four SHP2 mutants compared to WT SHP2, but it is not clear that the associated SHP2 was truly intramitochondrial, nor is it clear what fraction of the total cellular SHP2 is associated with mitochondria. Protease treatment of isolated mitochondria is commonly used to determine whether or not a protein is truly intramitochondrial.

7. Figure 5E: Here, they showed that expression of a subset of the SHP2 mutants enhanced formation of mitochondrial reactive oxygen species in an in vitro assay, but these effects do not appear to correlate with the increased mitochondrial protein interactions shown for these mutants in panel E. It is also unclear whether interaction with specific ETC proteins is involved in increased ROS production, or whether this effect requires SHP2 phosphatase activity.

8. Figure 5G: The authors imply that the observed coimmunoprecipitation of ATPAF1 and SHP2 involves the mitochondrial populations of ATPAF1 and SHP2 as opposed to cytoplasmic populations. All nuclearly encoded mitochondrial proteins are translated in the cytoplasm and then imported affording a chance for interaction with other cytoplasmic proteins.

9. Figure S6: The overall pattern of endogenous SHP2 staining in U2OS osteosarcoma cells is surprisingly punctate, and only a small fraction of the cytoplasmic puncta seem to correspond to mitochondria. What cellular structures do these other puncta

correspond to? Is this punctate staining pattern similar to that observed for SHP2 in other published studies, and is a similar pattern observed when different anti-SHP2 antibodies are used. To establish staining specificity, the proper experiment is demonstrate that all SHP2 staining is lost in SHP2 knockout U2-OS cells. While HEK293 cells are less suitable for IF staining experiments than the flat U2-OS cells, there are established protocols for doing this, and the authors need to demonstrate that the SHP2 staining patterns are similar in HE293 cells, including showing that SHP2 and SHP2-TurboID have similar staining patterns and also determining how EGF treatment affects SHP2 localization. As indicated above, what is really needed is direct evidence that SHP2 is present inside mitochondria.

10. Figure S6: There appears to be very little SHP2 staining in the nuclei of these U2-OS cells, but, what there is, is also punctate. This also needs discussion, and IF staining of HEK293 cells to determine if there is a nuclear population of SHP2-TurboID.

11. Figure 7E: The effects of the TNO155 allosteric inhibitor on the SHP2 interactome are interesting, but whether this is due to altered subcellular localization as the authors concluded is unclear and needs to be examined by IF staining studies.

12. There are some additional controls that the authors might consider: 1. The use of an EGFR kinase inhibitor to demonstrate that the observed effects of EGF on SHP2 interactions are dependent on EGFR-mediated phosphorylation. 2. The use of SHP2 FLVRES motif SH2 domains mutants to demonstrate that the interactions are due to pTyr-SH2 binding. 3. The use of a catalytically-dead SHP2 Asp to Ala mutant controls that might reveal interactions with pTyr substrates by trapping (did the authors compare their SHP2 interactomes with those reported for SHP2 substrate trapping mutants in ref. 98)? 4. The use of an outer mitochondrial membrane-anchored TurboID fusion protein to define which inner mitochondrial proteins can be biotinylated as they are imported into mitochondria.

Referee #2:

In this manuscript, the authors performed proximity-labeling proteomics, and mapped the interaction networks by wt and mutant SHP2 molecules. Furthermore, the authors also examined the SHP2 interactome in the presence of a specific allosteric inhibitor. It is interesting to observe the elevated mitochondrial location of some SHP2 mutants with potential impact on mitochondrial functions.

This article provides an excellent resource for further exploration of biochemical mechanisms of SHP2 functions in cell signaling in health and diseases.

One minor concern is that, as the authors admitted, the expression levels of some exogenous proteins were 4 folds higher than the endogenous protein, which could compound the results of the interactomes.

Related to this concern was the data on mitochondrial location of some mutants, whether this was due to over-expression? The authors need to discuss the possible caveats.

Referee #3:

The Authors used proximity labeling proteomics to study the localization and protein-protein interactions of wild-type SHP2 and 10 SHP2 variants. Some of these variants have clinical implications. Taking advantage of TNO155, an allosteric inhibitor of SHP2 binding between the c-SHP2 and PTP domains, the Authors find that TNO155 not only inhibits the phosphatase activity of SHP2 but also reorganizes protein interactions. The insights provided by the data presented in the manuscript offer an opportunity to understand the effects of SHP2 allosteric inhibitors that cannot be explained by the effects on the phosphatase inhibitory activity. The manuscript is very interesting, and well written.

Classical immunoprecipitation-based proteomics has limitations for the detection of physiological/pathological protein-protein interactions. Immunoprecipitation requires the destruction of the cell structures, which can remove weak protein-protein interactions. Enzyme-substrate interactions do not require strong protein-protein interactions. SHP2 is a phosphatase whose activity can be transient. Thus, proximity labeling proteomics has the potential to overcome this limitation. The Authors provide a new dataset of the SHP2 interactome in the setting of WT SHP2 and SHP2 mutants, which is very informative for the broad community and may serve as a basis for new avenues of investigation.

However, the results are totally dependent on the strength of the assay system and many conclusions in this paper are based solely on the results of proximity labeling proteomics. Unfortunately, proximity labeling proteomics with TurboID has technical limitations.

Major comments

1. Potential steric inhibition of the fused TurboID in the rewiring interactome of SHP2 WT and mutants

The focus of the manuscript is on the effects of SHP2 mutations on the WT SHP2 interactome. The biotin ligase, TurboID, was

fused to the C-terminus of SHP2. TurboID is approximately 35 kDa. This size could be sufficient to disrupt certain protein-protein interactions by steric inhibition. Many sections draw conclusions without considering this possibility.

There are other proximity labeling methods that do not need fusion of biotin ligase, e.g., proximity labeling proteomics using peroxidase-conjugated antibody. This type of method and the data would be useful to evaluate the effects of the steric inhibition in the SHP2 interactome. Did the authors consider validating their assay with other approaches to proximity labeling?

2. The SHP2 protein-protein interactome in the HEK293 cell line could differ in other cell types.

In this paper, most of the experiments are highly dependent on the HEK293 cell line, a special cell line. In addition, the Authors have relied upon SHP2-depleted cells reconstituted with WT or mutant SHP2 types, enhancing expression above the physiological levels in these cells. It is known that many proteins have different binding partners in different cells, often because gene expression may differ in different cell types. The Authors should avoid giving the impression that the results presented here represent the comprehensive and ubiquitous interactome of SHP2 in different cell types. Additional data are required using additional cell types to detect potential variability across cell types. If the Authors are unable to provide new datasets using other cell types, they should discuss this limitation in the Discussion.

3. Mitochondrial localization in Supplementary Figure 6 needs appropriate controls.

The immunofluorescent staining images shown in Supplementary Figure 6 can be interpreted as having extremely high noise. The SHP2 antibody used (Cell Signaling Technology D50F2) was validated by Western blotting in 293T cells with SHP-2 knockout. Although SHP2-specific bands are removed after SHP2 knockout some "nonspecific" bands remain.

a. In the Methods section of Supplementary Figure 6, it is stated that primary antibodies were added without blocking during staining. How can non-specific antibody binding be prevented without blocking?

b. The legend for Supplementary Figure 6 notes: "Endogenous SHP2," but in the Methods section, it says "Transfection." Which is correct?

c. Why were U3-OS cells used instead of HEK293 cells? HEK293 KO cells should serve as a useful negative control for comparison with the original HEK293 cells. Please provide better images for endogenous SHP2 with appropriate controls.

d. Please also provide fluorescent images of SHP2 mutants to support changes in mitochondrial localization of SHP2 mutants, especially R138Q, T468M, Q510K, and E76K.

e. Please analyze more than just 2 cells. The conclusion should be made based on statistical analysis.

Minor comments

4. Non-specific biotinylation of mitochondrial proteins without exogenous biotin

Endogenous biotin and ATP are abundant in mitochondria. In mitochondria, both substrates of TurboID are endogenously expressed. Please provide additional data to Figure 5B that compares exogenous biotin addition to no biotin addition. The results from this suggested experiment may provide useful information to better interpret the mitochondrial results from proximity labeling.

5. In the original report of the assay, 500 micromolar biotin was used. Here, the Authors have used 100 micromolar. How did the Authors validate this concentration of Biotin?

Response to Reviewers

We thank the editors for the opportunity to resubmit our revised manuscript, and we thank the reviewers for their constructive feedback. Guided by the reviewers' comments, we have made a number of improvements that help support our claims. In brief, the major changes we made are as follows:

1. We have conducted an extensive comparison of our datasets with BioGrid, and the overlap between this vetted database and our datasets are now highlighted in the paper (**Figure 3E and Dataset EV5**).
2. We have added further validation of one of the novel mitochondrial hits from our study, ATPFAP1, showing that mutation of a predicted phosphotyrosine binding site diminishes SHP2 binding to this protein (**Figure 5H**).
3. We expanded the section of the manuscript that talks about how endogenous biotin in mitochondria could bias the mitochondrial TurboID results. Our findings suggest that this may be an issue when no exogenous biotin is added, but not an issue when excess biotin is used, as we did in our proteomics experiments (**Appendix Figure S3**). This will be a valuable finding for people in the field who are using BioID/TurboID to study mitochondrial interactions.
4. We expanded our microscopy analysis to address reviewers' technical concerns and to substantiate our claims that SHP2 is partly localized to the mitochondria (**Figure EV4**).
5. We have significantly revised the section of the manuscript that focuses on SHP2 mitochondrial interactions and function, highlighting specific interactors in our dataset that are also present in BioGrid or in other published papers.
6. We have also rewritten a large portion of our analysis of inhibitor-dependent changes in the interactome of SHP2 and replaced associated **Figure 7E** with one that provides more insights..
7. We generated several AlphaFold 3 models of SHP2-interactor complexes. While many of these models were not really interpretable, one involving cyclophilin D was particularly intriguing and we added it to the manuscript (**Appendix Figure S5**).

The specific detailed responses to editorial and reviewer comments are given line-by-line below. Where we have made changes to the manuscript file, those sections are colored in red text.

Editorial points

We have included this file in our resubmission. All of the changed parts of the manuscript text are colored in red text.

3) We replaced Supplementary Information with Expanded View (EV) Figures and Tables that are collapsible/expandable online. A maximum of 5 EV Figures can be typeset. EV Figures should be cited

as 'Figure EV1, Figure EV2" etc... in the text and their respective legends should be included in the main text after the legends of regular figures.

<https://www.embopress.org/page/journal/14693178/authorguide#expandedview>;

We have re-organized all of our Supplementary Information figures into 5 Expanded View and 5 Appendix figures.

The Supplementary Tables/Datasets have been relabeled as EV, and a separate tab has been added to each file with the legend.

4) a .docx formatted letter INCLUDING the reviewers' reports and your detailed point-by-point responses to their comments. As part of the EMBO publication's Transparent Editorial Process, EMBO reports publishes online a Review Process File (RPF) to accompany accepted manuscripts. This File will be published in conjunction with your paper and will include the referee reports, your point-by-point response and all pertinent correspondence relating to the manuscript.

<https://www.embopress.org/page/journal/14693178/authorguide#transparentprocess>

The point-by-point responses have been included in the resubmission as a .docx file.

5) a complete author checklist, which you can download from our author guidelines <https://www.embopress.org/page/journal/14693178/authorguide>. Please insert information in the checklist that is also reflected in the manuscript. The completed author checklist will also be part of the RPF.

The completed author checklist has been included in the resubmission.

6) Please note that all corresponding authors are required to supply an ORCID ID for their name upon submission of a revised manuscript (<<https://orcid.org/>>). Please find instructions on how to link your ORCID ID to your account in our manuscript tracking system in our Author guidelines

<<https://www.embopress.org/page/journal/14693178/authorguide#authorshipguidelines>>;

The corresponding author's ORCID has been associated with their account.

7) Before submitting your revision, primary datasets produced in this study need to be deposited in an appropriate public database (see

<https://www.embopress.org/page/journal/14693178/authorguide#datadeposition>). Please remember to provide a reviewer password if the datasets are not yet public. The accession numbers and database should be listed in a formal "Data Availability" section placed after Materials & Method (see also <https://www.embopress.org/page/journal/14693178/authorguide#datadeposition>). Please note that the Data Availability Section is restricted to new primary data that are part of this study. * Note - All links should resolve to a page where the data can be accessed. *

We have deposited all of our primary datasets in public databases (PRIDE, Dryad, BioImage Archive), and this information has been added to the Data Availability section.

Additional information on source data and instruction on how to label the files are available: <https://www.embopress.org/page/journal/14693178/authorguide#sourcedata>

The source data has been compiled as per the instructions and included with the resubmission.

9) Our journal encourages inclusion of *data citations in the reference list* to directly cite datasets that were re-used and obtained from public databases. Data citations in the article text are distinct from normal bibliographical citations and should directly link to the database records from which the data can be accessed. In the main text, data citations are formatted as follows: "Data ref: Smith et al, 2001" or "Data ref: NCBI Sequence Read Archive PRJNA342805, 2017". In the Reference list, data citations must be labeled with "[DATASET]". A data reference must provide the database name, accession number/identifiers and a resolvable link to the landing page from which the data can be accessed at the end of the reference. Further instructions are available at

Noted.

10) Regarding data quantification (see Figure Legends:

<https://www.embopress.org/page/journal/14693178/authorguide#figureformat>)

- the name of the statistical test used to generate error bars and P values,
- the number (n) of independent experiments (please specify technical or biological replicates) underlying each data point,
- the nature of the bars and error bars (s.d., s.e.m.),
- If the data are obtained from n less than 2, use scatter blots showing the individual data points.

We have updated the figure legends with all of this information.

This statement has been added to the manuscript.

12) Please also note our reference format:

We have updated our reference format throughout the manuscript.

13) All Materials and Methods need to be described in the main text using our 'Structured Methods' format, which is required for all research articles. According to this format, the Methods section includes a Reagents and Tools Table (listing key reagents, experimental models, software and relevant equipment and including their sources and relevant identifiers) followed by a Methods and Protocols section describing the methods using a step-by-step protocol format. The aim is to facilitate adoption of the methodologies across labs. More information on how to adhere to this format as well as a downloadable template (.docx) for the Reagents and Tools Table can be found in our author guidelines: <https://www.embopress.org/page/journal/14693178/authorguide#structuredmethods>.

We have converted our Materials and Methods section to the Structure Methods format and added the Reagents and Tools Table.

Thank you for the opportunity, but we will opt out of this.

Referee #1:

Here, the authors have used TurboID-based proximity proteomics in living cells to catalogue the proteins that interact transiently with the PTPN11/SHP2 protein-tyrosine phosphatase, comparing the interactors observed for WT SHP2 with those observed with 10 different mutant SHP2 proteins, representative of both GOF and LOF mutations that are causally implicated in human disease. For this purpose, they fused TurboID to the C-terminus of SHP2, the activated E76K SHP2 mutant and 9 other mutants. These were transiently expressed in SHP2 KO HEK293 cells, which were exposed to biotin for 10 min, followed by MS-based proteomic analysis of biotinylated proteins, leading to identification of 78 core interactors, including several known SHP2 interactors, such as GAB1, as well as novel interactors, including the GDI1 and GDI2 Rab protein regulators, the UBL5 and SNRP spliceosome components, the PPIF mitochondrial PPIase, and the HER2/ERBB2 RTK. When the transfected HEK293 cells were stimulated with EGF, 31 SHP2 interacting proteins were identified, most of which were unique to the stimulated condition, with another 30 proteins showing reduced interaction upon EGF stimulation. Some of the EGF-dependent interactors have been reported to contain pTyr sites, and to determine whether these sites might be directly recognized by the N- or C-SH2 domain of SHP2, they exploited a degenerate pTyr peptide library approach to define the primary sequence specificities of the N- and C-terminal SH2 domains; this allowed them to identify 26 and 10 predicted tight-binding phosphosites in unstimulated and EGF-stimulated conditions, respectively, including the known SHP2 binding site on GAB1. Using this baseline information, they went on to determine the consequence of disease-causing mutations on SHP2's interaction with cellular proteins. In the absence of EGF stimulation, the SHP2 mutants shared between 13 and 39 hits with WT SHP2, although with more relaxed cutoff criteria between 52 and 61 shared proteins were noted. Out of an average of ~240 core interacting proteins, significant enrichment over the control was found for 4/11 SHP2 variants in unstimulated cells and 3/11 variants in EGF stimulated cells, but different SHP2 mutants also had unique interactors. By analysis of T42A N-SH2 and R138Q C-SH2 domain mutations, known to reduce pTyr-protein recognition, they showed that T42A SHP2 enriched MPZL1 and ACOX1, which in prior work they had predicted to be SHP2 SH2 domain interactors. Although the R138Q mutation weakens pTyr binding to the C-SH2 domain by removing the conserved FLVRES motif Arg, SHP2 R138Q enriched more proteins than WT, but these were less dependent on pTyr. Mitochondrial proteins, including NDUF and COX proteins, constituted a large part of the core SHP2 interactome, with 21 mitochondrial proteins enriched by WT SHP2; several SHP2 mutants showed enhanced interaction with mitochondrial proteins, including ETC proteins. They found increased association with Tom20, the MLS receptor on the mitochondrial outer membrane, for SHP2 mutants that exhibited increased mitochondrial protein interactions, and they proposed that these mutants have a partially unfolded "open" state that exposes a cryptic MLS, that promotes mitochondrial import; consistent with this, they found increased association with the Hsp40/DnaJ family mitochondrial chaperone proteins, which they propose might drive SHP2 mutant protein unfolding and thereby enhance its mitochondrial import. Finally, they tested the effects of the TNO155 (batoprotafib) allosteric SHP2 inhibitor, currently in clinical trials, on the SHP2 interactome. Consistent with its known mechanism of binding, incubation with TNO155 increased the melting temperatures of SHP2 WT and several SHP2 mutants, albeit to varying extents. They also found that TNO155 treatment reduced mitochondrial protein interactions and conclude that this was due to bound TNO155 altering both SHP2 interactions and its localization by changing its overall conformation.

This is a worthwhile proximity proteomics survey of SHP2 PTP-interacting proteins in control and EGF-treated transfected HEK293 cells that has uncovered new pTyr-dependent and independent SHP2 interactors, whose patterns were changed upon EGF treatment. Concurrent analysis of the effects of 10

SHP2 disease mutations on the SHP2 interactome revealed some disease-specific interactors that might be of functional relevance in the disease they cause. In some ways, the most interesting findings are the mitochondrial SHP2 interactors, where some SHP2 mutants showed elevated association, which could be of significance in altered mitochondrial function in disease. In this regard, the authors imply that SHP2 is imported into mitochondria to exert mitochondrial functions through interactions with mitochondrial proteins, in some cases by pTyr/SH2 domain binding. However, they do not provide any compelling evidence that SHP2 is intramitochondrial or that the observed interactions occur in the mitochondrial matrix. Although in general the proximity proteomics data are convincing, there are some weaknesses, and, overall, these studies lack true functional validation of any of the new SHP2 interactions. Nevertheless, the WT and mutant SHP2 interactome databases should prove to be a valuable resource to the community for elucidating the roles of SHP2 mutations in human disease.

We appreciate the reviewer's summary of our work. We agree that our study and data are worthwhile resources to the community, and we hope that they will fuel future SHP2 mechanistic studies. Further details about the mitochondrial localization of SHP2 and validation of interactors are given in the responses to specific points below.

1. Why was the number of SHP2 binders decreased in the EGF-treated cells? The authors argue this could be due to the fact that SHP2 is diffusely localized in untreated cells, whereas it is translocated to major areas of Tyr phosphorylation in EGF-treated cells, such as the plasma membrane, but did not demonstrate such translocation experimentally. An alternative possibility is that this is due to negative feedback phosphorylation of SHP2, or possibly oxidation of the active site Cys in response to EGF-induced H₂O₂ production?

This is a great question. As noted by the reviewer, in our original manuscript, we posit that this is due to SHP2 concentrating at pY phosphorylation sites at the membrane upon EGFR stimulation. In fact, this has been seen previously by Paulsen et al (PMID: 22158416), which we now point out in the corresponding section. Notably, SHP2 is, indeed, also Cys-oxidized in response to EGF treatment, which was shown in that same paper (PMID: 22158416), and we agree with the reviewers that this is also likely to disrupt interactions. This is now also mentioned in the corresponding section, and we thank the reviewers for taking us a little further down this line of thought!

2. The authors went to great lengths to define primary pTyr sequence specificities of the SHP2 N- and C-SH2 domains and then use this information to predict which pTyr residues in interactors were likely to bind to one or other SHP2 SH2 domain, but did not validate any of the predicted pTyr/SH2 interactions by Tyr to Phe mutations. In this connection, it might be possible to carry out pTyr-enrichment proteomics on tryptic digests of the isolated biotinylated proteins that were in close proximity with SHP2 as a means of directly identifying pTyr sites in interactors that might be bound to the SHP2 SH2 domains (which would protect the interacting pTyr residues from dephosphorylation)?

We appreciate the reviewer's point about validating some pTyr/SH2 interactions. To this end, we wanted to point out a few supporting pieces of data in the original manuscript, and one new piece of data:

(i) Our SH2 profiling method faithfully predicts many known SH2 binding sites (**Figure 2F**), and notably, a few of those proteins are hits in our TurboID data for one or more SHP2 variants (eg. Gab1, Gab2 MPZL1). While this is not an orthogonal validation experiment, it supports our claims that looking at the TurboID data through an SH2 prediction lens can identify specific pY interaction sites.

(ii) For MPZL1/Pzr, the specific phosphosites have been well-studied and Y-to-F mutations have been made by others to confirm the binding sites. In this paper, we showed that the T42A mutation in the pY binding pocket of the N-SH2 domain further enhances this binding. This was observed both in peptide-level screens with libraries containing the specific phosphopeptides (**Figure 4C**) and via co-IP experiments with full-length SHP2 and MPZL1 (**Figure 4D**). Given that the T42A mutational effect is localized to the pY binding pocket of the N-SH2 domain (PMID: 37502916, this is effectively a mutational validation).

(iii) For ATPAF1, which is a new interactor, in the original manuscript, we predicted Y23 as an SH2 binding site, but we only showed co-IP of SHP2 and ATPAF1 (**Figure 5G**). Now, in the revised manuscript, we show that the Y23F mutation reduces but does not abolish ATPAF1/SHP2 interaction (**Figure 5H**). We did some AlphaFold modeling that suggests other potential binding sites (**Appendix Figure S5**).

(iv) The R138Q mutation removes the key arginine in the FLVR motif of the C-SH2 domain, making it unable to tightly bind phosphotyrosine residues. Indeed, the R138Q mutant shows reduced proximity labeling of several proteins with predicted C-SH2 binding sites that are otherwise labeled well by SHP2 wild-type and other mutants (newly added **Figure EV3F**).

Finally, while we agree that phospho-proteomics on the biotinylated and streptavidin-enriched proteins would be extremely interesting, this would be very challenging due to the required sample amounts for pY detection/enrichment. Furthermore, we were unable to do additional proteomics for the revisions, due to both time and resource limitations.

3. The authors partially addressed the concern that overexpression of exogenously expressed SHP2-TurboID protein might result in spurious SHP2 interactions, by estimating that SHP2-TurboID overexpression was only ~4-fold over endogenous. However, because they used transient expression for these experiments, not every cell will have been transfected and in some transfected cells the concentration of SHP2-TurboID could have been significantly higher. As discussed below, it is also not clear that SHP2-TurboID has the same subcellular distribution as endogenous SHP2, and this needs to be ascertained by IF staining.

We agree that overexpression as well as TurboID fusion could potentially impact localization or lead to false-positive interactions. We have added a paragraph to the Discussion section that points out these and other limitations to our study, as they will be constructive for the field moving forward. We have also added some new immunofluorescence data that show the similarities between native SHP2 and SHP2-TurboID (Figure EV4). Overall, our immunofluorescence shows that SHP2-TurboID is primarily cytoplasmic, as is the case for endogenous SHP2 and SHP2 constructs lacking TurboID. However, we note that our TurboID construct has a nuclear exit sequence, and so we see diminished nuclear localization by IF for that construct.

4. For the TurboID proximity proteomic experiments, the authors used a standard 10-minute biotin treatment of transfected cells initiated at the same time as EGF was added. This protocol will identify early EGFR-dependent SHP2 interactions, but additional SHP2 partner information could be obtained if they carried out a time course, particularly for the EGF-simulation condition, e.g., by adding biotin at different times after EGF, which could reveal the kinetics of binding and dissociation of SHP2 interactors.

We agree with the reviewers that this is a great idea, and we might identify other SHP2 interactors at longer time points. We opted to focus on a single time-point given the scale of the project and the sheer number of samples to compare (many SHP2 variants +/- EGF, with matched negative controls lacking biotin). We opted to focus on an earlier timepoint as several studies have shown that much EGFR-proximal signaling happens in the first 30 minutes (PMID: 15951569). We also note that a recent phosphoproteomics study examining SHP2 inhibition in an EGFR stimulation context examined 0, 5, 10, and 30 minute timepoints (PMID: 33755016)

5. The authors explained the observed differences in the SHP2 interactome dependent on either EGF treatment or the individual SHP2 disease mutations in part by differences in the subcellular localization of SHP2, but the evidence that such localization differences exist is actually the case is rather minimal. There is one immunolocalization figure (**Figure S6**) where partial co-localization of endogenous SHP2 and mitochondria was observed, but this was done in a different cell line, i.e. U2-OS cells. Additional immunolocalization experiments with HEK293 cells are needed, particularly for SHP2-TurboID with and without EGF treatment to assess whether EGF treatment results in re-localization of SHP2-TurboID as they propose. The subcellular localizations of the mutant SHP2-TurboID proteins with and without EGF treatment also need to be determined to assess whether the SHP2 mutants exhibit increased mitochondrial association or nuclear localization, as is suggested.

We agree that further evidence of SHP2 localization in the mitochondria and mutant-dependent changes in this mitochondrial localization would strengthen the manuscript. To this end, we attempted to do the suggested microscopy studies. First, we note that all of our new microscopy is done in HEK 293 cells, both wild-type cells to look at endogenous SHP2 levels, and transfected SHP2 knock-out HEK 293 cells, as used in our proteomics paper (**Figure EV4**). Both in for endogenous SHP2 staining and staining of cells transfected with SHP2 and SHP2-TurboID constructs, we see co-localization of SHP2 and mitochondrial pixels. We note that our SHP2 antibody was validated by a complete lack of staining of untransfected SHP2 knock-out cells. In our endogenous SHP2 microscopy, we do see an EGF-dependent reduction in mitochondrial co-localization. However, it is difficult to ascertain whether this is due to SHP2 exiting mitochondria or due to EGF-dependent redistribution of overall SHP2 intensity. In the transfected cells, the SHP2 levels in the SHP2+ cells are a bit higher, and it is difficult to measure changes in co-localization. We discuss all of this transparently in a fully revised microscopy section.

Regarding mutant-dependent changes in SHP2 mitochondrial localization, we did some preliminary experiments, however we found that in transfected cells, microscopy was inconclusive, for the same reasons that it was inclusive for +/- EGF. Thus, we opted not to put any mutation-dependent microscopy data in the paper. Beyond these microscopy studies, we want to emphasize that in our original submission, we showed enhanced mitochondrial localization of a few mutants by mitochondrial isolation and immunoblotting (**Figure 5C**). We also note that many studies across a variety of cell types have now reported SHP2 mitochondrial localization and function. Notably, a few of these studies have further suggested that mutations can impact SHP2 mitochondrial localization and/or function. These points are discussed in greater detail in the response to the next question.

6. The large number of mitochondrial proteins found in the SHP2 interactome data are consistent with a mitochondrial localization for SHP2. However, even though there have been some reports of SHP2 interaction with mitochondrial proteins, SHP2/*PTPN11* is not listed as a mitochondrial protein in MitoCarta 3.0, the best validated mitochondrial protein database, indicating that SHP2 is either an extremely minor

mitochondrial protein or a cell type specific visitor. The authors did show in Figure 5C that SHP2 was present in a crude mitochondrial fraction, but it is unclear what fraction of cellular SHP2 was present in the mitochondrial fraction, and this needs to be estimated. As indicated below, further analysis is needed to determine whether the SHP2 associated with mitochondria is in the mitochondrial matrix or is associated with the outer mitochondrial membrane.

It is true that SHP2 is not in MitoCarta 3.0, which we agree is the gold-standard for the mitochondrial proteome. However, this database is quite stringent, in that it requires evidence from multiple cell lines and also omits proteins that have modest-confidence mitochondrial data coupled with strong evidence of other subcellular localization (PMID: 33174596). Indeed, in the paper describing MitoCarta 3.0, the authors point out that their approach to annotating the mitochondrial proteome can be subjective, and they also note that their database has fewer entries than several other high-quality databases that MitoCarta 3.0 overlaps significantly with. Thus, one could reasonably argue that absence from MitoCarta 3.0 does not absolutely preclude the possibility that a protein has some (maybe context-dependent) mitochondrial localization or function. Indeed, one complementary database mentioned in the MitoCarta 3.0 paper, COMPARTMENTS (PMID: 24573882, <https://compartments.jensenlab.org/Search>), annotates SHP2 as having medium-confidence mitochondrial localization. Similarly, the MitoCoP high-confidence mitochondrial proteome, which was published almost concurrently with MitoCarta 3.0 and is derived from a variety of proteomics experiments and literature analyses, identified ~1,100 mitochondrial proteins (slightly more than the ~900 in MitoCarta 3.0 database), and SHP2 was among these proteins in MitoCoP (PMID: 34800366). Furthermore, as cited extensively in our manuscript, *many* studies have shown that SHP2 can enter mitochondria, bind proteins, dephosphorylate proteins, and alter mitochondrial functions (for example, PMID: 19835954, 21063443, 29255148, 15520383, 36980473, 38451719, 22952679, 23884424, 18583343, 23675459, 15378208). Critically, some of these studies did not involve SHP2 overexpression, and they have used a variety of cell lines. Thus, while SHP2 may not be a major resident of mitochondria, it is very likely either a transient/minor mitochondrial protein or a cell-type specific mitochondrial protein, as the reviewers point out.

From the analyses and experiments in our paper, we are unable to estimate what fraction of total SHP2 in the cell is mitochondrial. While this would be interesting to know, we don't believe that knowledge of this number has any real bearing on defining the mitochondrial interactome or functions of SHP2. With respect to sub-mitochondrial localization, we note that the mitochondrial SHP2-TurboID hits in our study are primarily inner membrane and mitochondrial matrix proteins (**Figure 6E**), consistent with many of the studies suggesting that SHP2 has roles in the matrix and also that it modulates respiration. This is also consistent with one very extensive study that conducted mitochondrial fractionation and protease treatment experiments, along with knock-down of outer and inner membrane import machinery, to convincingly show that SHP2 translocates through both mitochondrial membranes and enters the matrix (PMID: 29255148). Notably, that study did some of these experiments with endogenous SHP2, and they also identified the same import machinery (Tom20/40 and Tim23 complexes) that are labeled by our SHP2-TurboID constructs (**Figure 6A,B** and **Figure EV5B**). While we appreciate that further fractionation experiments could be done to orthogonally assess the sub-mitochondrial localization of SHP2-TurboID, we feel that the concordance between our data and other studies lends credence to our results, and a deeper dive into SHP2 mitochondrial function can be the focus of a follow-up study.

7. Some of the HSP40/DnaJ family proteins do have reported pTyr sites, but these were all identified through HTP studies, and it is not clear whether their phosphorylation would regulate their chaperone

activity or what tyrosine kinase would phosphorylate these sites - there is little convincing evidence the cytoplasmic tyrosine kinases get into mitochondria.

We agree with the reviewer that there is no direct evidence that any of the reported pTyr sites on the DnaJ proteins of interest actually regulate DnaJ function. However, we want to emphasize that we are not claiming this in our paper. Rather, we were merely pointing out that some reported DnaJ phosphorylation sites are potentially good SHP2 binding sites, based on their sequence, and that this is intriguing given that those DnaJ proteins are enriched interactors in our TurboID datasets. In any case, in our revised manuscript, the mentions of DnaJ phosphorylation have actually been removed, as we fully rewrote the first sections about SHP2 interactions with mitochondrial proteins.

Regarding the evidence of tyrosine phosphorylation in mitochondria, it is true that there are no tyrosine kinases in MitoCarta 3.0. However, many studies have documented that tyrosine kinases can phosphorylate mitochondrial proteins. One such example is a study that compared Src +/+ and Src -/- mice, isolated mitochondria from various tissues in these mice, and did phosphoproteomics (PMID: 32428391). This study found Src deletion-dependent changes in tyrosine phosphorylation on several mitochondrial proteins. Another recent study showed that during ER stress, Abl kinase is targeted to mitochondria (PMID: 11509666). These are just two of many papers that talk about Src-family kinases and Abl modulating the mitochondrial proteome.

Furthermore, there is quite a large and increasing body of data showing that mitochondrial proteins are tyrosine phosphorylated. For example, a recent study extensively characterized tyrosine phosphorylation in metabolic enzymes and found many tyrosine phosphorylation sites on mitochondrial enzymes that are likely to be functionally relevant (PMID: 40441152). Additionally, many mitochondrial proteins have high-confidence tyrosine phosphorylation sites in PhosphoSitePlus. We compiled a list of tyrosine phosphorylation sites with high PhosphoSitePlus prevalence (extracted from the database two years ago), which are found on mitochondrial proteins that are present in MitoCarta 3.0. Notably, many of these proteins show up as a statistically significant hit in one or more of our SHP2-TurboID datasets relative to the negative control:

- PDHA1 Y301, 1162 references (also a SHP2-TurboID hit)
- PDHA2 Y229, 1162 references
- LDHB Y240, 948 references
- NAXD Y85, 762 references
- SSBP1 Y73, 494 references (also a SHP2-TurboID hit)
- PDHA1 Y289, 367 references (also a SHP2-TurboID hit)
- PDHA2 Y287, 367 references
- NOA1 Y77, 275 references
- BCS1L Y181, 260 references
- PHB Y249, 255 references
- CEP89 Y157, 243 references
- BID Y54, 231 references (also a SHP2-TurboID hit)
- GLUD1 Y451, 228 references (also a SHP2-TurboID hit)
- GLUD2 Y451, 228 references
- SLC25A5 Y191, 204 references
- SLC25A6 Y191, 204 references (also a SHP2-TurboID hit)
- ACADSB Y198, 191 references (also a SHP2-TurboID hit)
- ECI2 Y285, 177 references (also a SHP2-TurboID hit)

- NDUFB7 Y89, 99 references (also a SHP2-TurboID hit)
- HSPD1 Y227, 93 references (also a SHP2-TurboID hit)
- ATP5PO (ATP5O), 88 references (also a SHP2-TurboID hit)

While it is formally possible that every single one of these sites is phosphorylated outside of mitochondria, before import into this organelle, this seems somewhat implausible, particularly if some of these phosphorylation sites are regulatory. Thus, we feel that the signaling community is approaching a consensus that tyrosine phosphorylation in mitochondria happens and can be functionally relevant.

8. With further regard to the proposed SHP2 mitochondrial localization, the authors suggest that this is due to a cryptic mitochondrial localization signal (MLS) in SHP2, but did not identify/mutate this putative MLS to test this possibility. This raises the important question of which SHP2-TurboID interactors are mitochondrially anchored on the outer membrane, as opposed to being truly intramitochondrial? Either protease protection experiments with intact purified mitochondria or IEM localization of SHP2/SHP2-TurboID is required. Alternatively, can mitochondrial SHP2 observed by GFP-tagging SHP2? Is it possible that association of SHP2 with a bona fide mitochondrial protein facilitates its import by acting as a carrier. How does the mitochondrial SHP2 interactome differ if a potent N-terminal MLS is appended to SHP2-TurboID?

These are all great points/questions. We wish to stress that the mitochondrial localization sequence on SHP2 was not first identified by our group, but rather, it was reported in this paper (PMID: 29255148). In that paper, R4 and R5 in the putative SHP2 mitochondrial localization sequence were mutated to alanine, and this double mutation abrogated mitochondrial localization. We recapitulated this result by making the same mutations and showing, via mitochondrial isolation and immunoblotting, that SHP2 R4A/R5A no longer localizes to the mitochondria (**Figure 5C**). Notably, that previous study also showed via knock-down that SHP2 import into the mitochondrial intermembrane space and matrix depends on Tom20/40 and Tim23, which is also consistent with our proximity-labeling proteomics data. With respect to the specific mitochondrial proteins that we observe in our interactomics study: the majority of these proteins are in the mitochondrial matrix, inner membrane, and intermembrane space, and many are part of the respiratory chain complexes (**Figures 5D and 6E**). Our sub-mitochondrial localization annotations come from an extensive study mapping mitochondrial protein topology (PMID: 38632225), as noted in the Materials and Methods section. Given all of these points, the most salient explanation of our data is that our SHP2-TurboID construct is able to enter the mitochondrial matrix. Finally, we note that for this study, we did not attempt to enhance the mitochondrial localization sequence, we only ablated it (the aforementioned R4A+R5A) mutation, as an enhanced MLS would be even more likely than mild overexpression and TurboID fusion to produce physiologically irrelevant interactions.

9. With regard to the observed spliceosome SHP2 interactors, while there have been reports of nuclear SHP2, and Figure S6F shows a few nuclear puncta, both SHP2 itself (68 kDa) and the SHP2-TurboID fusion (~100 kDa = SHP2 68 kDa + TurboID 34 kDa) are too large to enter the nucleus passively. Is there an identified SHP2 NLS (or NES), and could the NLS be mutated to establish that the observed interactions occur within the nucleus?. Based on the N/C-SH2 consensus, are there relevant pTyr-containing protein targets for SHP2 in the nucleus, such as the spliceosome subunit proteins. The alternative to the interactions occurring with intranuclear SHP2 is that there is a population of spliceosome subunits in the cytoplasm that SHP2 could interact. Can a nuclear population of WT SHP2-TurboID be detected by GFP-tagging or IF staining and does this differ for the SHP2 mutants.

We agree with the reviewer that it is possible that the few nuclear interactors that we pick up in our datasets could be due to a sub-population of those proteins in cytosol. For this study, our TurboID construct had a nuclear exit sequence, and so any nuclear proteins we see in our proteomics data are largely due to leakiness of the NES coupled with the extremely high sensitivity proximity-labeling proteomics, or due to nuclear proteins in the cytoplasm. We agree that it would be interesting if some of the mutations in our study altered nuclear localization, as they do mitochondrial localization, but this is beyond the scope of our data.

10. Have the authors ruled out that some of the observed SHP2 interactions occur post lysis , a potential problem with high activity TurboID -? The authors did use EDTA in the lysis buffer but not denaturing detergents?)

While it is formally possible that some of our TurboID signal is due to biotinylation post-lysis, we followed the published protocols that include EDTA, which should stop TurboID function, and lysis at 4 degrees Celsius, which should further slow down any biochemical processes (PMID: 33139955). Given these lysis conditions, coupled with the concordance between our data and SHP2 interactions reported in other studies (BioGrid analysis in **Figure 3E** and **Dataset EV5**), we believe that the majority of our TurboID signal should be due to pre-lysis labeling events.

11. How many of the pTyr sites on SHP2 interactors can be dephosphorylated by SHP2, e.g. via catch and release proximity activity?

This is a great question, and one that we have also wondered about. Unfortunately, at this stage, we do not know the answer for a few reasons: (1) The field still has a relatively limited list of bona fide SHP2 substrates, and not all of the known ones are in our proteomics dataset. (2) It is still unclear whether there is any kind of consensus sequence or strong substrate sequence specificity for the SHP2 phosphatase domain, making predictions from primary sequence difficult (unlike for SH2 domains and many protein kinases). We originally sought to address these questions by using the D425A+C459S double mutant (catalytically dead and also substrate-trapping) in conjunction with TurboID. The Tonks lab recently implemented a similar approach with PTP1B, comparing wild-type PTP1B + BioID with substrate-trapping PTP1B + BioID (PMID: 36871762). This was a lovely paper, and the differential analysis between these two constructs led to the discovery of new PTP1B substrates. When we implemented the same approach, but using the D425A+C459S mutations in SHP2, we initially noticed large changes in subcellular proximity-labeling signal, which we attributed to changes in subcellular localization. On reflection, this makes sense. The C459S mutation not only inactivates SHP2 catalytic activity and traps substrates, but it also disrupts autoinhibition and structurally mimics the E76K mutation (PMID: 30375376 and 36806475). As such, this approach was actually not ideal for SHP2, and we omitted these experiments from the paper due to complications in their interpretation. We think D425A mutation, alone, might be worth trying in combination with TurboID moving forward, as it is less likely to perturb SHP2 autoinhibition. We could also try the T466A mutation, which may not impact autoinhibition but is substrate-trapping (PMID: 16413071). Some of these points were mentioned in the original manuscript in the Discussion section as a limitation/future-direction. We expanded on the limitations and future directions in the revised manuscript but still opted not to include our C459S data in the paper. In future work, we hope to try this again using less structurally disruptive substrate-trapping mutations.

12. Despite the convincing proximity proteomics data, only a few interactions were validated by additional approaches, and in the end we do not learn how the observed SHP2 interactions might affect the

functions of the target proteins or whether any of the novel/unique interactions with disease mutant SHP2s are functionally significant. Specifically, for pTyr-independent interactions, one would need to define the contact sites in SHP2 and the interacting protein, and show that mutation of key contact residues disrupts the TurboID labeling. Admittedly, this would be beyond the scope of this paper, but the authors could exploit AlphaFold 3 modeling to see if contact surfaces between SHP2 and interactors of interest could be identified.

We completely agree with the reviewer that the next phase of this work has to involve validating and functionally interpreting some of the interactions we have identified. As noted, much of this is beyond the scope of our study, but this is still a great point. For MPZL1/Pzr, it was previously shown that mutation of the tyrosine phosphorylation sites disrupts SHP2 binding, and so we opted not to repeat that here (PMIDs: 24865967 and 32584792). In the revised manuscript, we mutated the predicted high-affinity binding site on ATPAF1, and found diminished but not completely abrogated binding, suggesting additional contact points and or binding to a second pY site (**FIGURE 5H**). We produced AlphaFold 3 models of tyrosine-phosphorylated ATPAF1 bound to SHP2, which we hoped might provide some insights. While AlphaFold3 does dock pY23 into an SH2 pocket, it does so into the N-SH2, not C-SH2 pocket (as per our SH2 binding predictions), and more puzzlingly, SHP2 remains in an autoinhibited state in every model produced. This is very unlikely (but not impossible) for N-SH2-engaged SHP2 structures. Given this, we opted not to include this model in our manuscript.

We also explored AF3 models for many other interactors. For the most part, we were not able to gain any significant insights from these models, except in one case. For example, for MPZL1/Pzr, a known interactor, we see a sensible model of an active-state SHP2 bound to MPZL1, when it is phosphorylated, but an implausible interaction when MPZL1 is unphosphorylated, providing some sense that AF3 might “understand” SHP2 interactions. However, as noted above, the phospho-ATPAF1 modeled complex has SHP2 in an auto-inhibited state, despite N-SH2 engagement. One interesting interaction that stood out was that between SHP2 and PPIF (also known as Cyclophilin D, or CypD). In this case, there was a relatively consistent prediction of SHP2 residues 541-548 (EYTNIKYS) binding in the CypD active site in a manner that strongly resembles the binding mode of several known inhibitors (**Appendix Figure S5**, PMID: 36163383). Notably, PPIF/CypD is a peptidyl prolyl isomerase and it also has known scaffolding functions, both of which are important for its regulation of mitochondrial processes. Thus, while SHP2 is not known to be regulated by proline isomerization, it may be localized to CypD-relevant mitochondrial complexes via a direct interaction with CypD.

Other points:

1. Page 5: The cited Gingras group's protein phosphatase interactome paper (ref. 35) did not study PTPN11, but BioGrid lists 342 unique protein interactors for WT SHP2, found by IP/MS and proximity labeling, and some comparison of the authors' interactomes with this publicly available dataset would be instructive.

This is a great suggestion, and we agree that some juxtaposition with published datasets would be valuable. We note that the Gingras group paper that we cited (PMID: 27880917) does, in fact, do experiments with *PTPN11*/SHP2, however they only report two interactors via BioID, Paf1 and Ctr9. These are both present in our datasets, and the whole Paf1 complex is arguably the most persistent SHP2 interactor across all mutants tested in our study. It is noteworthy that the Gingras lab paper made an N-terminal BioID fusion and still picked up two well-established SHP2 interactors that we also see via

our C-terminal TurboID fusion. This is a good sign that the system is at least somewhat robust to fusion orientation.

The BioGrid list is definitely more expansive and spans many references. Thus, we compared the data from our study with the proteins reported as SHP2 interactors in BioGrid. There are 328 unique proteins reported to interact with SHP2 in BioGrid, documented across in 633 different entries in the database. Of these 328 proteins, 130 of them are present in our TurboID proteomics datasets, and of those, 49 are statistically significantly enriched in 1 or more of our SHP2 datasets over the TurboID-only negative control. Notably, many of these proteins are enriched for more than one SHP2 mutant and/or both in -EGF and +EGF samples. All of this analysis is summarized in newly added **Figure 3E** and **Dataset EV5**.

Finally, we want to specifically note that, among the SHP2 interactors in BioGrid, we found several mitochondrial proteins that further support our findings of SHP2 mitochondrial localization and function:

(i) As noted in the original manuscript and discussed above, PPIF/CypD was reported to interact with SHP2. We found this as a hit in our TurboID dataset, and it is also a reported interactor in BioGrid. Notably, PPIF plays a crucial role in the functioning of the mitochondrial permeability transition pore, and it interacts with a key regulator of the mitochondrial permeability transition, ANT1 (PMID: 12095984), which is a known substrate of SHP2 (PMID: 29255148). Finally, PPIF also regulates the mitochondrial ATP synthase machinery (PMID: 19801635, 29101324), which is relevant to other interactors found in our study (see iii below).

(ii) SSBP1 (single-stranded DNA binding protein 1) is a housekeeping protein that plays a role in mitochondrial biogenesis. SSBP1 is a reported interactor of SHP2 in BioGrid and also a statistically significant TurboID hit for several of our SHP2 variants. Notably, SSBP1 is reported to be phosphorylated at Y73 by mitochondrially-located Src kinase (PMID: 31819039), and this phosphorylation event regulates SSBP1 function. Y73 is a high-frequency phosphosite in PhosphoSitePlus (>500 references). It is plausible that SHP2 dephosphorylates this site to regulate SSBP1, however this remains to be demonstrated.

(iii) ATP5A1, the alpha subunit of mitochondrial ATP synthase (Complex 5) is annotated as a SHP2 interactor in BioGrid. While this protein does not show statistically significant TurboID labeling in our experiments, we note that many other components of Complex 5 are significant hits in our dataset (**Figure 5E**), as are regulators of Complex 5 assembly and function (e.g. PPIF and ATPAF1).

(iv) MRPL42 is a component of the mitochondrial ribosome and is annotated as a SHP2 interactor via BioGrid. While this protein is not in our TurboID dataset, many mitochondrial ribosome proteins are, and they show up in our core SHP2 interactome. Among these is MRPL28, which also has a predicted high-affinity C-SH2 binding site from our SH2 profiling (**FIGURE 5D**).

(v) HSPD1 encodes the mitochondrial chaperonin Hsp60, which is critical for protein folding and proteostasis in the mitochondrial matrix. Notably, this protein has been reported to interact with SHP2 both via co-IP experiments and in co-fractionation mass spectrometry proteomics. This protein is a statistically significant interactor for several of our SHP2 mutants via TurboID.

Overall, we wish to really thank the reviewers for promoting us to do a thorough juxtaposition of our datasets with BioGrid, as it expanded the scope of our study and strengthened some of our claims.

2. Figure 2A: Reasonably the authors decided to use C-terminally fused TurboID to avoid interference with the N-SH2 autoinhibitory interaction, but did they actually test TurboID fused at the SHP2 N-terminus to see if this affected enzyme activity, or whether there was any difference in the repertoire of biotin-tagged proteins, perhaps due to the topology of binder interactions in this configuration?

While we did not test this ourselves, given the relatively large labeling radius for TurboID proximity labeling, we would not expect a big difference in labeled proteins due to fusion protein configuration. As noted in the response to the previous point, however, Gingras and co-workers used an N-terminal BioID SHP2 fusion and observed two known interactors, Paf1 and Ctr9, which were also present in our data (PMID: 27880917). While this does not definitively show that the fusion configuration won't matter, it at least suggests that there should be some overlap.

3. Figure S2A/B: They showed that the pY248 PD-1 peptide-stimulated SHP2-TurboID fusion protein catalytic activity, but the initial velocity values on the y axis are plotted on a log scale, and it looks as though the fusion protein's catalytic activity was in fact significantly lower than that of WT. This deserves comment.

The reviewers are right. The activity of the purified fusion protein is about 4-fold lower than the native protein. We don't know the reason for this (likely due to some self-biotinylation). Now, we explicitly mention the reduction in activity in the manuscript. It is worth noting, though, that the phosphatase activity of the SHP2-TurboID fusion in cells, in our Ras dephosphorylation assay, is comparable to unfused SHP2 (about 80% as active). Furthermore, we stress that the SHP2 TurboID construct is both responsive to activator phosphopeptides and responsive to activating mutations suggests that the fusion is largely functioning like the wild-type protein (**Figures EV1A-E**).

4. Figure 2E: It would be helpful if the N- and C-SH2 domain consensus sequences were provided rather than the reader having to figure this out from the heat maps.

In general, we suggest that consensus sequences are not as helpful as actual position weight matrices, as they collapse multidimensional data into a one-dimensional description of binding. We do note that the numerical position weight matrices were associated with the original submission as supplementary datasets, alongside the predictions from these weight matrices for every tyrosine phosphosite in PhosphoSitePlus. So we would prefer to keep the heatmaps depicting these position-weight matrices in the main text along with the fully accessible numerical source data and the full list of predictions.

5. Figure 4C: Did the authors determine if the observed SHP2 interactions with MPZL1 and ACOX1 were dependent on these pTyr sites by making Tyr to Phe mutants.

The interaction between SHP2 and MPZL1 is well known to be tyrosine phosphorylation-dependent (PMIDs 24865967 and 32584792), and so we did not re-validate this experimentally. Notably, our SH2 specificity profiling experiments recapitulated that the isolated SH2 domains bind to the known pY binding sites on MPZL1. Similarly, for ACOX1, our initial finding came from SH2 phospho-peptide screens, which identified specific binding sites, and then we observed this protein via TurboID labeling. Critically, both MPZL1 and ACOX1 show enhanced proximity labeling with the T42A mutation, as well as enhanced phospho-peptide binding in SH2 specificity screens. This mutation is in the pY binding pocket and well-established by us and others to enhance phosphopeptide/protein binding affinity (e.g. PMIDs: 39012820,

31936901). Thus, while we did not test the phosphorylation dependence on MPZL1/ACOX1 binding, our data and published results strongly support a model where both of these proteins bind due to tyrosine phosphorylation. As an aside, related to this point, we note that the R138Q mutation, which ablates C-SH2 binding affinity to phosphotyrosine, also does not enrich MPZL1 in our datasets, whereas most other mutations do (**Figure EV3E**). This is also seen for Gab1, Gab2, and other proteins with predicted high-affinity SH2 binding sites in newly added **Figure EV3F**.

6. Figure 5C: The mitochondrial fractionation experiment shows that there was an increase in mitochondrial association of the three of the four SHP2 mutants compared to WT SHP2, but it is not clear that the associated SHP2 was truly intramitochondrial, nor is it clear what fraction of the total cellular SHP2 is associated with mitochondria. Protease treatment of isolated mitochondria is commonly used to determine whether or not a protein is truly intramitochondrial.

We completely agree that our fractionation experiments cannot differentiate intra-mitochondrial SHP2 from mitochondria-associated extra-mitochondrial SHP2. The reasons we did not pursue a detailed protease treatment protocol to definitively determine intra-mitochondrial SHP2 localization are that: (1) This has already been done very carefully and convincingly in a study by Guo et al (PMID: 29255148), who also discovered the mitochondrial localization sequence on SHP2 and identified the disrupting R4A+R5A mutations, (2) we were able to recapitulate the effects of the R4A+R5A mutation in our mitochondrial isolation experiments (**Figure 5C**), and (3) a large fraction of the enriched mitochondrial proteins in our TurboID datasets are matrix proteins (**Figure 6E**). We felt that it would be extremely surprising if SHP2-TurboID was not entering the mitochondrial intermembrane space and matrix, given the numerous reports of SHP2 in the mitochondrial matrix (including that by Guo et al.), coupled with the large intra-mitochondrial signal in our datasets.

7. Figure 5E: Here, they showed that expression of a subset of the SHP2 mutants enhanced formation of mitochondrial reactive oxygen species in an in vitro assay, but these effects do not appear to correlate with the increased mitochondrial protein interactions shown for these mutants in panel E. It is also unclear whether interaction with specific ETC proteins is involved in increased ROS production, or whether this effect requires SHP2 phosphatase activity.

These are great points. At this stage, we do not know the mechanistic basis for the impact of SHP2 mutations on ROS production. However, we note that there may not be a direct correlation between extent of mitochondrial localization and ROS production, because each mutation has its own mechanistically-distinct protein-intrinsic effects on activity. For example, T42A enhances N-SH2 affinity for select phosphoproteins, whereas R138Q kills C-SH2 function, both without significantly perturbing autoinhibition (PMID: 39012820). E76K and Q510K both dramatically disrupt autoinhibition, but Q510K also perturbs catalytic activity and substrate specificity (PMID: 40631144). By contrast, we hypothesize that the differences in mitochondrial import across the variants are due to differences in overall stability, accessibility of the mitochondrial import sequence, and/or differences in chaperone engagement. Future efforts will focus on dissecting the precise mechanisms by which SHP2 engages and modulates the respiratory complexes.

8. Figure 5G: The authors imply that the observed coimmunoprecipitation of ATPAF1 and SHP2 is involves the mitochondrial populations of ATPAF1 and SHP2 as opposed to cytoplasmic populations. All nuclearly encoded mitochondrial protein are translated in the cytoplasm and then imported affording a chance for interaction with other cytoplasmic proteins.

We want to note that Panel 5G was not meant to validate localization in mitochondria. Rather, we did this experiment with co-expressed SHP2 and ATPAF1 just to show that the two proteins can interact. It is definitely possible that SHP2-TurboID labels mitochondrial proteins while they are produced/trafficked through cytosol, however, we see disproportionate labeling of mitochondrial proteins by some mutants, suggesting that SHP2 is actually in mitochondria, as opposed to only labeling mitochondrial proteins in cytosol while they are being trafficked there.

9. Figure S6: The overall pattern of endogenous SHP2 staining in U2OS osteosarcoma cells is surprisingly punctate, and only a small fraction of the cytoplasmic puncta seem to correspond to mitochondria. What cellular structures do these other puncta correspond to? Is this punctate staining pattern similar to that observed for SHP2 in other published studies, and is a similar pattern observed when different anti-SHP2 antibodies are used. To establish staining specificity, the proper experiment is demonstrate that all SHP2 staining is lost in SHP2 knockout U2-OS cells. While HEK293 cells are less suitable for IF staining experiments than the flat U2-OS cells, there are established protocols for doing this, and the authors need to demonstrate that the SHP2 staining patterns are similar in HE293 cells, including showing that SHP2 and SHP2-TurboID have similar staining patterns and also determining how EGF treatment affects SHP2 localization. As indicated above, what is really needed is direct evidence that SHP2 is present inside mitochondria.

We agree with the reviewer that our original manuscript did not include enough of the right microscopy experiments, both due to the cell line used (different from proteomics) and due to a lack of antibody control. We have significantly revised the microscopy and mitochondrial localization section substantially (see **Figure EV4A** and associated text). Some of this is already described in detail above in the response to questions 5 and 5 from Reviewer 1. Briefly, we conducted microscopy experiments in wild-type and SHP2 knock-out HEK 293 cells, initially untransfected, to look at endogenous SHP2 localization. These experiments confirmed our antibody is SHP2-specific. They also show that endogenous SHP2 can colocalize with mitochondria. There is an EGF-dependent effect in these samples, but it is inconclusive whether this is due to a change in localization or a change in SHP2 abundance upon EGF treatment. We then transfected the knock-out cells with SHP2 or SHP2-TurboID constructs and did imaging. We see mitochondrial co-localization of SHP2 in all of these samples, but the SHP2 levels are higher, and we cannot differentiate between +/- EGF. Notably, in the endogenous SHP2 samples, there is less nuclear SHP2 than cytoplasmic SHP2. This is somewhat mimicked in the transfected SHP2-TurboID samples, because our SHP2-TurboID construct has a nuclear exit sequence, whereas our SHP2 alone construct does not. Finally, regarding the SHP2 puncta in our original U2-OS images, we do not know for certain what this is, but we do see it for endogenous SHP2 in HEK 293 cells. There is some evidence in the literature that SHP2 can phase separate and form puncta in cells (PMID: 33002410), however we do not know if we are seeing the same phenomenon here.

10. Figure S6: There appears to be very little SHP2 staining in the nuclei of these U2-OS cells, but, what there is, is also punctate. This also needs discussion, and IF staining of HEK293 cells to determine if there is a nuclear population of SHP2-TurboID.

See the previous point, as well as the response to questions 3 and 9 from Reviewer 1.

11. Figure 7E: The effects of the TNO155 allosteric inhibitor on the SHP2 interactome are interesting, but whether this is due to altered subcellular localization as the authors concluded is unclear and needs to be examined by IF staining studies.

We agree that the effects of TNO155 treatment are interesting and warrant further investigation. Upon deeper analysis of our proteomics data, we feel that we cannot conclusively state that TNO155 alters localization. Given that we identify mitochondrial protein interactors of SHP2 that are sensitive to TNO155 as well as those that are insensitive to TNO155 (revised **Figure 7E**), the effects on localization are ambiguous. Perhaps what is more interesting is that there are some interactors of SHP2 that are impervious to TNO155 treatment, whether mitochondrial or not. Given this, we have changed the final paragraphs of the TNO155 section to simply discuss the range of effects seen from inhibitor treatment, and to postulate the various mechanisms that could explain these effects. We did some immunofluorescence on SHP2-transfected HEK 293 cells, with and without TNO155, but as noted in other responses above, it is difficult to measure any subtle localization effects of perturbations in the transfected cells via fluorescence microscopy, in the format that we did the experiment. As mentioned in the response to Reviewer 3, point 3d, below, we likely need to move to a different imaging modality to see this more clearly.

12. There are some additional controls that the authors might consider: 1. The use of an EGFR kinase inhibitor to demonstrate that the observed effects of EGF on SHP2 interactions are dependent on EGFR-mediated phosphorylation. 2. The use of SHP2 FLVRES motif SH2 domains mutants to demonstrate that the interactions are due to pTyr-SH2 binding. 3. The use of a catalytically-dead SHP2 Asp to Ala mutant controls that might reveal interactions with pTyr substrates by trapping (did the authors compare their SHP2 interactomes with those reported for SHP2 substrate trapping mutants in ref. 98)? 4. The use of an outer mitochondrial membrane-anchored TurboID fusion protein to define which inner mitochondrial proteins can be biotinylated as they are imported into mitochondria.

These are really interesting ideas/suggestions:

(1) Unfortunately, for this study, we felt that new proximity-labeling proteomics work with an EGFR inhibitor is out of scope, but it would certainly be an interesting follow-up study.

(2) Regarding the SH2 mutant, we wish to note that R138Q is indeed the relevant FLVR motif mutant for the C-SH2 domain. We added text to the R138Q section pointing out that R138 is the FLVR motif arginine, and we also added a figure panel showing that some known SHP2 interactors lose their signal upon R138Q mutation (**Figure EV3F**).

(3) This point was also discussed in the response to main point 11, above. We initially tried TurboID proximity labeling with the D425A+C459S mutation. However, we saw a dramatic change in apparent localization (based on TurboID labeling patterns) that resembled that seen for some of the open-conformation mutants with enhanced mitochondrial signal. On reflection, this is likely because the C459S mutation is known to induce an open conformation in SHP2 (PMID: 30375376, 36806475). Thus, this was not the right mutation to use. Regarding another SHP2 substrate trapping proteomics study (PMID: 36103635), we note that this report also used the D425A+C459S mutations. Nonetheless, we examined the list of 13 high-confidence substrates (12 if we exclude SHP2) from that paper to our TurboID data. Unfortunately, we found that while most of those proteins were in our datasets, only Gab1 and Gab2

were enriched by SHP2 variants over the negative control. This fact underscores the need for complementary methods to identify substrates.

(4) This last point is very interesting to us, but would require a totally new and extensive study. Other groups have been exploring questions like this. For example, one recent bioRxiv preprint conducted proximity-labeling proteomics with Tomm20 and Tomm70 to compare the proteomics that engage in two different mitochondrial import processes (<https://doi.org/10.1101/2024.10.25.620316>). Indeed, SHP2 emerges as a hit in the Tomm20 dataset from this work, and we cited this study in our manuscript.

Referee #2:

In this manuscript, the authors performed proximity-labeling proteomics, and mapped the interaction networks by wt and mutant SHP2 molecules. Furthermore, the authors also examined the SHP2 interactome in the presence of a specific allosteric inhibitor. It is interesting to observe the elevated mitochondrial location of some SHP2 mutants with potential impact on mitochondrial functions.

This article provides an excellent resource for further exploration of biochemical mechanisms of SHP2 functions in cell signaling in health and diseases.

We appreciate the reviewers enthusiasm for our study and agree that this will be a useful resource to enable future SHP2 signaling functions.

One minor concern is that, as the authors admitted, the expression levels of some exogenous proteins were 4 folds higher than the endogenous protein, which could compound the results of the interactomes. Related to this concern was the data on mitochondrial location of some mutants, whether this was due to over-expression? The authors need to discuss the possible caveats.

We agree that overexpression is a caveat and limitation of our approach. We have added some points about this to the Discussion section, along with a broader discussion of other limitations of our study. As noted in other responses to the reviewers, we do not believe that the mitochondrial import of SHP2 and enhanced mitochondrial signal for some mutants is due to overexpression, given that very large growing body of literature reporting SHP2 in mitochondrial (including several studies that have not involved SHP2 overexpression).

Referee #3:

The Authors used proximity labeling proteomics to study the localization and protein-protein interactions of wild-type SHP2 and 10 SHP2 variants. Some of these variants have clinical implications. Taking advantage of TNO155, an allosteric inhibitor of SHP2 binding between the c-SHP2 and PTP domains, the Authors find that TNO155 not only inhibits the phosphatase activity of SHP2 but also reorganizes protein interactions. The insights provided by the data presented in the manuscript offer an opportunity to understand the effects of SHP2 allosteric inhibitors that cannot be explained by the effects on the phosphatase inhibitory activity. The manuscript is very interesting, and well written.

Classical immunoprecipitation-based proteomics has limitations for the detection of physiological/pathological protein-protein interactions. Immunoprecipitation requires the destruction of the cell structures, which can remove weak protein-protein interactions. Enzyme-substrate interactions do not require strong protein-protein interactions. SHP2 is a phosphatase whose activity can be transient.

Thus, proximity labeling proteomics has the potential to overcome this limitation. The Authors provide a new dataset of the SHP2 interactome in the setting of WT SHP2 and SHP2 mutants, which is very informative for the broad community and may serve as a basis for new avenues of investigation. However, the results are totally dependent on the strength of the assay system and many conclusions in this paper are based solely on the results of proximity labeling proteomics. Unfortunately, proximity labeling proteomics with TurboID has technical limitations.

Major comments

1. Potential steric inhibition of the fused TurboID in the rewiring interactome of SHP2 WT and mutants
The focus of the manuscript is on the effects of SHP2 mutations on the WT SHP2 interactome. The biotin ligase, TurboID, was fused to the C-terminus of SHP2. TurboID is approximately 35 kDa. This size could be sufficient to disrupt certain protein-protein interactions by steric inhibition. Many sections draw conclusions without considering this possibility.

There are other proximity labeling methods that do not need fusion of biotin ligase, e.g., proximity labeling proteomics using peroxidase-conjugated antibody. This type of method and the data would be useful to evaluate the effects of the steric inhibition in the SHP2 interactome. Did the authors consider validating their assay with other approaches to proximity labeling?

This is a reasonable concern, and one that we also had going into this project. However, we feel that many lines of evidence, including our own controls and data, suggest that this is not an issue. First, we show in **Figure EV1** that the allosteric regulation of SHP2 is intact in a TurboID fusion context (it can be activated by activator peptides and by autoinhibition-disrupting mutations). Second, as noted particularly in the revised manuscript, our experiment identifies many known interactors that have also been identified through a range of other methods (**Figure 3E and Dataset EV5**), and in some cases (e.g. MPZL1 and Gab1) we see the expected mutation-dependent effects of these interactions (**Figure EV3E,F**). Third, some of the proteins that we identified as interactors are known to bind at different sites on SHP2 (e.g. SH2 domain binders like Gab1 and MPZL1, and tail phosphorylation site binders like Grb2), suggesting that many parts of the protein are available for binding, even when fused to TurboID. Fourth, we validate MPZL1 and ATPAF1 interactions via IPs in the absence of TurboID fusion (**Figures 4D and 5G**), further substantiating our results. We also conduct our mitochondrial isolations and MitoSOX assays as validation in the absence of TurboID fusion (**Figure 5C,F**). All of these observations are consistent with the fact that TurboID is fused at the C-terminus of a ~60 residue disordered linker, where it is unlikely to impact interactions within or proximal to the globular domains of SHP2. Finally, it is important to note that all of the proteins we define as 'hits' are ones that are statistically significantly enriched for SHP2-TurboID constructs over the negative control, and all mutants are analyzed in this same background. Thus, mutational effects are effectively normalized for the background effect of TurboID fusion.

Based on these points, we feel that TurboID fusion is not a significant perturbation, although it is still possible that some subset of interactors are lost. Given this, we did not further validate our TurboID proximity-labeling experiments using an orthogonal proximity-labeling approach, and we note that this would be a significant undertaking, effectively doubling the proteomics work in this project.

2. The SHP2 protein-protein interactome in the HEK293 cell line could differ in other cell types.
In this paper, most of the experiments are highly dependent on the HEK293 cell line, a special cell line. In addition, the Authors have relied upon SHP2-depleted cells reconstituted with WT or mutant SHP2 types, enhancing expression above the physiological levels in these cells. It is known that many proteins have different binding partners in different cells, often because gene expression may differ in different

cell types. The Authors should avoid giving the impression that the results presented here represent the comprehensive and ubiquitous interactome of SHP2 in different cell types. Additional data are required using additional cell types to detect potential variability across cell types. If the Authors are unable to provide new datasets using other cell types, they should discuss this limitation in the Discussion.

We completely agree that this study could be expanded upon by looking at other cell types beyond HEK293 cells. This study represents a first step toward a comprehensive cataloging of SHP2 interactions, where we used SHP2 knock-out HEK293 cells as a model system, both for methods development and for defining a portion of the SHP2 interactome. Given time and resource constraints, we deemed experiments with additional cell lines out of scope for this study, but we have added points about this and other limitations to the Discussion section.

3. Mitochondrial localization in Supplementary Figure 6 needs appropriate controls.

The immunofluorescent staining images shown in Supplementary Figure 6 can be interpreted as having extremely high noise. The SHP2 antibody used (Cell Signaling Technology D50F2) was validated by Western blotting in 293T cells with SHP-2 knockout. Although SHP2-specific bands are removed after SHP2 knockout some "nonspecific" bands remain.

This is a fair point. We further analyzed our antibody in more blotting experiments to show that the vast majority of signal is SHP2 specific (**Figure EV1F**). Perhaps even more importantly, we have extensively revised the microscopy section of our paper and now include a comparison of endogenous SHP2 staining in wild-type HEK 293 cells and SHP2 knock-out HEK 293 cells (**Figure EV4A**). This comparison shows that all SHP2 signal is lost in the knock-out cells.

a. In the Methods section of Supplementary Figure 6, it is stated that primary antibodies were added without blocking during staining. How can non-specific antibody binding be prevented without blocking?

This was an erroneous omission on our part. Thanks for catching this! The revised microscopy methods sections clarify that the samples were blocked with a buffer containing 5% BSA for 1 hour prior to applying the primary antibody.

b. The legend for Supplementary Figure 6 notes: "Endogenous SHP2," but in the Methods section, it says "Transfection." Which is correct?

Thanks for catching this error. In the original manuscript, we imaged endogenous SHP2, and so there was no transfection. In the revision, we fully removed the U2-OS cell microscopy, and now we are analyzing both endogenous SHP2 and transfected SHP2 in separate experiments (Figure EV4). The methods section has been fixed to reflect this.

c. Why were U3-OS cells used instead of HEK293 cells? HEK293 KO cells should serve as a useful negative control for comparison with the original HEK293 cells. Please provide better images for endogenous SHP2 with appropriate controls.

We originally used U2-OS cells because their morphology makes them easier to image, and the mitochondria are more discernable. But the reviewer is right that this should have been done in HEK 293 cells, and appropriate controls were needed. These points have now been addressed through a complete overhaul of the microscopy section and figure (**Figure EV4**), as discussed in previous responses above.

d. Please also provide fluorescent images of SHP2 mutants to support changes in mitochondrial localization of SHP2 mutants, especially R138Q, T468M, Q510K, and E76K.

Unfortunately, we were unable to do these experiments in any kind of conclusive way. As shown in **Figure EV4** and discussed in the corresponding part of the manuscript, it is very difficult to quantitatively ascertain differences in mitochondrial localization of SHP2 by microscopy in the transfected cells. Staining of Wild-type SHP2 and SHP2-TurboID already shows substantial overlap with mitochondrial pixels in our microscopy images, so any increase (or subtle decrease) is hard to discern. We can see these differences by proximity-labeling proteomics (**Figure 5B**) and mitochondrial isolation and blotting (**Figure 5C**), providing at least two orthogonal ways of making the same observation. In order to do this via microscopy, we would likely have to generate stable mutant cell lines, and/or move to a higher-resolution imaging modalities, which we felt were out of the scope of this study and beyond our resources.

e. Please analyze more than just 2 cells. The conclusion should be made based on statistical analysis.

This is a fair point. While we did analyze more than 2 cells in the prior version of the manuscript, we only showed a few representative images. Our revised microscopy section now includes analysis of hundreds of cells per cell type and condition tested, including some quantification (**Figure EV4**).

Minor comments

4. Non-specific biotinylation of mitochondrial proteins without exogenous biotin

Endogenous biotin and ATP are abundant in mitochondria. In mitochondria, both substrates of TurboID are endogenously expressed. Please provide additional data to Figure 5B that compares exogenous biotin addition to no biotin addition. The results from this suggested experiment may provide useful information to better interpret the mitochondrial results from proximity labeling.

We were also curious whether the enhanced mitochondrial signal in our experiments might be somehow due to a bias caused by the intrinsically higher biotin and ATP levels in mitochondria. In the original manuscript, we did some analysis (Supplementary Figure 5B, as noted by the reviewer), but we agree that this could be explored and discussed more. To that end, we expanded the initial mitochondrial section of manuscript to first focus on this question, and we included a whole new figure with new analyses (**Appendix Figure S3**). In this new figure, we look at the MS signal intensities in TurboID-only and SHP2(WT)-TurboID samples, both with and without biotin, and with and without EGF stimulation. From these comparisons, it is clear that, in the absence of endogenous biotin, there is elevated proximity-labeling signal in the mitochondria, relative to the whole proteome, both for TurboID alone and for the SHP2 construct. However, when exogenous biotin is added, this basically normalizes out any mitochondrial biotin bias for the TurboID-only control, and any remaining elevation seems to be SHP2-specific. Thus, we feel that in the presence of exogenous biotin, and when referenced against our TurboID-only negative control, the enriched proteins are truly enriched due to the presence of SHP2.

5. In the original report of the assay, 500 micromolar biotin was used. Here, the Authors have used 100 micromolar. How did the Authors validate this concentration of Biotin?

We chose to use 100 micromolar after a series of optimization steps early in the development of this project, but we note that our protocol very closely follows that laid out in the original *Nature Protocols* paper by Alice Ting's lab outlining detailed methods (PMID: 33139955). In that paper, they suggest

starting with 50 micromolar biotin and increasing up to 500 micromolar only if needed. In our hands, we first explored reducing the amount of FBS used during the TurboID labeling step from 10% for normal growth to 2.5% during biotin treatment, to reduce the amount of background biotinylation. Then we tested a range of biotin concentrations on cells expressing a GFP-TurboID fusion and found that 100 micromolar was sufficient to give detectable biotinylation across the proteome. An example of one of those experiments is shown below:

Dear Dr. Shah

Thank you for the submission of your revised manuscript to EMBO reports. I sincerely apologize for the delay in handling your manuscript, which is due to the currently high number of submissions. We have meanwhile received the full set of referee reports that is copied below.

As you will see, both referees support publication in EMBO reports after some minor revisions. Referee #1 comments on the lack of compelling evidence that mitochondrial SHP2 activity towards tyrosine phosphorylated mitochondrial proteins is functionally important. This limitation of functional validation does not preclude publication of a resource-type article, but should be discussed in the manuscript.

Referee #3 raises concerns about the lack of orthogonal validation of SHP2-interactors using antibody-based proximity labeling. The referee suggests being transparent about this limitation by adding a statement to the effect that novel interactors not validated by an orthogonal method (not only IP-mass spectrometry) could be real, reflecting the SHP2 interactome, but that artifacts cannot be excluded. The referee further suggests toning down the statement of having identified real interactors. These suggestions should be implemented as important information for future readers. Generally, conclusions on physiologically meaningful interactions and altered mitochondrial localization should be carefully phrased and limitations due to overexpression artefacts clearly articulated.

I suggest adding a "Limitations" statement at the end of the article to be fully transparent.

From the editorial side, there are also a few things that we need before we can proceed with the official acceptance of your study.

- Please describe your findings in the abstract in present tense.
- Please remove the main figures from the manuscript text.
- The PDF with EV figures and legends is not needed. The individual EV figure files and their legends in the manuscript text are sufficient.
- Please reduce the number of keywords to 5 and place them after the Abstract.
- Please use the header "Disclosure and Competing Interests Statement" instead of "Conflict of Interest".
- There is an error in the Appendix table of content: Appendix Figure S3 is listed twice while Appendix Figure S5 is missing).
- Appendix Figure S1, S4: please specify the nature of the replicates in the legend (biological, technical).
- It is sufficient to have the Appendix Figures in the Appendix PDF. The separately uploaded Appendix Figures are not needed. Moreover, the Appendix figure legends should be removed from the manuscript file.
- The Reagents and Tools table needs to be removed from the manuscript and uploaded as a separate Word file. Our production team will add it to the article during typesetting.
- Source data: please upload each figure folder as a separate zip archive. Thank you.
- I did a spot check on the source data and noted for Figure 4D that you provided a Western blot for the IP that probes for SHP2 and MPZL1 on the same blot. Would this not be the more appropriate band/result to show, since you probed for both proteins on the same blot?
- For Figure 5C the Western blots are missing from the Source data folders, it seems.
- We perform a routine plagiarism check on all revised manuscripts. Here I noted that the paragraphs on "Purification of SH2 domains", "Purification of full-length SHP2 proteins", "SH2 specificity profiling" appear to be identical to the corresponding sections in Vlimmeren et al, PNAS 2024. This is perfectly fine, no need to reinvent the wheel for methods, but I suggest citing this paper.
- Could you please cite the preprint from Akram et al 2024 in the text as (preprint: Akram et al, 2024)?
- Our data editors asked you to address the following points in the figure legends:
 - Please note that the exact p values are not provided in the legends of figures 4B, D; 5C, F, H; EV3 E; EV5 A-C.
 - Please indicate the statistical test used for data analysis in the legends of figures 3E, 4D, E, F; 5A, 7C, D; S1 A, B; S3 B
 - Please note that the box plots need to be defined in terms of centre, bounds of box and percentile in the legend of figure 6D
 - Please note that information related to n is missing in the legends of figures 4E, F; 5A, B, C; 6A, B, D, F; 7C, D, E; EV1 G; EV3

E, F; S3 A, B

- Please note that the error bars are not defined in the legends of figures 5C, EV1 G

- Please note that the measure of center for the error bars needs to be defined in the legends of figures 4B, 6A, F; 7E, EV1 A, B, E; EV3 E, F; EV5 A-C

- Figures 4B, 5C, 6A, B, F, 7E, EV1A, B, EV3E, F, EV5A, B, C miss a display of individual datapoints. Please add these in addition to the mean and error bars, if the number of 'n' permits.

- Regarding the Author Contributions, we now use CRedit to specify the contributions of each author in the journal submission system. Therefore, please remove the Author Contributions from the manuscript file and make sure that the author contributions in our online manuscript tracking system are correct and up-to-date. The information you specified in the system will be automatically retrieved and typeset into the article. You can enter additional information in the free text box provided, if you wish. See also our guide to authors <https://www.embopress.org/page/journal/14693178/authorguide#authorshipguidelines>.

- Finally, EMBO Reports papers are accompanied online by

A) a short (1-2 sentences) summary of the findings and their significance,

B) 2-3 bullet points highlighting key results and

C) a schematic summary figure that provides a sketch of the major findings (not a data image).

Please provide the summary figure as a separate file in PNG or JPG format at a size of 550x300-600 pixels (width x height).

Please note that the size is rather small and that text needs to be readable at the final size. Please send us this information along with the revised manuscript.

With kind regards,

Martina Rembold, PhD

Senior Editor

EMBO reports

=====

Referee #1:

In general, the authors have done a good job of addressing the reviewers' comments, adding some new data, including a comparison of their SHP2 interactome with that in BioGRID, stronger support for endogenous SHP2 co-localizing with mitochondria, and evidence that interaction of SHP2 with mitochondrial ATPAF1 is partially dependent on the phospho-Tyr23. However, although the correlative evidence for a mitochondrial role for SHP2 is suggestive, they still provide no compelling evidence that mitochondrial SHP2 activity towards tyrosine phosphorylated mitochondrial proteins is functionally important, even though some disease-associated, activating SHP2 mutants have been shown, as confirmed by the authors, to impact mitochondrial functions. Nevertheless, the expanded WT and mutant SHP2 interaction datasets reported here will undoubtedly be of value to the SHP2 community.

Referee #3:

The authors have done a good job at revising the manuscript. We are satisfied with the authors' responses to our major comments 3, 3a, 3b, 3c, 3e and minor comments. However, we are not entirely satisfied with the responses to our comments 1, 2 and 3d. In our view, these remaining points need to be addressed before the publication.

Remaining concerns

Regarding our comments 1 and 2:

The authors decided not to confirm their results using the orthogonal proximity labeling approach (antibody-based proximity labeling experiments). The authors claim that "Thus, our methodology is likely to be revealing true SHP2 interactions and biological functions."

In our view, the authors over-interpret their results. The authors selected 1) to use only HEK 293 cells, and 2) not to exclude false positives SHP2-binding partners by using as a control antibody-based proximity labeling, an approach that would be useful for excluding effects of TurboID fusion to SHP2 (especially for proteins in the mitochondria and the nucleus).

The authors reported many novel SHP2-interacting partners. If true, these interactions may underlie currently unknown functions of SHP2 in the mitochondria and the nucleus, which cannot be explained by SHP2 canonical mechanisms as a phosphatase downstream of receptor activation. Therefore, the authors reliance on canonical mechanisms to explain novel SHP2 interactions

is inadequate.

The many novel binding partners of SHP2, e.g., mitochondrial and nuclear proteins, suggest molecular mechanisms that differ from those underlying receptor-SHP2 mechanisms. These are very interesting data; however, caution is needed. The newly added Figure EV4E also supports this perspective (see also the later part). Based on the authors' expertise, antibody-based proximity labeling could be an ideal orthogonal approach to address this issue.

It seems that the authors find it difficult to perform antibody-based proximity labeling experiments at present, although they clearly have considerable specific expertise. Therefore, as an alternative, we would suggest adding a statement to the effect that novel interactors not validated by an orthogonal method (not only IP-mass spectrometry) could be real, reflecting the SHP2 interactome, but artifacts cannot be excluded. Additionally, these findings and the cellular localizations of SHP2 should be interpreted with great caution. Our hope is that other investigators following up on new information in this paper be advised to confirm the results before proceeding.

Regarding the concluding sentence in the Discussion ("Thus, our methodology is likely to be revealing true SHP2 interactions and biological functions."), please remove "true". This change is justified not only by the potential for artifacts mentioned above (my point 1) but also by the authors' selection to use only the HEK293 cell line (related to my point 2). The conclusion should reflect that the novel SHP2 interactions identified here may not represent the interactome of SHP2 across different, and more physiological cellular contexts.

Regarding our comment 3d:

The authors did not provide the requested data. Figure EV4E, now provided by the authors, reveals that:

- 1) SHP2-WT seems to be expressed in only a small number of HEK293 cells, even with the very high transfection efficiency in these cells, and
- 2) Transfected cells expressing SHP2-WT exhibit a smeared cytoplasmic distribution pattern, suggesting extreme overexpression of SHP2.

These observations suggest the possibility that this paper detected SHP2 interactions in non-physiologic conditions of extremely high expression, where SHP2-WT was transfected. This also suggests that such extremely high expression levels may saturate physiological molecular transport and distribution pathways. Therefore, the changes in mitochondrial distribution of SHP2 mutants concluded in this study should be interpreted with caution. As the authors state, it would be necessary to control the expression level within an appropriate range, such as by using a stable mutant cell line. These limitations have already been added to the Discussion section, so although the authors' statements responding to this reviewer are insufficient, I believe that a minimum requirements was met.

Dear Dr. Shah

Thank you for the submission of your revised manuscript to EMBO reports. I sincerely apologize for the delay in handling your manuscript, which is due to the currently high number of submissions. We have meanwhile received the full set of referee reports that is copied below.

As you will see, both referees support publication in EMBO reports after some minor revisions. Referee #1 comments on the lack of compelling evidence that mitochondrial SHP2 activity towards tyrosine phosphorylated mitochondrial proteins is functionally important. This limitation of functional validation does not preclude publication of a resource-type article, but should be discussed in the manuscript.

Referee #3 raises concerns about the lack of orthogonal validation of SHP2-interactors using antibody-based proximity labeling. The referee suggests being transparent about this limitation by adding a statement to the effect that novel interactors not validated by an orthogonal method (not only IP-mass spectrometry) could be real, reflecting the SHP2 interactome, but that artifacts cannot be excluded. The referee further suggests toning down the statement of having identified real interactors. These suggestions should be implemented as important information for future readers. Generally, conclusions on physiologically meaningful interactions and altered mitochondrial localization should be carefully phrased and limitations due to overexpression artefacts clearly articulated.

I suggest adding a "Limitations" statement at the end of the article to be fully transparent.

In our previous version of the manuscript, the 2nd and 3rd to last paragraphs of the Discussion section started to address some of these limitations. We have since expanded this section to highlight several key points raised by the reviewers. In particular, throughout the discussion, we have added points about the need for additional validation, as well as the caveats to using a single cell line, transient transfection, and over-expression. We also re-iterate, when talking about mitochondrial SHP2 at the end of the discussion, that those results will require further corroboration/validation, and that future studies of mitochondrial SHP2 will benefit from using disease-relevant cell lines with endogenous SHP2. Finally, we have removed the phrasing about identifying "real" or "true" interactors.

From the editorial side, there are also a few things that we need before we can proceed with the official acceptance of your study.

- Please describe your findings in the abstract in present tense.

We have updated the abstract.

- Please remove the main figures from the manuscript text.

We've removed the figures from the main text file.

- The PDF with EV figures and legends is not needed. The individual EV figure files and their legends in the manuscript text are sufficient.

We no longer include a PDF with the EF figures and legends. We still include the EV Figure legends in the main manuscript file, and we will upload the individual EV Figure files to the manuscript submission portal.

- Please reduce the number of keywords to 5 and place them after the Abstract.

We have reduced the number of keywords to 5 and placed them after the Abstract.

- Please use the header "Disclosure and Competing Interests Statement" instead of "Conflict of Interest".

We have updated our header.

- There is an error in the Appendix table of content: Appendix Figure S3 is listed twice while Appendix Figure S5 is missing).

Fixed - thanks for catching this!

- Appendix Figure S1, S4: please specify the nature of the replicates in the legend (biological, technical).

We have updated the figure legends to specify the nature of the replicates.

- It is sufficient to have the Appendix Figures in the Appendix PDF. The separately uploaded Appendix Figures are not needed. Moreover, the Appendix figure legends should be removed from the manuscript file.

Appendix figure legends have been deleted from the main text document.

- The Reagents and Tools table needs to be removed from the manuscript and uploaded as a separate Word file. Our production team will add it to the article during typesetting.

We have removed the Reagents and Tools table from the main text document and now provide this as a separate Word document.

- Source data: please upload each figure folder as a separate zip archive. Thank you.

Done.

- I did a spot check on the source data and noted for Figure 4D that you provided a Western blot for the IP that probes for SHP2 and MPZL1 on the same blot. Would this not be the more appropriate band/result to show, since you probed for both proteins on the same blot?

Thank you for raising this. We chose to show the IP MPZL1 band from a different blot, on which SHP2 is not probed, as we probed for phospho-MPZL1 on this blot. Since the antibodies for MPZL1 and phospho-MPZL1 target the same protein, we felt it was more appropriate to match these bands.

- For Figure 5C the Western blots are missing from the Source data folders, it seems.

We've added these blots to the source data folder.

- We perform a routine plagiarism check on all revised manuscripts. Here I noted that the paragraphs on "Purification of SH2 domains", "Purification of full-length SHP2 proteins", "SH2 specificity profiling" appear to be identical to the corresponding sections in Vlimmeren et al, PNAS 2024. This is perfectly fine, no need to reinvent the wheel for methods, but I suggest citing this paper.

Thank you for flagging this. We have cited this paper in the section headers to clarify the origins of these methods.

- Could you please cite the preprint from Akram et al 2024 in the text as (preprint: Akram et al, 2024)?

We have updated the citation in the text.

- Our data editors asked you to address the following points in the figure legends:

- Please note that the exact p values are not provided in the legends of figures 4B, D; 5C, F, H; EV3 E; EV5 A-C.

We have updated the figure legends to include the exact p-values for each of these figures.

- Please indicate the statistical test used for data analysis in the legends of figures 3E, 4D, E, F; 5A, 7C, D; S1 A, B; S3 B.

We have updated the figure legends to include this information.

- Please note that the box plots need to be defined in terms of centre, bounds of box and percentile in the legend of figure 6D

To make this panel less ambiguous, we opted to plot the data showing all datapoints, as this better represents the whole distribution than a box and whiskers plot.

- Please note that information related to n is missing in the legends of figures 4E, F; 5A, B, C; 6A, B, D, F; 7C, D, E; EV1 G; EV3 E, F; S3 A, B

We have added this information to the figure legends.

- Please note that the error bars are not defined in the legends of figures 5C, EV1 G

In all cases, the error bars represent standard deviation. We have added this information to the figure legends.

- Please note that the measure of center for the error bars needs to be defined in the legends of figures 4B, 6A, F; 7E, EV1 A, B, E; EV3 E, F; EV5 A-C

In all cases, the bar heights (center of error bars) represents the mean value. We have added this information to the figure legends.

- Figures 4B, 5C, 6A, B, F, 7E, EV1A, B, EV3E, F, EV5A, B, C miss a display of individual datapoints. Please add these in addition to the mean and error bars, if the number of 'n' permits.

We have added this where possible (5C and EV1A,B). For the other cases, the bar values represent the ratios of mean SHP2 and control samples, and the error bars represent propagated error. Therefore, we cannot plot individual values.

- Regarding the Author Contributions, we now use CRediT to specify the contributions of each author in the journal submission system. Therefore, please remove the Author Contributions from the manuscript file and make sure that the author contributions in our online manuscript tracking system are correct and up-to-date. The information you specified in the system will be automatically retrieved and typeset into the article. You can enter additional information in the free text box provided, if you wish. See also our [guide to authors](https://www.embopress.org/page/journal/14693178/authorguide#authorshipguidelines) <https://www.embopress.org/page/journal/14693178/authorguide#authorshipguidelines>.

We have removed this section from the manuscript and confirmed that the author contributions are correct in the online manuscript tracking system.

- Finally, EMBO Reports papers are accompanied online by
A) a short (1-2 sentences) summary of the findings and their significance,

This summary has been added to the manuscript file, after the Abstract and Keywords.

B) 2-3 bullet points highlighting key results and

These bullet points have been added to the manuscript file, after the Abstract, Keywords, and Summary.

C) a schematic summary figure that provides a sketch of the major findings (not a data image). Please provide the summary figure as a separate file in PNG or JPG format at a size of 550x300-600 pixels (width x height). Please note that the size is rather small and that text needs to be readable at the final size. Please send us this information along with the revised manuscript.

This figure has been separately uploaded in the manuscript tracking portal.

With kind regards,

=====

Referee #1:

In general, the authors have done a good job of addressing the reviewers' comments, adding some new data, including a comparison of their SHP2 interactome with that in BioGRID, stronger support for endogenous SHP2 co-localizing with mitochondria, and evidence that interaction of SHP2 with mitochondrial ATPAF1 is partially dependent on the phospho-Tyr23. However, although the correlative evidence for a mitochondrial role for SHP2 is suggestive, they still provide no compelling evidence that mitochondrial SHP2 activity towards tyrosine phosphorylated mitochondrial proteins is functionally important, even though some disease-associated, activating SHP2 mutants have been shown, as confirmed by the authors, to impact mitochondrial functions. Nevertheless, the expanded WT and mutant SHP2 interaction datasets reported here will undoubtedly be of value to the SHP2 community.

We thank the reviewer for their constructive feedback throughout this process. Although it is true that our study does not provide definitive evidence for SHP2 activity toward tyrosine-phosphorylated proteins in mitochondria, we are glad that the reviewer recognizes that our data are in line with past observations that SHP2 mutants can modulate mitochondrial function. We also appreciate that the reviewer sees our study and datasets as a valuable contribution to the SHP2 community.

Referee #3:

The authors have done a good job at revising the manuscript. We are satisfied with the authors' responses to our major comments 3, 3a, 3b, 3c, 3e and minor comments. However, we are not entirely satisfied with the responses to our comments 1, 2 and 3d. In our view, these remaining points need to be addressed before the publication.

We thank the reviewer for their constructive feedback, and we are glad that our revisions have addressed many of the reviewer's comments.

Remaining concerns

Regarding our comments 1 and 2:

The authors decided not to confirm their results using the orthogonal proximity labeling approach (antibody-based proximity labeling experiments). The authors claim that "Thus, our methodology is likely to be revealing true SHP2 interactions and biological functions."

In our view, the authors over-interpret their results. The authors selected 1) to use only HEK 293 cells, and 2) not to exclude false positives SHP2-binding partners by using as a control antibody-based proximity labeling, an approach that would be useful for excluding effects of TurboID fusion to SHP2 (especially for proteins in the mitochondria and the nucleus).

These are fair points about the limitations of our study. While we acknowledged the caveat of only using HEK293 cells in the prior version of the manuscript, we did not discuss the possibility that TurboID fusion could create artifacts. We have added sentences to the Discussion section indicating this point and also noting that orthogonal validation would be valuable in future studies.

The authors reported many novel SHP2-interacting partners. If true, these interactions may underlie currently unknown functions of SHP2 in the mitochondria and the nucleus, which cannot be explained by SHP2 canonical mechanisms as a phosphatase downstream of receptor activation. Therefore, the authors reliance on canonical mechanisms to explain novel SHP2 interactions is inadequate.

We agree that our results may be pointing to currently unknown functions of SHP2. At this stage, without a deeper exploration of each of these potentially new realms of SHP2 signaling, we cannot readily speculate about the specific mechanisms behind these potentially novel SHP2 functions.

The many novel binding partners of SHP2, e.g., mitochondrial and nuclear proteins, suggest molecular mechanisms that differ from those underlying receptor-SHP2 mechanisms. These are very interesting data; however, caution is needed. The newly added Figure EV4E also supports this perspective (see also the later part). Based on the authors' expertise, antibody-based proximity labeling could be an ideal orthogonal approach to address this issue.

We now acknowledge these points explicitly in an expanded series of sentences/paragraphs about limitations of our study, in the Discussion section.

It seems that the authors find it difficult to perform antibody-based proximity labeling experiments at present, although they clearly have considerable specific expertise. Therefore, as an alternative, we would suggest adding a statement to the effect that novel interactors not validated by an orthogonal method (not only IP-mass spectrometry) could be real, reflecting the SHP2 interactome, but artifacts cannot be excluded. Additionally, these findings and the cellular localizations of SHP2 should be interpreted with great caution. Our hope is that other investigators following up on new information in this paper be advised to confirm the results before proceeding.

We appreciate the reviewer's acceptance that we were unable to perform additional antibody-based proximity-labeling studies during the revision stages of this work. At the time of revision, several of the primary authors on this study had already left the lab, and significant funding uncertainty at our institution required that we make some judgement calls about how to expend resources.

We completely agree that additional orthogonal validations would be valuable before moving forward with mechanistic studies on any given hit from our TurboID experiments. We have also added sentences to the expanded Discussion section indicating that researchers should proceed with caution and orthogonally validate interesting hits, as there may be artifacts.

Regarding the concluding sentence in the Discussion ("Thus, our methodology is likely to be revealing true SHP2 interactions and biological functions."), please remove "true". This change is justified not only by the potential for artifacts mentioned above (my point 1) but also by the authors' selection to use only the HEK293 cell line (related to my point 2). The conclusion should reflect that the novel SHP2 interactions identified here may not represent the interactome of SHP2 across different, and more physiological cellular contexts.

We rephrased this to say, "Thus, our approach is likely capturing many biologically important SHP2 interactions and functions", both removing the word "true" and adding the qualifier "likely".

Regarding our comment 3d:

The authors did not provide the requested data. Figure EV4E, now provided by the authors, reveals that:

1) SHP2-WT seems to be expressed in only a small number of HEK293 cells, even with the very high transfection efficiency in these cells, and

2) Transfected cells expressing SHP2-WT exhibit a smeared cytoplasmic distribution pattern, suggesting extreme overexpression of SHP2.

These observations suggest the possibility that this paper detected SHP2 interactions in non-physiologic conditions of extremely high expression, where SHP2-WT was transfected. This also suggests that such extremely high expression levels may saturate physiological molecular transport and distribution pathways. Therefore, the changes in mitochondrial distribution of SHP2 mutants concluded in this study should be interpreted with caution. As the authors state, it would be necessary to control the expression level within an appropriate range, such as by using a stable mutant cell line. These limitations have already been added to the Discussion section, so although the authors' statements responding to this reviewer are insufficient, I believe that a minimum requirements was met.

We agree that the variability in transfection efficiency across cells could mean that some cells have very high SHP2 levels, and that this could artificially impact transport and localization. Given that other groups have also observed mitochondrial SHP2, we feel that, while this concern exists, our results are likely still reporting on some true SHP2 localization. Nonetheless, as noted by the reviewer, we discuss the limitations in our study that are associated expression levels and transient transfection in the Discussion section. We have also added extra phrasing to the final paragraph of the paper, where we mention future studies on mitochondrial SHP2, as follows, "These follow-up studies should be carried out using complementary approaches to proximity labeling, in disease-relevant cell lines with endogenously expressed wild-type SHP2 and SHP2 mutants."

Neel Shah
Columbia University
Chemistry
3000 Broadway
Havemeyer Hall, MC3105
New York, NY 10027
United States

Dear Dr. Shah,

I am very pleased to accept your manuscript for publication in the next available issue of EMBO reports. Thank you for your contribution to our journal.

Yours sincerely,
